# Global-Local Dirichlet Processes for Clustering Grouped Data in the Presence of Group-Specific Idiosyncratic Variables

**Arhit Chakrabarti** [1]   **Yang Ni** [1]   **Debdeep Pati** [2]   **Bani K. Mallick** [1]

## Abstract

We consider the problem of clustering grouped data for which the observations may include group-specific variables in addition to the variables that are shared across groups. This type of data is quite common; for example, in cancer genomic studies, molecular information is available for all cancers whereas cancer-specific clinical information may only be available for certain cancers. Existing grouped clustering methods only consider the shared variables but ignore valuable information from the group-specific variables. To allow for these group-specific variables to aid in the clustering, we propose a novel Bayesian nonparametric approach, termed global-local (GLocal) Dirichlet process, that models the "global-local" structure of the observations across groups. We characterize the GLocal Dirichlet process using the stick-breaking representation and the representation as a limit of a finite mixture model. We theoretically quantify the approximation errors of the truncated prior, the corresponding finite mixture model, and the associated posterior distribution. We develop a fast variational Bayes algorithm for scalable posterior inference, which we illustrate with extensive simulations and a TCGA pan-gastrointestinal cancer dataset.

## 1. Introduction

This article considers the clustering of grouped data that includes both shared variables across groups and group-specific *idiosyncratic* variables, as often seen in practice. For example, large-scale studies like The Cancer Genome Atlas (TCGA) provide molecular and clinical profiles across cancers, enabling a systematic pan-cancer classification. While molecular data (e.g., mRNA expression) are shared across tumors, clinical variables may be cancer-specific (e.g., prostate-specific antigen for prostate cancer). These cancer-specific clinical variables may provide valuable information in clustering. Moreover, it is of scientific interest to investigate if patients with different clinical characteristics show differential gene expression patterns. Thus, while it is desirable to utilize both molecular and clinical information to identify pan-cancer subpopulations, their varying availability across cancers makes it a challenging problem. This paper introduces a novel Bayesian nonparametric method for clustering such grouped data.

The *Dirichlet process* (DP, Ferguson, 1973) is at the core of numerous model-based Bayesian nonparametric clustering methods (Antoniak, 1974; Escobar & West, 1995; Mallick & Walker, 1997; Hjort et al., 2010; Müller et al., 2015). One of the advantages of DP mixture models (Lo, 1984; Escobar & West, 1995; Maceachern & Müller, 1998) is its ability to perform clustering without having to fix the number of clusters *a priori*.

When analyzing grouped data (e.g., tumor tissues of different origins in our application), one could naively apply a separate DP mixture model to each group, treating them independently. However, it is often desirable to identify group-specific clusters while allowing the groups to be linked so that clusters are comparable across groups. The hierarchical Dirichlet process (HDP, Teh et al., 2006) is a remarkable contribution in this direction. The HDP formulation relies on modeling groups of observations using distinct DPs with a common base measure, which, in turn, is itself a realization from another DP. Since the draws from this DP are almost surely discrete, all group-specific distributions share the same set of atoms. Another classic group clustering method is the nested Dirichlet process (nested DP, Rodríguez et al., 2008), which focuses on simultaneously clustering groups as well as observations within each group cluster. However, the nested DP is known to suffer from a degeneracy property (Camerlenghi et al., 2019) – two distributions sharing even one atom in their support are automatically assigned to the same cluster.

Both the HDP and the nested DP fall under the general

---

[1] Department of Statistics, Texas A&M University, College Station, TX, USA [2] Department of Statistics, University of Wisconsin, Madison, WI, USA. Correspondence to: Arhit Chakrabarti <arhit.chakrabarti@stat.tamu.edu>.

*Proceedings of the $42^{nd}$ International Conference on Machine Learning*, Vancouver, Canada. PMLR 267, 2025. Copyright 2025 by the author(s).

framework of dependent DP (MacEachern, 1999; 2000). See Quintana et al., 2022 for a recent review of different dependent DPs. Several recent works (Beraha et al., 2021; Balocchi et al., 2022; Bi & Ji, 2023; Lijoi et al., 2023) have been proposed to take advantage of the cluster-sharing feature of the HDP and the group-clustering feature of the nested DP. In contrast to methods relying on the HDP or its variants, some other works rely on models with additive structure or common atoms (Camerlenghi et al., 2019; Denti et al., 2023; D'Angelo et al., 2023; Chandra et al., 2023; D'Angelo & Denti, 2024). However, the aforementioned methods assume that the observations across the groups are measured on the same set of variables (with possible missing values for some variables within a group). In this paper, we develop a grouped clustering framework that explicitly accounts for group-specific variables.

Let $\boldsymbol{x}_{ji}$ represent observation $i$ from group $j$. They are assumed to be *partially exchangeable* (de Finetti, 1938), entailing that observations are exchangeable within but not across groups. Partition $\boldsymbol{x}_{ji}$ into $\left(\boldsymbol{x}_{ji}^L, \boldsymbol{x}_{ji}^G\right)$, where $\boldsymbol{x}_{ji}^G$ includes variables shared across groups (e.g., age, sex, gene expression) and $\boldsymbol{x}_{ji}^L$ includes group-specific variables (e.g., prostate-specific antigen for prostate cancer). Unlike existing group clustering methods, which consider a common set of variables across groups, we distinguish between *global* ($\boldsymbol{x}_{ji}^G$) and *local* ($\boldsymbol{x}_{ji}^L$) variables. We propose a Bayesian nonparametric approach to cluster observations while incorporating this "global-local" structure.

To model both global and local variables, we let the group-specific random measure $G_j$ be supported on a shared subspace across groups and an idiosyncratic subspace unique to each group. Specifically, we assume that conditionally on $\alpha$ and $V$, $G_j \sim \mathrm{DP}(\alpha, U_j \otimes V)$, where $U_j$ is a group-specific base measure, $V$ is a common base measure, and $\otimes$ is the measure product. To share clustering information across groups, we assume $V$ is also DP-distributed. This model, termed the global-local (GLocal) DP, allows $G_j$ to share atoms in the common subspace across groups, enabling global clustering. As we will explain later, the idiosyncratic base measure $U_j$ refines the global clusters into local clusters using the local variables.

GLocal DP generalizes HDP by handling group-specific variables. Unlike HDP, the GLocal DP does not assume group exchangeability, as $G_j$ has a group-specific base measure. Although HDP can still be generalized to avoid the exchangeability of the groups by introducing group-specific concentration hyperparameters, the variables share the same support and, thus, HDP cannot be used to cluster observations with varying variable sets across groups. Addressing this limitation and enabling the clustering of such data constitute the main motivation of the proposed GLocal DP.

Recently, Dinari & Freifeld (2020) proposed the versatile hierarchical Dirichlet process mixture model (vHDPMM) for modeling grouped data that includes both shared and group-specific variables. Although the modeling motivation behind the vHDPMM is closely related to that of our proposed GLocal DP, the two approaches differ fundamentally in their underlying constructions. We detail these differences in Section 2.2. Importantly, the models are not special cases of one another.

**Summary**: **First**, we propose a general Bayesian nonparametric approach, GLocal DP, to incorporate group-specific local variables for clustering of grouped data. **Second**, we provide two characterizations of GLocal DP, each providing a different perspective. **Third**, we provide some theoretical results relating to the use of finite truncation of the GLocal DP in posterior inference. **Fourth**, we develop an efficient variational Bayes algorithm for scalable inference. **Finally**, we demonstrate our model through experiments on synthetic data as well as a real TCGA pan-gastrointestinal cancer dataset. Both simulations and real data analysis demonstrate excellent performance of our model in identifying clusters of observations shared across groups. Furthermore, our method highlights the importance of incorporating group-specific variables in refining the shared clusters into smaller subclusters through the local variables. All codes used for simulations and real data analysis, as well as the datasets themselves, are available here.

## 2. Model

Because of space limit, we provide an overview of the DP mixture model for a single population/group and the HDP mixture model for multiple groups in the Appendix Section A. When data contain varying sets of variables across groups, the HDP prior (25) is not appropriate (e.g., $G_j$ does not have the correct support). We address this challenge of clustering grouped data with varying variable sets by proposing a joint distribution for $G_j$'s that incorporates both local and global variables.

Recall that $\boldsymbol{x}_{ji} = \left(\boldsymbol{x}_{ji}^L, \boldsymbol{x}_{ji}^G\right)$ denotes the $i$th observation from the group $j$. We assume that each observation is drawn independently from a mixture model with factor $\boldsymbol{\theta}_{ji}$. Similar to the observations, factor $\boldsymbol{\theta}_{ji}$ can be partitioned into local and global factors, $\boldsymbol{\theta}_{ji} = \left(\boldsymbol{\theta}_{ji}^L, \boldsymbol{\theta}_{ji}^G\right)$. By later construction, there is a positive prior probability that the global factors are equal across groups (e.g., $\boldsymbol{\theta}_{ji}^G = \boldsymbol{\theta}_{j'i'}^G$), thereby inducing the sharing of global clusters. Furthermore, local factors ($\boldsymbol{\theta}_{ji}^L$) can modify the global clusters and may refine them into smaller local clusters.

Let $F(\boldsymbol{x}_{ji} \mid \boldsymbol{\theta}_{ji})$ be the distribution of $\boldsymbol{x}_{ji}$, conditional on factor $\boldsymbol{\theta}_{ji}$. For simplicity, we assume that this distribution can be expressed in a factorized form:

$$F(\boldsymbol{x}_{ji} \mid \boldsymbol{\theta}_{ji}) = F_1(\boldsymbol{x}_{ji}^L \mid \boldsymbol{\theta}_{ji}^L)\, F_2(\boldsymbol{x}_{ji}^G \mid \boldsymbol{\theta}_{ji}^G). \qquad (1)$$

Here, $F_1(\boldsymbol{x}_{ji}^L \mid \boldsymbol{\theta}_{ji}^L)$ represents the conditional distribution of local variables $\boldsymbol{x}_{ji}^L$, given local factors $\boldsymbol{\theta}_{ji}^L$, while $F_2(\boldsymbol{x}_{ji}^G \mid \boldsymbol{\theta}_{ji}^G)$ represents the conditional distribution of global variables $\boldsymbol{x}_{ji}^G$, given global factors $\boldsymbol{\theta}_{ji}^G$. This implies that $\boldsymbol{x}_{ji}^G$ and $\boldsymbol{x}_{ji}^L$ are conditionally independent, although they are not independent marginally. If additional dependency between $\boldsymbol{x}_{ji}^G$ and $\boldsymbol{x}_{ji}^L$ is desired, $F_1(\boldsymbol{x}_{ji}^L \mid \boldsymbol{\theta}_{ji}^L)$ in (1) could be replaced with $F_1(\boldsymbol{x}_{ji}^L \mid \boldsymbol{x}_{ji}^G, \boldsymbol{\theta}_{ji}^L)$; however, this direction is not pursued in this paper. Let $G_j$ denote the group-specific prior distribution for factors $\boldsymbol{\theta}_{ji}$. We assume that the factors are conditionally independent given $G_j$, resulting in the following probability model:

$$\boldsymbol{\theta}_{ji} = \left(\boldsymbol{\theta}_{ji}^L, \boldsymbol{\theta}_{ji}^G\right) \mid G_j \sim G_j \qquad (2)$$

Consider $(\Theta_j, \mathcal{A}_j)$ as the measurable space associated with the local factors specific to group $j$, and $(\Omega, \mathcal{B})$ as the measurable space corresponding to the shared global factors across all groups. The GLocal DP defines a collection of random probability measures $G_j$, one for each group, on the product space $(\Theta_j \times \Omega, \mathcal{A}_j \otimes \mathcal{B})$,

$$G_j \mid \alpha, V \sim \mathrm{DP}(\alpha, U_j \otimes V), \qquad (3)$$

where $\alpha$ is the positive concentration parameter. The base measure $U_j \otimes V$ is a random product probability measure of the local measure $U_j$ and the global measure $V$ defined on the product space $(\Theta_j \times \Omega, \mathcal{A}_j \otimes \mathcal{B})$. In other words, $U_j$ is defined on $(\Theta_j, \mathcal{A}_j)$ and $V$ is defined on $(\Omega, \mathcal{B})$. To allow for the sharing of global factors across groups, we further assume,

$$V \mid \gamma \sim \mathrm{DP}(\gamma, H), \qquad (4)$$

where $\gamma$ and $H$ are the concentration parameter and base probability measure, respectively. Equations (1) and (2) along with the prior specifications given in (3) and (4) complete the specification of the proposed GLocal DP mixture model. The GLocal DP reduces to the HDP when group-specific local variables (and consequently local factors) are absent across all groups. However, when local variables are present, they play a significant role in clustering grouped data. In addition to defining group-specific local clusters, the local variables can also influence the clustering of global variables across populations, as discussed at the end of Section 2.1.1. This makes our method different from the HDP, even at the global level. Furthermore, following Ascolani et al. 2022, we assume non-informative gamma priors on the concentration parameters to avoid inconsistencies in the estimation of the number of clusters in DP mixture models (Miller & Harrison, 2013; Yang et al., 2020).

## 2.1. Representations

### 2.1.1. THE STICK-BREAKING REPRESENTATION

Since the global measure $V$ is distributed as a DP, it can be expressed using a stick-breaking representation (Sethuraman, 1994),

$$V = \sum_{k=1}^{\infty} \beta_k \delta_{\phi_k}, \qquad (5)$$

where $\boldsymbol{\beta} = (\beta_k)_{k=1}^{\infty} \sim \mathrm{GEM}(\gamma)$ and $\phi_k \overset{iid}{\sim} H$ independent of $\boldsymbol{\beta}$. Here GEM stands for Griffiths, Engen and McCloskey distribution (Pitman, 2002). Furthermore, as each $G_j$ is distributed as a DP, a similar stick-breaking representation gives,

$$G_j = \sum_{t=1}^{\infty} \pi_{jt} \delta_{\psi_{jt}}, \qquad (6)$$

where $\boldsymbol{\pi}_j = (\pi_{jt})_{t=1}^{\infty} \sim \mathrm{GEM}(\alpha)$ and $\psi_{jt} \mid V \overset{ind}{\sim} U_j \otimes V$ independent of $\boldsymbol{\pi}_j$. Since each factor $\boldsymbol{\theta}_{ji}$ is distributed according to $G_j$, it takes on the value $\psi_{jt} = \left(\psi_{jt}^L, \psi_{jt}^G\right)$ with probability $\pi_{jt}$, where $\psi_{jt}^L \overset{iid}{\sim} U_j$ and $\psi_{jt}^G \mid V \overset{iid}{\sim} V$. Because $V$ has support at the points $\boldsymbol{\phi} = (\phi_k)_{k=1}^{\infty}$, the marginal distribution of each $G_j$ with $\psi_{jt}^L$ marginalized out also has support at these points through $\psi_{jt}^G$. That is, $\left(\psi_{jt}^G\right)_{t=1}^{\infty}$ are necessarily the same as $(\phi_k)_{k=1}^{\infty}$. Indeed, $\psi_{jt}^G$ takes on the value $\phi_k$ with probability $\beta_k$. This sharing of global factors across the groups facilitates the sharing of clustering of the global variables. To make the clustering aspect of our model explicit, we introduce the latent variables $t_{ji}$ and $k_{jt}$, where

$$t_{ji} \mid \boldsymbol{\pi}_j \overset{ind}{\sim} \boldsymbol{\pi}_j, \qquad (7)$$

$$k_{jt} \mid \boldsymbol{\beta} \overset{ind}{\sim} \boldsymbol{\beta}, \qquad (8)$$

such that, conditional on the latent indicators $t_{ji}$ and $(k_{jt})_{t=1}^{\infty}$, we have $\boldsymbol{x}_{ji} \sim F_1(\boldsymbol{x}_{ji}^L \mid \psi_{jt_{ji}}^L) F_2(\boldsymbol{x}_{ji}^G \mid \phi_{k_{jt_{ji}}})$. We refer to the latent indicator $k_{jt_{ji}}$ as the *global-level* cluster label as it indicates the shared clustering across groups. For example, if $k_{jt_{ji}} = k_{j't_{j'i'}}$, the $i$-th observation from group $j$ and the $i'$-th observation from group $j'$ belong to the same global cluster. Similarly, we refer to the latent variable $t_{ji}$ as the *local-level* cluster label as it identifies the finer sub-clusters within each group. Specifically, for two observations $i$ and $i'$, if $t_{ji} \neq t_{ji'}$, then the local variable(s) in group $j$ refines the corresponding global clusters $k_{jt_{ji}}$ and $k_{jt_{ji'}}$ into two distinct sub-clusters. The dissimilarity in the local variable(s) between observations $i$ and $i'$ drives this refinement, providing insight into how the local variables influence the clustering of global variables. With these two sets of latent indicators, we obtain an equivalent representation of the GLocal DP mixture via the following conditional distributions:

$$\begin{aligned}
\boldsymbol{\beta} &\sim \mathrm{GEM}(\gamma), & k_{jt} &\sim \boldsymbol{\beta}, \\
\boldsymbol{\pi}_j &\sim \mathrm{GEM}(\alpha), & t_{ji} &\sim \boldsymbol{\pi}_j, \\
\phi_k &\sim H, & \psi_{jt}^L &\sim U_j, \\
\boldsymbol{x}_{ji} &\sim F_1(\boldsymbol{x}_{ji}^L \mid \psi_{jt_{ji}}^L) F_2(\boldsymbol{x}_{ji}^G \mid \phi_{k_{jt_{ji}}}).
\end{aligned}$$
$$(9)$$

We remark that our clusters have hierarchical structure where the local-level clusters (given by $t_{ji}$) are nested within the global-level clusters (corresponding to $k_{jt_{ji}}$). This hierarchical nature of our clusters indicates that the local variables help refine the global clusters. In our motivating pan-cancer application, this plays a pivotal role in the finer understanding of molecular subpopulations modified by cancer-specific clinical variables. The Figure 5 in the Appendix Section B shows the graphical model representation of the GLocal DP mixture model. Clearly, given the data $\{\boldsymbol{x}_{ji}^L, \boldsymbol{x}_{ji}^G\}$, the marginalized global-level assignment of observation $i$ in group $j$, denoted $k_{ji}$, is influenced by the corresponding local variables $\boldsymbol{x}_{ji}^L$. Therefore, the local variables can impact the clustering at the global level.

### 2.1.2. THE INFINITE LIMIT OF FINITE MIXTURE MODELS

Alternatively to the stick-breaking representation, the GLocal DP mixture model in (9) can be derived as the infinite limit of a finite mixture model. Specifically, consider the following finite mixture model,

$$
\begin{aligned}
\boldsymbol{\beta} &\sim \mathrm{Dir}(\gamma/K, \ldots, \gamma/K), & k_{jt} &\sim \boldsymbol{\beta}, \\
\boldsymbol{\pi}_j &\sim \mathrm{Dir}(\alpha/T, \ldots, \alpha/T), & t_{ji} &\sim \boldsymbol{\pi}_j, \\
\phi_k &\sim H, & \psi_{jt}^L &\sim U_j, \\
\boldsymbol{x}_{ji} &\sim F_1(\boldsymbol{x}_{ji}^L \mid \psi_{jt_{ji}}^L) F_2(\boldsymbol{x}_{ji}^G \mid \phi_{k_{jt_{ji}}}),
\end{aligned}
\tag{10}
$$

with $K \leq T$, where $\boldsymbol{\beta}$ is the global mixing proportion vector, $\boldsymbol{\pi}_j$ is the group-specific mixing proportion vector, $K$ is the number of global mixture components, and $T$ is the number of local mixture components. As $K \to \infty$ the model converges to the infinite limit, which is precisely the proposed GLocal DP mixture model as shown in Appendix Section C.

### 2.2. Comparison with Versatile Hierarchical Dirichlet Process

The vHDPMM (Dinari & Freifeld, 2020) was introduced to model grouped data comprising both shared and group-specific variables/features, aligning in motivation with our proposed GLocal DP. Despite this similarity in modeling objectives, there are crucial differences in the modeling formulations of the two approaches. Notably, the GLocal DP mixture model and the vHDPMM are not special cases of one another. To model the global (shared) variables $\boldsymbol{x}_{ji}^G$, Dinari & Freifeld (2020) employ a *HDP-type* mixture model of the form:

$$
p\big(\{\boldsymbol{x}_{ji}^G\}_{i=1}^{n_j}|\boldsymbol{\theta}, \boldsymbol{\pi}_j\big) = \prod_{i=1}^{n_j} \sum_{k=1}^{\infty} \pi_{jk} f(\boldsymbol{x}_{ji}^G; \theta_k),
\tag{11}
$$

where $\pi_{jk} > 0$ for all $k$, $\sum_{k=1}^{\infty} \pi_{jk} = 1$, $f$ denotes a group-independent probability density function (pdf) or probability mass function (pmf) parameterized by $\theta_k$, with $\theta_k \stackrel{iid}{\sim} H$ for

some base measure $H$, and the mixing proportions $\boldsymbol{\pi}_j \sim \mathrm{GEM}(\alpha)$. Introducing latent allocation variables, $\boldsymbol{z}_j = (z_{j1}, \ldots, z_{jn_j})$, with $z_{ji} = k$ if and only if $\boldsymbol{x}_{ji}^G$ is drawn from global component $k$, equivalently, (11) is written as,

$$
\begin{aligned}
z_{ji} &\stackrel{iid}{\sim} \mathrm{Cat}(\boldsymbol{\pi}_j), \quad i = 1, \ldots, n_j, \\
p(\{\boldsymbol{x}_{ji}^G\}_{i=1}^{n_j}|\boldsymbol{\theta}, \boldsymbol{z}_j) &= \prod_{i=1}^{n_j} f(\boldsymbol{x}_{ji}^G; \theta_{z_{ji}}).
\end{aligned}
\tag{12}
$$

The global cluster $k$ is defined as $c_k = (\boldsymbol{x}_{ji}^G)_{z_{ji}=k, \, j=(1,\ldots,J), \, i=(1,\ldots,n_j)}$ and let $K$ be a latent random variable denoting the number of global clusters. Furthermore, for each $j = 1, \ldots, J$ and each $k = 1, \ldots, K$, $s_j^k = (\boldsymbol{x}_{ji}^L)_{i:z_{ji}=k}$ is defined as the collection of local features having the global features in global cluster $k$. Consequently, each $s_j^k$ is modeled with an infinite mixture model as,

$$
p(s_j^k|\boldsymbol{\theta}_j^k, \boldsymbol{\pi}_j^k) = \prod_{i:z_{ji}=k} \sum_{w=1}^{\infty} \pi_{jw}^k f_j(\boldsymbol{x}_{ji}^L; \theta_{jw}^k),
\tag{13}
$$

where $\pi_{jw}^k > 0$ for all $k$, $\sum_{w=1}^{\infty} \pi_{jw}^k = 1$, $f_j$ is a group-specific pdf or pmf parametrized by $\theta_{jw}^k$, $\boldsymbol{\pi}_j^k \sim \mathrm{GEM}(\eta)$, and $\theta_{jw}^k \stackrel{ind}{\sim} L_j$. Equivalently, using hidden local clusters, (13) is defined as,

$$
\begin{aligned}
z_{ji}^l &\stackrel{iid}{\sim} \mathrm{Cat}(\boldsymbol{\pi}_j^k), \quad \forall i \text{ s.t. } z_{ji} = k, \\
p(\boldsymbol{s}_j^k|\boldsymbol{\theta}_j^k, \boldsymbol{z}_j^l) &= \prod_{i:z_{ji}=k} f_j(\boldsymbol{x}_{ji}^L; \theta_{jz_{ji}^l}^k).
\end{aligned}
\tag{14}
$$

In summary, the vHDPMM is defined hierarchically by first modeling the shared global variables and then conditionally modeling the local variables conditional on the global clusters. Contrarily, our proposed GLocal DP mixture model is defined jointly as in (9). Using our formulation, for two distinct observations $i$ and $i'$ in the same group $j$, if $t_{ji} = t_{ji'}$, then automatically, they share the same global clusters i.e., $k_{jt_{ji}} = k_{jt_{ji'}}$. In other words, if the $i$-th and the $i'$-th observation from group $j$ share the same local cluster, then they also belong to the same global cluster. However, this is not the case for the vHDPMM of Dinari & Freifeld, 2020, where the local clusters for observations $i$ and $i'$ are defined conditional on their global clusters. In other words, for any group $j$, if $i \in s_j^k$ and $i' \in s_j^{k'}$, where $k \neq k'$, then the two observations $i$ and $i'$ cannot have the same global cluster, even if they share the same local feature. Furthermore, our model exactly reduces to the HDP in the absence of local variables for all the groups. However, the vHDPMM, even in the absence of local variables for all the groups, is not exactly the HDP mixture model. Additionally, the posterior inference procedures for the GLocal DP and the vHDPMM are distinct, as discussed at the end of Section 4.

## 3. Truncation Approximation Bounds

In Section 2.1.2, we presented the finite mixture model representation of the GLocal DP, derived from the finite truncation of the GLocal DP prior. The finite truncation is critical in many Bayesian nonparametric posterior inference algorithms including ours. Therefore, it is important to evaluate the error arising from the truncated GLocal DP prior, the corresponding GLocal DP mixture model, and the resulting posterior distribution.

We recall that $G_j | \alpha, V \sim \mathrm{DP}(\alpha, U_j \otimes V)$, where $V | \gamma \sim \mathrm{DP}(\gamma, H)$. We hierarchically define the truncated versions of $G_j$ as follows,

$$V^K = \sum_{k=1}^{K} \beta_k^K \delta_{\phi_k}, \qquad (15)$$

where $\phi_k \overset{iid}{\sim} H$, $k = 1, \ldots, K$, and

$$\beta_k^K = \begin{cases} \beta_k & \text{if } k \leq K-1, \\ 1 - \sum_{k=1}^{K-1} \beta_k & \text{if } k = K, \end{cases} \qquad (16)$$

and

$$G_j^{T,K} = \sum_{t=1}^{T} \pi_{jt}^{T,K} \delta_{\psi_{jt}}, \qquad (17)$$

where $\psi_{jt} \overset{ind}{\sim} U_j \otimes V^K$, $t = 1, \ldots, T$, and

$$\pi_{jt}^{T,K} = \begin{cases} \pi_{jt} & \text{if } t \leq T-1, \\ 1 - \sum_{t=1}^{T-1} \pi_{jt} & \text{if } t = T. \end{cases} \qquad (18)$$

Here $T, K > 0$ define the truncation levels for the different random probability measures.

Consider $J$ groups, each of them containing $n_j$ observations, $j = 1, \ldots, J$. Denote by $(\boldsymbol{x}_j^L, \boldsymbol{x}_j^G) \equiv \boldsymbol{x}_j = (\boldsymbol{x}_{j1}, \ldots, \boldsymbol{x}_{jn_j})$ the collection of all observations from the $j$-th group arising from the mixture model $\boldsymbol{x}_{ji} | \boldsymbol{\theta}_{ji} \sim F(\cdot | \boldsymbol{\theta}_{ji})$ with $\boldsymbol{\theta}_{ji} | G_j \sim G_j$ where the $G_j$'s are generated according to the proposed GLocal DP. Let $f(\cdot | \boldsymbol{\theta}_{ji})$ be the density of $F(\cdot | \boldsymbol{\theta}_{ji})$ with respect to some dominating measure. We assume that $\boldsymbol{\theta}_{ji} \in (\Theta_j \times \Omega)$, where $(\Theta_j \times \Omega)$ is a Polish space equipped with its corresponding Borel $\sigma$–field $\mathcal{A}_j \otimes \mathcal{B}$. Finally, we denote by $\boldsymbol{x} = (\boldsymbol{x}_1, \ldots, \boldsymbol{x}_J)$, the vector containing the observations from all $J$ groups. Define

$$P^{\infty,\infty}(\boldsymbol{\theta}) =$$

$$\int_{\Omega} \left[ \prod_{j=1}^{J} \int_{\Theta_j} \left\{ \prod_{i=1}^{n_j} P(\boldsymbol{\theta}_{ji}|G_j) \right\} P^{\infty}(dG_j|V) \right] P^{\infty}(dV) \qquad (19)$$

and

$$P^{T,K}(\boldsymbol{\theta}) =$$

$$\int_{\Omega} \left[ \prod_{j=1}^{J} \int_{\Theta_j} \left\{ \prod_{i=1}^{n_j} P(\boldsymbol{\theta}_{ji}|G_j) \right\} P^T(dG_j|V) \right] P^K(dV), \qquad (20)$$

as the prior distribution of the parameters $\boldsymbol{\theta}$ under the GLocal DP and its corresponding truncated version after integrating out the random distributions. Here $P(\boldsymbol{\theta}_{ji} \mid G_j)$ denotes the prior distribution of the parameters $\boldsymbol{\theta}_{ji}$ given the random measure $G_j$. Furthermore, let $m^{\infty,\infty}(\boldsymbol{x})$ and $m^{T,K}(\boldsymbol{x})$ denote the marginal distribution of the data $\boldsymbol{x}$ derived from these priors. Then we have the following result.

**Proposition 3.1.** *Let $P^{\infty,\infty}(\boldsymbol{\theta})$ and $P^{T,K}(\boldsymbol{\theta})$ denote the prior distribution of the parameters $\boldsymbol{\theta}$ under the GLocal DP prior and its corresponding truncated version with the random measures integrated out. Furthermore, let $m^{\infty,\infty}(\boldsymbol{x})$ and $m^{T,K}(\boldsymbol{x})$ denote the marginal distribution of the data $\boldsymbol{x}$, derived from these priors. Then,*

$$\int_{\mathcal{X}^N} \left| m^{T,K}(\boldsymbol{x}) - m^{\infty,\infty}(\boldsymbol{x}) \right| d\boldsymbol{x} \leq$$

$$\int_{\Xi^N} \left| P^{T,K}(\boldsymbol{\theta}) - P^{\infty,\infty}(\boldsymbol{\theta}) \right| d\boldsymbol{\theta} \leq \epsilon^{T,K}(\alpha,\gamma),$$

*where*

$$\epsilon^{T,K}(\alpha,\gamma) = 4 \left[ 1 - \left\{ \left( 1 - \left( \frac{\alpha}{1+\alpha} \right)^{T-1} \right) \right\}^N \times \right.$$

$$\left. \times \left\{ \left( 1 - \left( \frac{\gamma}{1+\gamma} \right)^{K-1} \right) \right\}^N \right],$$

*$N = n_1 + \cdots + n_J$, $\Xi^N = \prod_{j=1}^{J}(\Theta_j \times \Omega)^{n_j}$, and $\mathcal{X}^N$ denotes the sample space of observations $\boldsymbol{x}$.*

Note that the bounds approach zero in the limit and hence, the truncated prior and the prior predictive distribution ( marginal data distribution) converge in total variation (and therefore in distribution) to the GLocal DP. Furthermore, the approximation errors decay exponentially in both $T$ and $K$. Consequently, we have the following result relating to the posterior distribution of the parameters $\boldsymbol{\theta}$ under the truncated GLocal DP prior.

**Proposition 3.2.** *The posterior distribution of the parameters $\boldsymbol{\theta}$ under the GLocal DP prior and its truncated version,*

$$\pi^{\infty,\infty}(\boldsymbol{\theta}|\boldsymbol{x}) = \frac{f(\boldsymbol{x}|\boldsymbol{\theta})P^{\infty,\infty}(\boldsymbol{\theta})}{m^{\infty,\infty}(\boldsymbol{x})},$$

$$\pi^{T,K}(\boldsymbol{\theta}|\boldsymbol{x}) = \frac{f(\boldsymbol{x}|\boldsymbol{\theta})P^{T,K}(\boldsymbol{\theta})}{m^{T,K}(\boldsymbol{x})},$$

*satisfies*

$$\int_{\mathcal{X}^N} \int_{\Xi^N} \left| \pi^{T,K}(\boldsymbol{\theta}|\boldsymbol{x}) - \pi^{\infty,\infty}(\boldsymbol{\theta}|\boldsymbol{x}) \right| m^{\infty,\infty}(\boldsymbol{x}) \, d\boldsymbol{\theta} \, d\boldsymbol{x}$$

$$= \mathcal{O}\left( \epsilon^{T,K}(\alpha,\gamma) \right).$$

Thus, Proposition 3.2 tells us that the posterior distribution for $\boldsymbol{\theta}$ under the truncated GLocal DP is exponentially accurate when integrated with respect to the marginal density of the data $m^{\infty,\infty}(\boldsymbol{x})$ under the original GLocal DP.

## 4. Variational Posterior Inference

The standard posterior inference approach is Markov chain Monte Carlo (MCMC). However, it is well-known that MCMC methods have limited scalability when dealing with large datasets and/or in high-dimensional settings. We instead develop a variational posterior inference (VI) algorithm for scalable computation, which aims to find, among a set of simple distributions (called variational distributions), the one that minimizes the Kullback-Leibler (KL) divergence from the posterior distribution, which is equivalent to maximizing the evidence lower bound (ELBO). We adopt a mean-field approach, which assumes the variational distributions are factorized. Furthermore, we assume a multivariate Gaussian likelihood for both global and local variables, although, this approach can be adapted in a straightforward manner whenever the data distribution is a member of the exponential family, as discussed in Blei & Jordan, 2006. We refer the reader to Appendix Section F for more details on the assumed form of the likelihood, prior distributions, and an alternative finite mixture model representation of the GLocal DP that simplifies the VI algorithm.

Following Blei & Jordan (2006), we use a truncated variational distribution to deal with the nonparametric mixture, where the truncation levels are denoted by $T$ and $K$. Specifically, we assume,

$$q(\boldsymbol{t}, \boldsymbol{k}, \boldsymbol{u}, \boldsymbol{v}, \boldsymbol{\phi}, \boldsymbol{\psi}, \alpha, \gamma; \boldsymbol{\lambda}) =$$

$$\prod_{j=1}^{J} \prod_{i=1}^{n_j} q(t_{ji}; \{\xi_{jit}\}_{t=1}^{T}) \prod_{j=1}^{J} \prod_{t=1}^{T} q(k_{jt}; \{\rho_{jtl}\}_{l=1}^{K}) \times$$

$$\prod_{j=1}^{J} \prod_{t=1}^{T-1} q(u_{jt}; \bar{a}_{jt}, \bar{b}_{jt}) \prod_{k=1}^{K-1} q(v_k; \bar{a}_k, \bar{b}_k) \times$$

$$\prod_{k=1}^{K} q(\boldsymbol{\mu}_k, \boldsymbol{\Lambda}_k; \boldsymbol{m}_k, \lambda_k, c_k, \boldsymbol{D}_k) \, q(\gamma; r_1, r_2) \times$$

$$\prod_{j=1}^{J} \prod_{t=1}^{T} q(\boldsymbol{\mu}_{jt}, \boldsymbol{\Lambda}_{jt}; \boldsymbol{m}_{jt}, \lambda_{jt}, c_{jt}, \boldsymbol{D}_{jt}) \, q(\alpha; s_1, s_2),$$

where $q(t_{ji}; \{\xi_{jit}\}_{t=1}^{T})$ and $q(k_{jt}; \{\rho_{jtl}\}_{l=1}^{K})$ are multinomial distributions; $q(v_k; \bar{a}_k, \bar{b}_k)$ and $q(u_{jt}; \bar{a}_{jt}, \bar{b}_{jt})$ are beta distributions, and they are such that $q(v_K = 1) = 1$ and $q(v_g = 0) = 1$ for $g > K$ and similarly, $q(u_{jT} = 1) = 1$ and $q(u_{jh} = 0) = 1$ for $h > T$; $q(\alpha; s_1, s_2)$ and $q(\gamma; r_1, r_2)$ are gamma distributions; $q(\boldsymbol{\mu}_k, \boldsymbol{\Lambda}_k; \boldsymbol{m}_k, \lambda_k, c_k, \boldsymbol{D}_k)$ and $q(\boldsymbol{\mu}_{jt}, \boldsymbol{\Lambda}_{jt}; \boldsymbol{m}_{jt}, \lambda_{jt}, c_{jt}, \boldsymbol{D}_{jt})$ are normal-Wishart distributions. Under this representation, the set of latent variables is

$\boldsymbol{\Theta} = \left(\boldsymbol{t}, \boldsymbol{k}, \boldsymbol{u}, \boldsymbol{v}, \{\boldsymbol{\mu}_k, \boldsymbol{\Lambda}_k\}_{k=1}^{K}, \{\{\boldsymbol{\mu}_{jt}, \boldsymbol{\Lambda}_{jt}\}_{t=1}^{T}\}_{j=1}^{J}, \alpha, \gamma\right)$ and the set of variational parameters is $\boldsymbol{\lambda} = (\boldsymbol{\rho}, \boldsymbol{\xi}, \bar{\boldsymbol{a}}, \bar{\boldsymbol{b}}, \{\bar{\boldsymbol{a}}_j\}_{j=1}^{J}, \{\bar{\boldsymbol{b}}_j\}_{j=1}^{J}, s_1, s_2, r_1, r_2, \boldsymbol{m}, \boldsymbol{t}, \boldsymbol{c}, \boldsymbol{D}, \{\boldsymbol{m}_j\}_{j=1}^{J}, \{\boldsymbol{t}_j\}_{j=1}^{J}, \{\boldsymbol{c}_j\}_{j=1}^{J}, \{\boldsymbol{D}_j\}_{j=1}^{J})$. Finally, the variational parameters $\boldsymbol{\lambda}^\star$ that maximize the ELBO are found by the coordinate ascent variational inference (CAVI - see, for example, Bishop, 2006) algorithm; see the Algorithm 2 in Appendix F, where we also provide additional details on the evaluation of the ELBO.

As noted at the end of Section 2.2, the posterior inference strategy for our proposed GLocal DP differs fundamentally from that employed in the vHDPMM of Dinari & Freifeld (2020). Specifically, their approach utilizes a split-merge MCMC sampler that jointly accounts for both global and local variables. In contrast, we adopt a scalable variational inference framework, which is more computationally efficient and better suited for large-scale applications.

## 5. Experiments

We conduct experiments to demonstrate the usefulness of the proposed GLocal DP on both synthetic data as well as a publicly available pan-cancer genomics data.

### 5.1. Simulations

Throughout the simulations, we considered three groups or populations. We assumed that there were three shared global variables across the populations and there were two, three, and four local variables for populations 1, 2, and 3, respectively. We varied the degree of separation in the local variables for the three populations. The detailed simulation strategy is presented in Appendix Section G.4.

We assessed the clustering accuracy by adjusted Rand index (ARI, Hubert & Arabie, 1985). We compared GLocal DP with HDP in terms of global-level clustering and with a Gaussian mixture model (GMM) applied to each group separately in terms of local-level clustering. HDP used global variables only whereas GLocal DP and GMM used both global and local variables. All simulations were replicated 50 times. We ran the HDP MCMC sampler for 50,000 iterations. The first half of the iterations were discarded as burn-in, and posterior samples were retained at every 25th iteration after burn-in. For the proposed VI, we considered the difference in ELBO in successive iterations, $\Delta(t-1, t) < 10^{-5}$ as a stopping rule to define the convergence of the algorithm. Although the discrepancy between the variational distribution and the target posterior is reduced at each iteration, there is no guarantee that the CAVI algorithm will converge to a global optimum; rather it will likely obtain a local solution depending on the initialization. Hence, we executed 20 distinct runs of the algorithm with different starting points, keeping the one with the highest

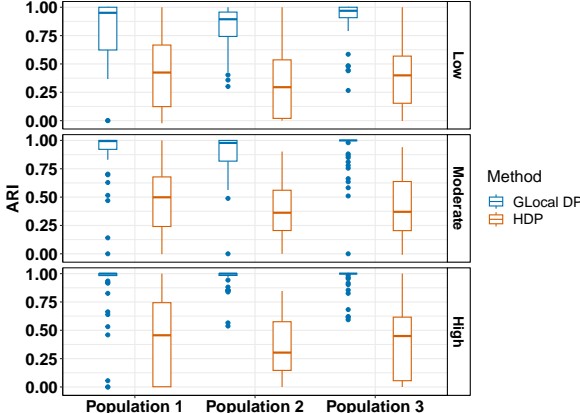

(a) Global-level clustering.

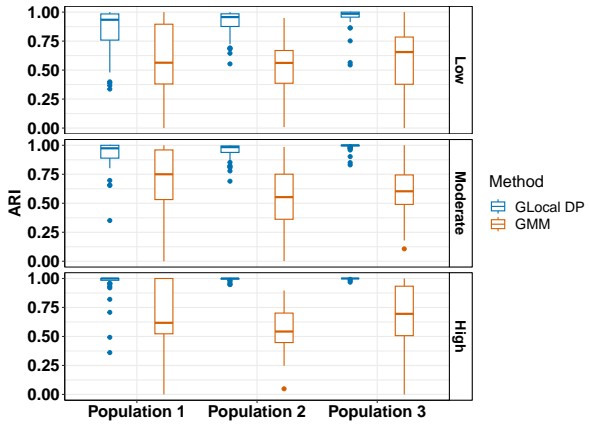

(b) Local-level clustering.

*Figure 1.* Comaprison of clustering accuracy of GLocal DP with (a) HDP at the global-level and (b) GMM at the local-level with varying separation in the local variables.

ELBO to draw the inference. Furthermore, we chose the truncation levels, $K = T = 30$.

For the HDP MCMC sampler, we estimated the clusters by minimizing the variation of information loss (Wade & Ghahramani, 2018). For the proposed VI algorithm, it returns the optimized variational parameters corresponding to the cluster assignment probabilities, i.e., $\hat{\rho}_{jtk} = q^*(k_{jt} = k)$ and $\hat{\xi}_{jit} = q^*(t_{ji} = t)$ where $q^*(\cdot)$ denotes the variational probability under the optimized variational parameters. Hence, in this case, the clusters were estimated as

$$\hat{k}_{jt} = \arg\max_{k=1,\ldots,K} \hat{\rho}_{jtk} \quad \text{and} \quad \hat{t}_{ji} = \arg\max_{t=1,\ldots,T} \hat{\xi}_{jit}$$

for $j = 1, \ldots, J, t = 1, \ldots, T$, and $i = 1, \ldots, n_j$.

Clearly, Figure 1 shows that the clustering performance of the proposed GLocal DP is better than HDP. Furthermore, the clustering performance of our method clearly improves

with the increasing separation in the local variables. Moreover, at the local level, the GLocal DP clustering accuracy is higher than the GMM. In the Appendix Section G.4 we present additional simulations comparing the GLocal DP with HDP and GMM with different dimensions of global variables and varying sample sizes. In all cases, GLocal DP outperforms both the competing models in terms of clustering accuracy.

Additionally, we have performed simulations comparing the clustering accuracy of the GLocal DP using MCMC vs VI. The results are presented in Appendix Section G.1. In summary, the clustering performance of the two algorithms is comparable. However, VI offers significant advantages in computational efficiency and resource utilization.

Given the shared modeling motivation between the vHDPMM and our GLocal DP, we also performed a series of simulation experiments to compare the clustering performance of the two models under various scenarios, with detailed simulation strategies and results provided in Appendix Section G.6. In summary, both models perform comparably well in scenarios with well-separated clusters, regardless of the data-generating mechanism. However, under less separated conditions, the GLocal DP outperforms the vHDPMM even when the latter is the true generative model, highlighting the advantages of the GLocal DP's joint modeling framework over the hierarchical modeling approach of the vHDPMM. These results underscore the greater flexibility of the GLocal DP in capturing complex global and local clustering patterns across diverse settings.

### 5.2. Real Data

In this section, we showcase the usefulness of the proposed GLocal DP by analyzing a pan-cancer genomics dataset. Integrated clustering analyses across cancers can objectively identify cancer subpopulations beyond the tumor site of origin, enhancing our understanding of both intra-tumor and inter-tumor heterogeneity and potentially repurposing existing cancer treatments across tumor types (Schein, 2021; Rodrigues et al., 2022). In databases like TCGA, genomic data are often accompanied by clinical data, providing largely orthogonal information regarding tumor heterogeneity, and some clinical data may be cancer-specific. Existing methods for clustering grouped data are limited to using a common set of variables, necessitating the exclusion of critical cancer-specific clinical data. In this application, we aim to identify pan-cancer subpopulations using both shared and cancer-specific data through GLocal DP.

We analyzed four gastrointestinal (GI) tract cancers: esophageal, stomach, colon, and rectal. Esophageal and stomach cancers, categorized as upper GI tract cancers, originate in the food pipe and stomach lining, respectively. Colon and rectal cancers, collectively termed colorectal can-

cer (Paschke et al., 2018), belong to the lower GI tract and share many features (Libutti et al., 2018a;b). Our study explores tumor heterogeneity within and across these cancers, using gene expression and clinical data from the TCGA database (Goldman et al., 2020). The dataset includes log-transformed gene expression for 60,483 genes across 173, 407, 512, and 177 patients with esophageal, stomach, colon, and rectal cancers, respectively. The selection of clinical variables to include in our analysis is explained in the following. First, smoking has been identified as a major risk factor for esophageal cancer (Fan et al., 2008). Second, the number of positive lymph nodes serve as an indicator for the degree of tumor spread in stomach cancer (Wu et al., 1996). Third, CEA is an important prognostic marker for monitoring tumor progression in colorectal cancer. However, CEA is not collected for esophageal and stomach cancers. Finally, recent studies have shown that several common cancers including colon cancer have been linked to obesity (Pati et al., 2023). According to Frezza et al., 2006, measuring BMI is crucial for assessing the obesity-related risk of developing colon cancer. In conclusion, such scientifically relevant aspects led us to consider the number of cigarettes smoked per day as a local variable for esophageal cancer, the number of positive lymph nodes for stomach cancer, the pre-operative and pre-treatment CEA as the local variable for both colon and rectal cancers, and BMI as an additional variable specific to colon cancer. After excluding patients with missing clinical data, the final sample sizes were 92, 363, 173, and 120 for esophageal, stomach, colon, and rectal cancers, respectively.

Following the common practice, we performed PCA on the combined gene expression data from the four cancers and retained the top ten principal components (PC) as the global variables. We considered the truncation levels, $K = T = 30$ and ran the VI algorithm 20 times with different initializations in parallel, choosing the one with the highest ELBO to draw inference. The maximum runtime over the 20 runs was less than 9 minutes on a MacBook Pro with M1 chip and 16GB RAM. For visualization, we reduced the original combined gene expression data to two dimensions using the uniform manifold approximation and projection (UMAP, McInnes et al., 2018). Figure 2 shows the UMAP embeddings colored by the estimated global- and local-level clusters. The global-level clusters in Figure 2(a) show that patients with colon and rectal cancers share significant similarities. However, some colon cancer patients exhibit slight differences in gene expression patterns (corresponds to cluster 26). Further investigation reveals that the colon and rectal cancer patients corresponding to cluster 12 have similar CEA levels (median values of 3.2 and 3.625 ng/mL) while the colon cancer patients belonging to cluster 26 have a higher median CEA of 4.1 ng/mL. Contrary to the lower GI tract cancers, the two upper GI cancers are not similar

to each other. And they are also not similar to lower GI cancers.

Figure 2(b) shows the subclusters identified at the local level, which emerge through the refinement of global-level clusters due to the presence of group-specific clinical variables. Figure 3 shows how the local-level clusters are influenced by the local variables. For instance, distinct clusters are observed among esophageal cancer patients based on smoking history, highlighting group-specific heterogeneity. The corresponding gene expression patterns for these subgroups of patients may be investigated based on the local-level clusters. To understand if the identified cancer subpopulations possibly inform cancer prognosis, we plotted the Kaplan-Meier survival curves for each of the identified cancer subpopulations in Figure 4. Among esophageal cancer patients, the survival curves reveal distinct trajectories associated with smoking history, underscoring the prognostic implications of this variable. Furthermore, for stomach cancer, the subcluster 2g (Figure 3) is characterized by patients with a very high number of positive lymph nodes. The corresponding survival curve in Figure 4 indicates that this subgroup exhibits poorer survival outcomes compared to other groups, particularly in comparison to those with fewer positive lymph nodes. Clustering based on gene expression data alone cannot discern the tumor heterogeneity from the prognostic perspective.

## 6. Conclusion

We have introduced the GLocal DP as a stochastic process for grouped random measures and as a prior for group clustering that accommodates varying variable sets. The GLocal DP was characterized using its stick-breaking representation and as a limit of a finite mixture model, with truncation error results providing practical guidelines for truncation level selection. We have developed a novel variational inference algorithm for scalable inference and have showcased our method through extensive simulations and an application to a pan-cancer dataset integrating shared gene expression and cancer-specific clinical data. For example, our approach identifies global clusters shared across cancers and finer cancer-specific sub-clusters using local variables, which existing methods cannot achieve.

There are a few possible future directions. First, our model could be extended to incorporate the group-clustering feature of the nested DP alongside the cluster-sharing feature of the HDP (Beraha et al., 2021; Balocchi et al., 2022; Lijoi et al., 2023), or to leverage shared-atom nested models (Denti et al., 2023; D'Angelo et al., 2023; D'Angelo & Denti, 2024). Second, in the presence of external predictors, our model may be extended to achieve covariate-assisted GLocal clustering in line with Ren et al., 2011; Wade et al., 2014; Rigon & Durante, 2021; Zhang et al., 2024.

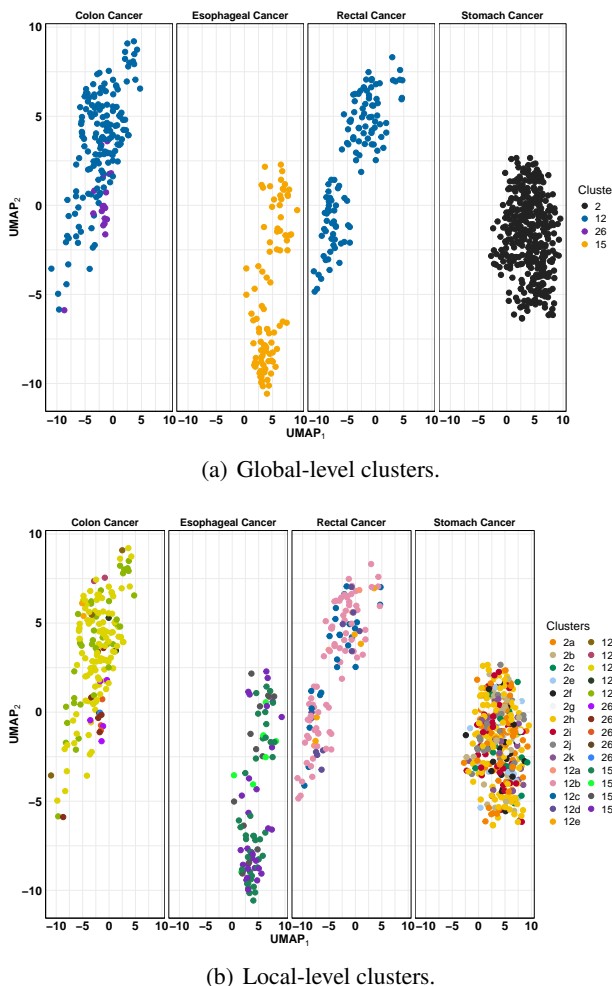

(a) Global-level clusters.

(b) Local-level clusters.

*Figure 2.* Global variables. (a) The colors indicate global-level clusters estimated from GLocal DP. (b) The colors indicate the estimated local-level clusters.

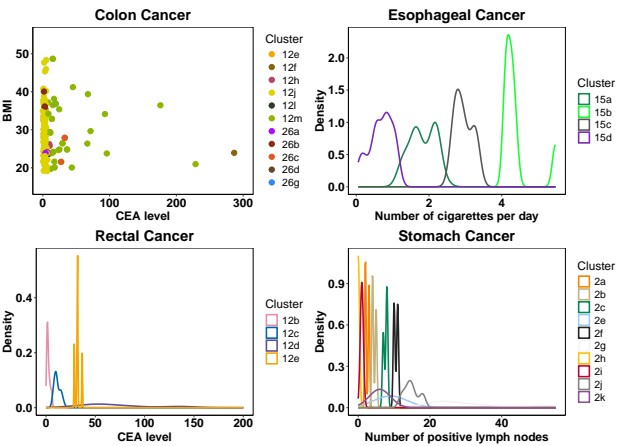

*Figure 3.* Kernel density/scatter plot of local variables, colored by the estimated local-level clusters.

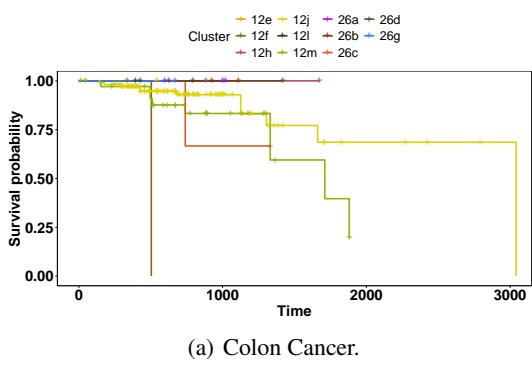

(a) Colon Cancer.

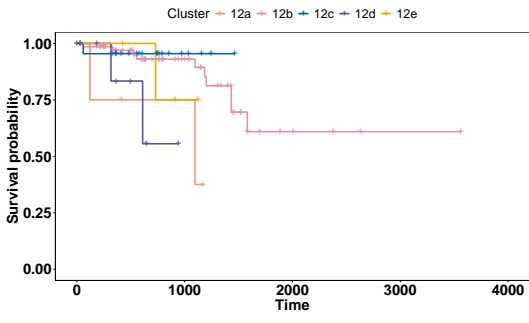

(b) Rectal Cancer.

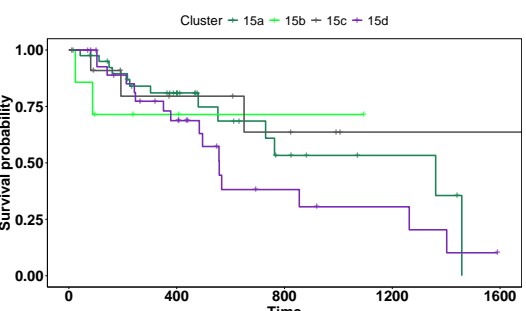

(c) Esophageal Cancer.

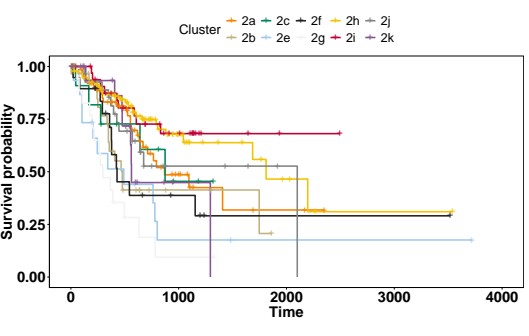

(d) Stomach Cancer.

*Figure 4.* Survival curves according to local-level clusters for the different cancers.

## Acknowledgments

The authors would like to thank the Program Chairs, the Area Chair, and the three anonymous reviewers, whose feedback led to a substantial improvement of the paper.

Ni's research is partially supported by NIH R01 GM148974 and NSF DMS-2112943. Pati's research is partially supported by NIH R01 DE031134, NIH R21 DE031879 and NSF DMS-2413715.

## Impact statement

This paper presents work whose goal is to advance the field of Machine Learning. There are many potential societal consequences of our work, none which we feel must be specifically highlighted here.

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

## A. Background

In this section, we present a brief overview of some preliminaries needed before introducing our model in Section 2 of the main manuscript. In particular, we provide a concise introduction to infinite mixture models for a single population, the DP mixture model, and for multiple exchangeable populations, the HDP mixture model.

### DIRICHLET PROCESS MIXTURE MODEL

For a single population, let $x_i$ denote the $i$th realization of a random variable $X$. Consider the following mixture model,

$$\theta_i \mid G \overset{iid}{\sim} G, \ \ x_i \mid \theta_i \ \overset{ind}{\sim} F(\theta_i), \tag{21}$$

where $F(\theta_i)$ denotes the distribution of $x_i$ parameterized by $\theta_i$. The parameters $\theta_i$'s are conditionally independent given the prior distribution $G$. In a DP mixture model, $G$ is assigned a DP prior, $G \sim \mathrm{DP}(\alpha_0, G_0)$ with concentration $\alpha_0$ and base probability measure $G_0$.

Sethuraman, 1994 presented the *stick-breaking representation* of the DP based on independent sequences of i.i.d. random variables $(\pi'_k)_{k=1}^\infty$ and $(\phi_k)_{k=1}^\infty$, which is given by,

$$\pi'_k \overset{iid}{\sim} Beta(1, \alpha_0), \qquad\qquad \phi_k \overset{iid}{\sim} G_0, \tag{22}$$

$$\pi_k = \pi'_k \prod_{l=1}^{k-1}(1 - \pi'_l), \qquad\qquad G = \sum_{k=1}^\infty \pi_k \delta_{\phi_k}, \tag{23}$$

where $\delta_\phi$ is a point mass at $\phi$ and $\phi_k$'s are called the *atoms* of $G$. The sequence of random weights $\boldsymbol{\pi} = (\pi_k)_{k=1}^\infty$ constructed from (22) and (23) satisfies $\sum_{k=1}^\infty \pi_k = 1$ with probability one. The random probability measure on the set of integers is denoted by $\boldsymbol{\pi} \sim \mathrm{GEM}(\alpha_0)$ for convenience where GEM stands for Griffiths, Engen and McCloskey (Pitman, 2002). It is clear from (21) and (23) that $\theta_i$ takes the value $\phi_k$ with probability $\pi_k$. Let $z_i$ be a categorical variable such that $z_i = k$ if $\theta_i = \phi_k$. An equivalent representation of a Dirichlet process mixture is given by,

$$\begin{aligned} \boldsymbol{\pi} &\sim \mathrm{GEM}(\alpha_0), & z_i \mid \boldsymbol{\pi} &\overset{iid}{\sim} \boldsymbol{\pi}, \\ \phi_k &\overset{iid}{\sim} G_0, & x_i \mid z_i, (\phi_k)_{k=1}^\infty &\overset{ind}{\sim} F(\phi_{z_i}). \end{aligned} \tag{24}$$

### HIERARCHICAL DIRICHLET PROCESS MIXTURE MODEL

Suppose observations are now organized into multiple groups. Let $x_{ji}$ denote the observation $i$ from group $j$. Let $F(\theta_{ji})$ denote the distribution of $x_{ji}$ parameterized by $\theta_{ji}$, and let $G_j$ denote a prior distribution for $\theta_{ji}$. The group-specific mixture model is given by,

$$\theta_{ji} \mid G_j \overset{ind}{\sim} G_j, \ \ x_{ji} \mid \theta_{ji} \ \overset{ind}{\sim} F(\theta_{ji}).$$

As with the DP mixture model, when the random measures $G_j$'s are assigned an HDP prior,

$$\begin{aligned} G_0 &\sim \mathrm{DP}(\gamma, H), \\ G_j \mid G_0 &\sim \mathrm{DP}(\alpha_0, G_0), \end{aligned} \tag{25}$$

the corresponding mixture model is referred to as the HDP mixture model. The global random probability measure $G_0$ is distributed as a DP with concentration parameter $\gamma$ and base probability measure $H$. The group-specific random measures $G_j$'s are conditionally independent given $G_0$ and hence are exchangeable. They are distributed as DP with the base measure $G_0$ and some concentration parameter $\alpha_0$. Because DP-distributed $G_0$ is almost surely discrete, the atoms of $G_j$'s are necessarily shared across groups. This leads to a positive probability of shared clusters across different groups.

## B. Graphical Representation of GLocal DP

In this section, we present the graphical model representation of the GLocal DP mixture model (Figure 5).

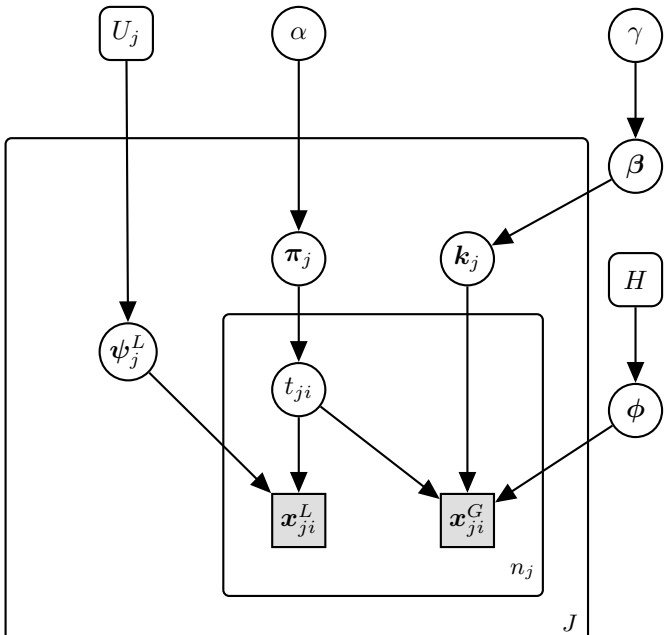

*Figure 5.* Graphical representation of the GLocal DP mixture model. Each node corresponds to a random variable, with shaded rectangles denoting observed variables. Rectangular plates indicate replication.

## C. Proof of the Infinite Limit of Finite Mixture Model

The finite mixture model represenatation of the GLocal DP is given by,

$$
\begin{aligned}
\boldsymbol{\beta} &\sim \text{Dir}(\gamma/K, \ldots, \gamma/K), & k_{jt} &\sim \boldsymbol{\beta}, \\
\boldsymbol{\pi}_j &\sim \text{Dir}(\alpha/T, \ldots, \alpha/T), & t_{ji} &\sim \boldsymbol{\pi}_j, \\
\phi_k &\sim H, & \psi_{jt}^L &\sim U_j, \\
\boldsymbol{x}_{ji} &\sim F_1(\boldsymbol{x}_{ji}^L \mid \psi_{jt_{ji}}^L) F_2(\boldsymbol{x}_{ji}^G \mid \phi_{k_{jt_{ji}}}),
\end{aligned}
\tag{C.26}
$$

where $\boldsymbol{\beta}$ is the global vector of mixing proportions, $\boldsymbol{\pi}_j$ is the group-specific vector of mixing proportions, $K$ is the number of global mixture components, and $T \geq K$ is the number of local mixture components. Further, as $K \to \infty$, the infinite limit of this model is the proposed GLocal DP mixture model.

*Proof.* Consider the random probability measure

$$
V^K = \sum_{k=1}^{K} \beta_k \delta_{\phi_k},
$$

where $\boldsymbol{\beta} = (\beta_k)_{k=1}^{L} \sim \text{Dir}(\gamma/K, \ldots, \gamma/K)$ and $\phi_k \overset{iid}{\sim} H, \ k = 1, \ldots, K$ independent of $\boldsymbol{\beta}$. Ishwaran & Zarepour, 2002 shows that for every measurable function $g$, integrable with respect to $H$, we have, as $K \to \infty$

$$
\int g(\theta) dV^K(\theta) \overset{\mathcal{D}}{\to} \int g(\theta) dV(\theta).
\tag{C.27}
$$

Further, for $T \geq K$, define

$$
G_j^{T,K} = \sum_{t=1}^{T} \pi_{jt} \delta_{\psi_{jt}},
$$

where $\boldsymbol{\pi}_j = (\pi_{jt})_{t=1}^T \sim \text{Dir}(\alpha/T, \ldots, \alpha/T)$ and $\psi_{jt} = (\psi_{jt}^L, \psi_{jt}^G) \overset{iid}{\sim} U_j \otimes V^K$ independent of $\boldsymbol{\pi}_j$. Let $B_j \times C$ be an arbitrary measurable subset of $\Theta_j \times \Omega$. Then,

$$G_j^{T,K}(B_j \times C) = \sum_{t=1}^T \pi_{jt} \mathbb{1}_{B_j}(\psi_{jt}^L) \mathbb{1}_C(\psi_{jt}^G)$$

$$= \sum_{t=1}^T \sum_{k=1}^K \pi_{jt} \mathbb{1}_{B_j}(\psi_{jt}^L) \mathbb{1}_C(\phi_k) \tag{C.28}$$

Here the indicator function $\mathbb{1}_A(x) = 1$ if $x \in A$ and is 0 otherwise. The second equality follows since for $T < \infty$ and any fixed $t$, $\psi_{jt}^G = \phi_k$, for some $k = 1, \ldots, K$. Since (C.28) holds for any arbitrary measurable $B_j \times C$, we have

$$G_j^{T,K} \sim \text{DP}(\alpha, U_j \otimes V^K). \tag{C.29}$$

It is clear from (C.27) and (C.29), that as $K \to \infty$, $T \to \infty$, and the marginal distribution that the finite mixture model induces on the observations approaches the proposed GLocal DP mixture model. $\qquad \square$

## D. Proof of Truncation Approximation Bounds

In this section we provide the proof of Proposition 3.1 and Proposition 3.2. Recall that the GLocal DP prior is defined as, $G_j | \alpha, V \sim \text{DP}(\alpha, U_j \otimes V)$, where $V | \gamma \sim \text{DP}(\gamma, H)$. Furthermore, define the truncated versions of $G_j$ as follows,

$$V^K = \sum_{k=1}^K \beta_k^K \delta_{\phi_k},$$

where $\phi_k \overset{i.i.d.}{\sim} H$, $k = 1, \ldots, K$,

$$\beta_k^K = \begin{cases} \beta_k & \text{if } k \leq K-1, \\ 1 - \sum_{k=1}^{K-1} \beta_k & \text{if } k = K, \end{cases}$$

$$G_j^{T,K} = \sum_{t=1}^T \pi_{jt}^{T,K} \delta_{\psi_{jt}},$$

where $\psi_{jt} \overset{ind}{\sim} U_j \otimes V^K$, $t = 1, \ldots, T$, and

$$\pi_{jt}^{T,K} = \begin{cases} \pi_{jt} & \text{if } t \leq T-1, \\ 1 - \sum_{t=1}^{T-1} \pi_{jt} & \text{if } t = T. \end{cases}$$

Here $T, K > 0$ define the truncation levels for the different random probability measures.

Consider $J$ groups, each of them containing $n_j$ observations, $j = 1, \ldots, J$. Denote by $(\boldsymbol{x}_j^L, \boldsymbol{x}_j^G) \equiv \boldsymbol{x}_j = (\boldsymbol{x}_{j1}, \ldots, \boldsymbol{x}_{jn_j})$ the collection of all observations from the $j$-th group arising from the mixture model $\boldsymbol{x}_{ji} | \boldsymbol{\theta}_{ji} \sim F(\cdot | \boldsymbol{\theta}_{ji})$ with $\boldsymbol{\theta}_{ji} | G_j \sim G_j$ where the $G_j$'s are generated according to the proposed GLocal DP. Let $f(\cdot | \boldsymbol{\theta}_{ji})$ be the density of $F(\cdot | \boldsymbol{\theta}_{ji})$ with respect to some dominating measure. We assume that $\boldsymbol{\theta}_{ji} \in (\Theta_j \times \Omega)$, where $(\Theta_j \times \Omega)$ is a Polish space equipped with its corresponding Borel $\sigma$–field $\mathcal{A}_j \otimes \mathcal{B}$. Finally, we denote by $\boldsymbol{x} = (\boldsymbol{x}_1, \ldots, \boldsymbol{x}_J)$ the vector containing the observations from all $J$ groups. Moreover, recall that

$$P^{\infty,\infty}(\boldsymbol{\theta}) = \int_\Omega \left[ \prod_{j=1}^J \int_{\Theta_j} \left\{ \prod_{i=1}^{n_j} P(\boldsymbol{\theta}_{ji} | G_j) \right\} P^\infty(dG_j | V) \right] P^\infty(dV) \qquad \text{and}$$

$$P^{T,K}(\boldsymbol{\theta}) = \int_\Omega \left[ \prod_{j=1}^J \int_{\Theta_j} \left\{ \prod_{i=1}^{n_j} P(\boldsymbol{\theta}_{ji} | G_j) \right\} P^T(dG_j | V) \right] P^K(dV)$$

denotes the prior distribution of the parameters $\boldsymbol{\theta}$ under the GLocal DP and its corresponding truncated version after integrating out the random distributions. Here $P(\boldsymbol{\theta}_{ji} \mid G_j)$ denotes the prior distribution of the parameters $\boldsymbol{\theta}_{ji}$ given the random measure $G_j$. Furthermore, let $m^{\infty,\infty}(\boldsymbol{x})$ and $m^{T,K}(\boldsymbol{x})$ denote the marginal distribution of the data $\boldsymbol{x}$ derived from these priors. Particularly,

$$m^{\infty,\infty}(\boldsymbol{x}) = \int_{\Xi^N} f(\boldsymbol{x}|\boldsymbol{\theta})P^{\infty,\infty}(\boldsymbol{\theta}) \, d\boldsymbol{\theta} \text{ and } m^{T,K}(\boldsymbol{x}) = \int_{\Xi^N} f(\boldsymbol{x}|\boldsymbol{\theta})P^{T,K}(\boldsymbol{\theta}) \, d\boldsymbol{\theta}.$$

Then we have the following result.

**Proposition 3.1.** *Let $P^{\infty,\infty}(\boldsymbol{\theta})$ and $P^{T,K}(\boldsymbol{\theta})$ denote the prior distribution of the parameters $\boldsymbol{\theta}$ under the GLocal DP prior and its corresponding truncated version with the random measures integrated out. Furthermore, let $m^{\infty,\infty}(\boldsymbol{x})$ and $m^{T,K}(\boldsymbol{x})$ denote the marginal distribution of the data $\boldsymbol{x}$, derived from these priors. Then,*

$$\int_{\mathcal{X}^N} \left| m^{T,K}(\boldsymbol{x}) - m^{\infty,\infty}(\boldsymbol{x}) \right| d\boldsymbol{x} \le \int_{\Xi^N} \left| P^{T,K}(\boldsymbol{\theta}) - P^{\infty,\infty}(\boldsymbol{\theta}) \right| d\boldsymbol{\theta} \le \epsilon^{T,K}(\alpha, \gamma), \tag{D.1}$$

*where*

$$\epsilon^{T,K}(\alpha, \gamma) = 4 \left[ 1 - \left\{ \left( 1 - \left( \frac{\alpha}{1+\alpha} \right)^{T-1} \right) \right\}^N \left\{ \left( 1 - \left( \frac{\gamma}{1+\gamma} \right)^{K-1} \right) \right\}^N \right], \tag{D.2}$$

$N = n_1 + \cdots + n_J$, $\Xi^N = \prod_{j=1}^{J}(\Theta_j \times \Omega)^{n_j}$, *and* $\mathcal{X}^N$ *denotes the sample space of observations* $\boldsymbol{x}$.

*Proof.* Note that,

$$
\begin{aligned}
\int_{\mathcal{X}^N} \left| m^{T,K}(\boldsymbol{x}) - m^{\infty,\infty}(\boldsymbol{x}) \right| d\boldsymbol{x} &= \int_{\mathcal{X}^N} \left| \int_{\Xi^N} f(\boldsymbol{x}|\boldsymbol{\theta})P^{T,K}(\boldsymbol{\theta})d\boldsymbol{\theta} - \int_{\Xi^N} f(\boldsymbol{x}|\boldsymbol{\theta})P^{\infty,\infty}(\boldsymbol{\theta})d\boldsymbol{\theta} \right| d\boldsymbol{x} \\
&= \int_{\mathcal{X}^N} \left| \int_{\Xi^N} f(\boldsymbol{x}|\boldsymbol{\theta}) \left\{ P^{T,K}(\boldsymbol{\theta}) - P^{\infty,\infty}(\boldsymbol{\theta}) \right\} d\boldsymbol{\theta} \right| d\boldsymbol{x} \\
&\le \int_{\mathcal{X}^N} \int_{\Xi^N} f(\boldsymbol{x}|\boldsymbol{\theta}) \left| P^{T,K}(\boldsymbol{\theta}) - P^{\infty,\infty}(\boldsymbol{\theta}) \right| d\boldsymbol{\theta} \, d\boldsymbol{x} \\
&= \int_{\Xi^N} \left\{ \int_{\mathcal{X}^N} f(\boldsymbol{x}|\boldsymbol{\theta}) \, d\boldsymbol{x} \right\} \left| P^{T,K}(\boldsymbol{\theta}) - P^{\infty,\infty}(\boldsymbol{\theta}) \right| d\boldsymbol{\theta} \\
&= \int_{\Xi^N} \left| P^{T,K}(\boldsymbol{\theta}) - P^{\infty,\infty}(\boldsymbol{\theta}) \right| d\boldsymbol{\theta} \\
&= 2 \sup_{A \in \Xi^N} \left| P^{T,K}(A) - P^{\infty,\infty}(A) \right| \\
&= 2 \, \mathrm{d_{TV}}(P^{T,K}, P^{\infty,\infty}), \tag{D.3}
\end{aligned}
$$

where $\mathrm{d_{TV}}(P^{T,K}, P^{\infty,\infty})$ denotes the distance in total variation between the marginalized GLocal DP prior on $\boldsymbol{\theta}$, $P^{\infty,\infty}$ and its corresponding truncated version, $P^{T,K}$. Let $\boldsymbol{\theta}_{ji} = \left( \boldsymbol{\theta}^L_{jt_{ji}}, \boldsymbol{\theta}^G_{jk_{jt_{ji}}} \right)$. The sampled values of $(\boldsymbol{\theta}_j)_{j=1}^{J}$ under $G^{T,K}$ and $G$ are identical when $t_{ji} < T$ and $k_{jt_{ji}} < K$, for $i = 1, \ldots, n_j; j = 1, \ldots, J$. Therefore, we have,

$$
\begin{aligned}
\mathrm{d_{TV}}(P^{T,K}, P^{\infty,\infty}) &\le 2 \left( 1 - G^{T,K}\{t_{ji} < T \text{ and } k_{jt_{ji}} < K \text{ for all } j \text{ and } i\} \right) \\
&= 2 \left( 1 - G^{T,K}\{k_{jt_{ji}} < K \text{ for all } j \text{ and } i \mid t_{ji} < T \text{ for all } j \text{ and } i\} \, G^{T,K}\{t_{ji} < T \text{ for all } j \text{ and } i\} \right) \\
&= 2 \left( 1 - \prod_{j=1}^{J} \left[ \mathbb{E} \left( \sum_{t=1}^{T-1} \pi_{jt} \right)^{n_j} \mathbb{E} \left( \sum_{k=1}^{K-1} \beta_k \right)^{n_j} \right] \right) \\
&\le 2 \left( 1 - \prod_{j=1}^{J} \left[ \mathbb{E} \left( 1 - \pi_{jT} \right)^{n_j} \mathbb{E} \left( 1 - \beta_K \right)^{n_j} \right] \right)
\end{aligned}
$$

$$= 2 \left[ 1 - \left\{ \left( 1 - \left( \frac{\alpha}{1+\alpha} \right)^{T-1} \right) \right\}^N \left\{ \left( 1 - \left( \frac{\gamma}{1+\gamma} \right)^{K-1} \right) \right\}^N \right],$$ (D.4)

where the second last inequality follows by the Jensen's inequality and the last equality follows by straightforward calculations based on the stick–breaking representation of the weights. The proof follows immediately from (D.3) and (D.4). □

Next, we prove the Proposition 3.2, which states,

**Proposition 3.2.** *The posterior distribution of the parameters $\boldsymbol{\theta}$ under the GLocal DP prior and its truncated version,*

$$\pi^{\infty,\infty}(\boldsymbol{\theta}|\boldsymbol{x}) = \frac{f(\boldsymbol{x}|\boldsymbol{\theta})P^{\infty,\infty}(\boldsymbol{\theta})}{m^{\infty,\infty}(\boldsymbol{x})},$$

$$\pi^{T,K}(\boldsymbol{\theta}|\boldsymbol{x}) = \frac{f(\boldsymbol{x}|\boldsymbol{\theta})P^{T,K}(\boldsymbol{\theta})}{m^{T,K}(\boldsymbol{x})}$$ (D.5)

*satisfies*

$$\int_{\mathcal{X}^N} \int_{\Xi^N} \left| \pi^{T,K}(\boldsymbol{\theta}|\boldsymbol{x}) - \pi^{\infty,\infty}(\boldsymbol{\theta}|\boldsymbol{x}) \right| m^{\infty,\infty}(\boldsymbol{x}) \, d\boldsymbol{\theta} \, d\boldsymbol{x} = \mathcal{O}\left( \epsilon^{T,K}(\alpha, \gamma) \right).$$ (D.6)

*Proof.* Write

$$P^{\infty,\infty}(\boldsymbol{\theta}) = P^{T,K}(\boldsymbol{\theta}) + \left( P^{\infty,\infty}(\boldsymbol{\theta}) - P^{T,K}(\boldsymbol{\theta}) \right).$$

Therefore we have,

$$
\begin{aligned}
\left| \pi^{T,K}(\boldsymbol{\theta}|\boldsymbol{x}) - \pi^{\infty,\infty}(\boldsymbol{\theta}|\boldsymbol{x}) \right| &= \left| \frac{f(\boldsymbol{x}|\boldsymbol{\theta})P^{T,K}(\boldsymbol{\theta})}{m^{T,K}(\boldsymbol{x})} - \frac{f(\boldsymbol{x}|\boldsymbol{\theta})P^{T,K}(\boldsymbol{\theta})}{m^{\infty,\infty}(\boldsymbol{x})} + \left( P^{\infty,\infty}(\boldsymbol{\theta}) - P^{T,K}(\boldsymbol{\theta}) \right) \frac{f(\boldsymbol{x}|\boldsymbol{\theta})}{m^{\infty,\infty}(\boldsymbol{x})} \right| \\
&\leq \left| \frac{f(\boldsymbol{x}|\boldsymbol{\theta})P^{T,K}(\boldsymbol{\theta})}{m^{T,K}(\boldsymbol{x})} \left( 1 - \frac{m^{T,K}(\boldsymbol{x})}{m^{\infty,\infty}(\boldsymbol{x})} \right) \right| + \left| \frac{f(\boldsymbol{x}|\boldsymbol{\theta})}{m^{\infty,\infty}(\boldsymbol{x})} \left( P^{\infty,\infty}(\boldsymbol{\theta}) - P^{T,K}(\boldsymbol{\theta}) \right) \right| \\
&= \frac{f(\boldsymbol{x}|\boldsymbol{\theta})P^{T,K}(\boldsymbol{\theta})}{m^{T,K}(\boldsymbol{x})} \left| 1 - \frac{m^{T,K}(\boldsymbol{x})}{m^{\infty,\infty}(\boldsymbol{x})} \right| + \frac{f(\boldsymbol{x}|\boldsymbol{\theta})}{m^{\infty,\infty}(\boldsymbol{x})} \left| P^{\infty,\infty}(\boldsymbol{\theta}) - P^{T,K}(\boldsymbol{\theta}) \right|,
\end{aligned}
$$

and consequently,

$$
\begin{aligned}
&\int_{\mathcal{X}^N} \int_{\Xi^N} \left| \pi^{T,K}(\boldsymbol{\theta}|\boldsymbol{x}) - \pi^{\infty,\infty}(\boldsymbol{\theta}|\boldsymbol{x}) \right| m^{\infty,\infty}(\boldsymbol{x}) \, d\boldsymbol{\theta} \, d\boldsymbol{x} \\
&\leq \int_{\mathcal{X}^N} \int_{\Xi^N} \frac{f(\boldsymbol{x}|\boldsymbol{\theta})P^{T,K}(\boldsymbol{\theta})}{m^{T,K}(\boldsymbol{x})} \left| m^{T,K}(\boldsymbol{x}) - m^{\infty,\infty}(\boldsymbol{x}) \right| d\boldsymbol{\theta} \, d\boldsymbol{x} + \int_{\mathcal{X}^N} \int_{\Xi^N} f(\boldsymbol{x}|\boldsymbol{\theta}) \left| P^{\infty,\infty}(\boldsymbol{\theta}) - P^{T,K}(\boldsymbol{\theta}) \right| d\boldsymbol{\theta} \, d\boldsymbol{x} \\
&= \int_{\mathcal{X}^N} \left\{ \int_{\Xi^N} \frac{f(\boldsymbol{x}|\boldsymbol{\theta})P^{T,K}(\boldsymbol{\theta})}{m^{T,K}(\boldsymbol{x})} \, d\boldsymbol{\theta} \right\} \left| m^{T,K}(\boldsymbol{x}) - m^{\infty,\infty}(\boldsymbol{x}) \right| d\boldsymbol{x} + \int_{\Xi^N} \left\{ \int_{\mathcal{X}^N} f(\boldsymbol{x}|\boldsymbol{\theta}) \, d\boldsymbol{x} \right\} \left| P^{\infty,\infty}(\boldsymbol{\theta}) - P^{T,K}(\boldsymbol{\theta}) \right| d\boldsymbol{\theta} \\
&= \int_{\mathcal{X}^N} \left| m^{T,K}(\boldsymbol{x}) - m^{\infty,\infty}(\boldsymbol{x}) \right| d\boldsymbol{x} + \int_{\Xi^N} \left| P^{\infty,\infty}(\boldsymbol{\theta}) - P^{T,K}(\boldsymbol{\theta}) \right| d\boldsymbol{\theta} \\
&\leq \epsilon^{T,K}(\alpha, \gamma) + \epsilon^{T,K}(\alpha, \gamma) \qquad \text{from Proposition 3.1} \\
&= \mathcal{O}\left( \epsilon^{T,K}(\alpha, \gamma) \right),
\end{aligned}
$$

which concludes the proof. □

# E. Blocked Gibbs Sampler

Based on the finite mixture model approximation of the GLocal DP in (10), with large enough truncation levels $K$ and $T$, it is straightforward to develop an MCMC-based Metropolis-within-blocked-Gibbs sampler for the GLocal DP. In this section,

we outline the blocked Gibbs sampling algorithm as an alternative to our VI algorithm as described in the next section. Recall the finite mixture model representation of the GLocal DP,

$$
\begin{aligned}
\boldsymbol{\beta} &\sim \mathrm{Dir}(\gamma/K, \ldots, \gamma/K), & k_{jt} &\sim \boldsymbol{\beta}, \\
\boldsymbol{\pi}_j &\sim \mathrm{Dir}(\alpha/T, \ldots, \alpha/T), & t_{ji} &\sim \boldsymbol{\pi}_j, \\
\phi_k &\sim H, & \psi_{jt}^L &\sim U_j, \\
\boldsymbol{x}_{ji} &\sim F_1(\boldsymbol{x}_{ji}^L \mid \psi_{jt_{ji}}^L) F_2(\boldsymbol{x}_{ji}^G \mid \phi_{k_{jt_{ji}}}).
\end{aligned}
\tag{E.1}
$$

Let $\boldsymbol{x} = (\boldsymbol{x}_j)_{j=1}^J$ to denote the observations from all $J$ groups. Similarly, $\boldsymbol{t} = (\boldsymbol{t}_j)_{j=1}^J$ and $\boldsymbol{k} = (\boldsymbol{k}_j)_{j=1}^J$ denotes the collection of all local-level and global-level latent indicators respectively. The collection of all local atoms are denoted by $\boldsymbol{\psi} = (\boldsymbol{\psi}_j)_{j=1}^J$, with $\boldsymbol{\psi}_j = (\psi_{jt}^L)_{t=1}^T$ denoting the local atoms of group $j$. Similarly, the collection of global atoms are given by $\boldsymbol{\phi} = (\phi_k)_{k=1}^K$. Furthermore, let $f_1(. \mid \psi_{jt}^L)$ and $f_2(. \mid \phi_k)$ be the density functions (with respect to some dominating measure) corresponding to the distributions $F_1(. \mid \psi_{jt}^L)$ and $F_2(. \mid \phi_k)$, respectively. The augmented likelihood is then given by,

$$
p(\boldsymbol{x}, \boldsymbol{t}, \boldsymbol{k} \mid \boldsymbol{\psi}, \boldsymbol{\phi}, (\boldsymbol{\pi}_j)_{j=1}^J, \boldsymbol{\beta}) = \left\{ \prod_{j=1}^J \prod_{i=1}^{n_j} f_1(\boldsymbol{x}_{ji}^L \mid \psi_{jt_{ji}}^L) f_2(\boldsymbol{x}_{ji}^G \mid \phi_{k_{jt_{ji}}}) \right\} \times \prod_{j=1}^J \prod_{i=1}^{n_j} \prod_{t=1}^T \pi_{jt}^{\mathbb{1}(t_{ji}=t)} \prod_{j=1}^J \prod_{t=1}^T \prod_{k=1}^K \beta_k^{\mathbb{1}(k_{jt}=k)}.
$$

The model parameters are $\{\boldsymbol{\psi}, \boldsymbol{\phi}, (\boldsymbol{\pi}_j)_{j=1}^J, \boldsymbol{\beta}, \alpha, \gamma\}$, with the joint prior distribution given by,

$$
p(\boldsymbol{\psi}, \boldsymbol{\phi}, (\boldsymbol{\pi}_j)_{j=1}^J, \boldsymbol{\beta}, \alpha, \gamma) \left\{ \prod_{j=1}^J \prod_{t=1}^T p(\psi_{jt}^L) \right\} \left\{ \prod_{k=1}^K p(\phi_k) \right\} \left\{ \prod_{j=1}^J p(\boldsymbol{\pi}_j \mid \alpha) \right\} \times p(\boldsymbol{\beta} \mid \gamma) \, p(\alpha) \, p(\gamma).
$$

We remark that $K$ and $T$ are the *maximal* numbers of global and local clusters specified by the users. They should be large enough so that the numbers of sampled clusters are always strictly smaller than them over the course of the MCMC. Picking the maximal number of clusters in our algorithm is much more straightforward than picking the *exact* number of clusters in many existing clustering algorithms. Alternatively, the user should choose a value for $K$ and $T$ that leads to a precise approximation of the truncated GLocal DP mixture model.

For simplicity of presentation, for the global variables, we assume a mixture of multivariate Gaussian kernels $p(\cdot \mid \boldsymbol{\phi}) = \mathcal{N}_p(\cdot \mid \boldsymbol{\mu}, \boldsymbol{\Lambda}^{-1})$, with $\boldsymbol{\mu}$ a $p$-dimensional mean vector and $\boldsymbol{\Lambda}$ a $p \times p$ precision matrix. Also, we assume a conjugate normal-Wishart prior distribution on the model parameters,

$$
(\boldsymbol{\mu}, \boldsymbol{\Lambda}) \sim \mathrm{NW}(\boldsymbol{\mu}_0, \tau_0, \nu_0, \boldsymbol{\Psi}_0).
$$

Similarly, for the local variables in population $j$, we assume, $p_j(\cdot \mid \boldsymbol{\psi}_j^L) = \mathcal{N}_{p_j}(\cdot \mid \boldsymbol{\mu}_j, \boldsymbol{\Lambda}_j^{-1})$, with $\boldsymbol{\mu}_j$ a $p_j$-dimensional mean vector and $\boldsymbol{\Lambda}_j$ a $p_j \times p_j$ precision matrix. Furthermore, we assume a conjugate normal-Wishart prior distribution on the model parameters,

$$
(\boldsymbol{\mu}_j, \boldsymbol{\Lambda}_j) \sim \mathrm{NW}(\boldsymbol{\mu}_{j0}, \tau_{j0}, \nu_{j0}, \boldsymbol{\Psi}_{j0}).
$$

The blocked Gibbs sampler for the GLocal DP is given in Algorithm 1. We remark that for other choices of kernels and prior distributions, the blocked Gibbs sampling algorithm is straightforward to derive from Algorithm 1. Particularly, the likelihood functions in steps 5 and 6 of Algorithm 1 need to be replaced by those of the chosen kernels. Similarly, the conditional posteriors of the global atoms and local atoms in steps 3 and 4 would change depending on the choice of the kernel and prior distributions.

---

**Algorithm 1** Blocked Gibbs Sampler for the GLocal DP

---

**for** $i = 1$ **to** $B$ **do**

    Sample the parameters from their conditional posterior distributions,

    1.    For each $j$, sample $\boldsymbol{\pi}_j$ from a Dirichlet distribution $\mathrm{Dir}(m_{j1}+\alpha/T, \ldots, m_{jT}+\alpha/T)$, where $m_{jt} = \sum_{i=1}^{n_j} \mathbb{1}(t_{ji} = t)$.

2. Sample $\boldsymbol{\beta}$ from a Dirichlet distribution $\mathrm{Dir}(d_1 + \gamma/K, \ldots, d_K + \gamma/K)$, where $d_k = \sum_{j=1}^{J} \sum_{t=1}^{T} \mathbb{1}(k_{jt} = k)$.

3. For each $k$, sample $\phi_k$ from a conjugate $\mathrm{NW}(\hat{\boldsymbol{\mu}}_k, \hat{\tau}_k, \hat{\nu}_k, \hat{\boldsymbol{\Psi}}_k)$ distribution with parameters

$$\hat{\boldsymbol{\mu}}_k = \hat{\lambda}_k^{-1}(\tau_0\,\boldsymbol{\mu}_0 + N_k \bar{\boldsymbol{x}}_k^G), \quad \hat{\lambda}_k = \tau_0 + N_k, \quad \hat{\nu}_k = \nu_0 + N_k,$$

$$\hat{\boldsymbol{\Psi}}_k^{-1} = \boldsymbol{\Psi}_0^{-1} + \frac{\tau_0 N_k}{\tau_0 + N_k} \left( \bar{\boldsymbol{x}}_k^G - \boldsymbol{\mu}_0 \right) \left( \bar{\boldsymbol{x}}_k^G - \boldsymbol{\mu}_0 \right)^T + \boldsymbol{S}_k^G,$$

where

$$N_k = \sum_{j=1}^{J} \sum_{i=1}^{n_j} \sum_{t=1}^{T} \mathbb{1}(t_{ji} = t)\,\mathbb{1}(k_{jt} = k), \qquad \bar{\boldsymbol{x}}_k^G = N_k^{-1} \left( \sum_{j=1}^{J} \sum_{i=1}^{n_j} \sum_{t=1}^{T} \left\{ \mathbb{1}(t_{ji} = t)\,\mathbb{1}(k_{jt} = k)\,\boldsymbol{x}_{ji}^G \right\} \right),$$

$$\boldsymbol{S}_k^G = \sum_{j=1}^{J} \sum_{i=1}^{n_j} \sum_{t=1}^{T} \left( \mathbb{1}(t_{ji} = t)\,\mathbb{1}(k_{jt} = k)\,\left( \boldsymbol{x}_{ji}^G - \bar{\boldsymbol{x}}_k^G \right) \left( \boldsymbol{x}_{ji}^G - \bar{\boldsymbol{x}}_k^G \right)^T \right).$$

4. For each $j$ and $t$, sample $\psi_{jt}^L$ from a conjugate $\mathrm{NW}(\hat{\boldsymbol{\mu}}_{jt}, \hat{\tau}_{jt}, \hat{\nu}_{jt}, \hat{\boldsymbol{\Psi}}_{jt})$ distribution with parameters

$$\hat{\boldsymbol{\mu}}_{jt} = \hat{\lambda}_{jt}^{-1}(\tau_{j0}\,\boldsymbol{\mu}_{j0} + N_{jt} \bar{\boldsymbol{x}}_{jt}^L), \quad \hat{\lambda}_{jt} = \tau_{j0} + N_{jt}, \quad \hat{\nu}_{jt} = \nu_{j0} + N_{jt},$$

$$\hat{\boldsymbol{\Psi}}_{jt}^{-1} = \boldsymbol{\Psi}_{j0}^{-1} + \frac{\tau_{j0} N_{jt}}{\tau_{j0} + N_{jt}} \left( \bar{\boldsymbol{x}}_{jt}^L - \boldsymbol{\mu}_{j0} \right) \left( \bar{\boldsymbol{x}}_{jt}^L - \boldsymbol{\mu}_{j0} \right)^T + \boldsymbol{S}_{jt}^L,$$

where

$$N_{jt} = \sum_{i=1}^{n_j} \mathbb{1}(t_{ji} = t), \qquad \bar{\boldsymbol{x}}_{jt}^L = N_{jt}^{-1} \left( \sum_{i=1}^{n_j} \mathbb{1}(t_{ji} = t)\,\boldsymbol{x}_{ji}^L \right),$$

$$\boldsymbol{S}_{jt}^L = \sum_{i=1}^{n_j} \left( \mathbb{1}(t_{ji} = t)\,\left( \boldsymbol{x}_{ji}^L - \bar{\boldsymbol{x}}_{jt}^L \right) \left( \boldsymbol{x}_{ji}^L - \bar{\boldsymbol{x}}_{jt}^L \right)^T \right).$$

5. For each $j$, $i$, and $t$, sample the local-level latent indicators from the following full conditional distribution:

$$\mathbb{P}(t_{ji} = t \mid -) \propto \pi_{jt} \mathcal{N}_{p_j}(\boldsymbol{x}_{ji}^L \mid \boldsymbol{\mu}_{jt}, \boldsymbol{\Lambda}_{jt}^{-1}) \mathcal{N}_p(\boldsymbol{x}_{ji}^G \mid \boldsymbol{\mu}_{k_{jt}}, \boldsymbol{\Lambda}_{k_{jt}}^{-1}).$$

6. For each $j$, $t$, and $k$, sample the global-level latent indicators from the following full conditional distribution:

$$\mathbb{P}(k_{jt} = k \mid -) \propto \beta_k \prod_{\substack{i=1 \\ \ni t_{ji} = t}}^{n_j} \mathcal{N}_p(\boldsymbol{x}_{ji}^G \mid \boldsymbol{\mu}_k, \boldsymbol{\Lambda}_k^{-1}).$$

7. Sample $\alpha$ following full conditional distribution:

$$p(\alpha \mid -) \propto \frac{\{\Gamma(\alpha)\}^J}{\{\Gamma(\alpha/T)\}^{JT}} \prod_{j=1}^{J} \prod_{t=1}^{T} \pi_{jt}^{\alpha/T - 1}\, p(\alpha),$$

using a Metropolis-Hastings (MH) step with a gamma proposal distribution.

8. Sample $\gamma$ following full conditional distribution:

$$p(\gamma \mid -) \propto \frac{\Gamma(\gamma)}{\{\Gamma(\gamma/K)\}^K} \prod_{k=1}^{K} \beta_k^{\gamma/K - 1}\, p(\gamma),$$

using a MH step with a gamma proposal distribution.

**end for**

# F. Variational Inference

In this section, we outline the variational inference algorithm for the GLocal DP for the specific case of multivariate Gaussian likelihood. However, this approach can be adapted in a straightforward manner whenever the data distribution is a member of the exponential family, as discussed in Blei & Jordan, 2006. As before, for the global variables, we assume a mixture of multivariate Gaussian kernels $p(\cdot \mid \phi) = \mathcal{N}_p(\cdot \mid \boldsymbol{\mu}, \boldsymbol{\Lambda}^{-1})$, with $\boldsymbol{\mu}$ a $p$-dimensional mean vector and $\boldsymbol{\Lambda}$ a $p \times p$ precision matrix. Also, we assume a conjugate normal-Wishart prior distribution on the model parameters,

$$(\boldsymbol{\mu}, \boldsymbol{\Lambda}) \sim \mathrm{NW}(\boldsymbol{\mu}_0, \tau_0, \nu_0, \boldsymbol{\Psi}_0).$$

Similarly, for the local variables in population $j$, we assume, $p_j(\cdot \mid \boldsymbol{\psi}_j^L) = \mathcal{N}_{p_j}(\cdot \mid \boldsymbol{\mu}_j, \boldsymbol{\Lambda}_j^{-1})$, with $\boldsymbol{\mu}_j$ a $p_j$-dimensional mean vector and $\boldsymbol{\Lambda}_j$ a $p_j \times p_j$ precision matrix. Furthermore, we assume a conjugate normal-Wishart prior distribution on the model parameters,

$$(\boldsymbol{\mu}_j, \boldsymbol{\Lambda}_j) \sim \mathrm{NW}(\boldsymbol{\mu}_{j0}, \tau_{j0}, \nu_{j0}, \boldsymbol{\Psi}_{j0}).$$

First, we provide an alternative representation of the proposed GLocal DP relying on the finite truncation of the stick-breaking representation. Specifically, consider the following finite mixture model,

$$
\begin{aligned}
&v_k \overset{i.i.d.}{\sim} \mathrm{Beta}(1, \gamma),\ k = 1, \ldots, K-1,\ v_K = 1, && \beta_1 = v_1,\ \beta_k = v_k \prod_{l=1}^{k-1}(1 - v_l),\ \text{for } k = 2, \ldots, K \\
&\text{Let } \boldsymbol{\beta} = (\beta_1, \ldots, \beta_K) && k_{jt} \sim \boldsymbol{\beta}, \\
&u_{jt} \overset{ind}{\sim} \mathrm{Beta}(1, \alpha),\ t = 1, \ldots, T-1,\ u_{jT} = 1, && \pi_{j1} = u_{j1},\ \pi_{jt} = u_{jt} \prod_{l=1}^{t-1}(1 - u_{jt}),\ \text{for } t = 2, \ldots, T \\
&\text{Let } \boldsymbol{\pi}_j = (\pi_{j1}, \ldots, \pi_{jT}) && t_{ji} \sim \boldsymbol{\pi}_j, \\
&(\boldsymbol{\mu}_k, \boldsymbol{\Lambda}_k) \sim \mathrm{NW}(\boldsymbol{\mu}_0, \tau_0, \nu_0, \boldsymbol{\Psi}_0), && (\boldsymbol{\mu}_{jt}, \boldsymbol{\Lambda}_{jt}) \sim \mathrm{NW}(\boldsymbol{\mu}_{j0}, \tau_{j0}, \nu_{j0}, \boldsymbol{\Psi}_{j0}), \\
&\boldsymbol{x}_{ji} \sim \mathcal{N}_{p_j}(\boldsymbol{x}_{ji}^L \mid \boldsymbol{\mu}_{jt_{ji}}, \boldsymbol{\Lambda}_{jt_{ji}}^{-1})\, \mathcal{N}_p(\boldsymbol{x}_{ji}^G \mid \boldsymbol{\mu}_{k_{jt_{ji}}}, \boldsymbol{\Lambda}_{k_{jt_{ji}}}^{-1}).
\end{aligned}
$$
$$\tag{F.1}$$

Furthermore, we assume non-informative priors on the concentration parameters, i.e., $\gamma \sim \mathrm{Gamma}(a_\gamma, b_\gamma)$ and $\alpha \sim \mathrm{Gamma}(a_\alpha, b_\alpha)$. Recall that we consider a fully factorized mean field family of distributions as the variational family. Furthermore, we use a truncated variational family to deal with the nonparametric mixture, where the truncation levels are denoted by $T$ and $K$. Specifically, we assume,

$$
\begin{aligned}
q(\boldsymbol{t}, \boldsymbol{k}, \boldsymbol{u}, \boldsymbol{v}, \phi, \psi, \alpha, \gamma; \boldsymbol{\lambda}) = {}& \prod_{j=1}^{J} \prod_{i=1}^{n_j} q(t_{ji}; \{\xi_{jit}\}_{t=1}^{T}) \prod_{j=1}^{J} \prod_{t=1}^{T} q(k_{jt}; \{\rho_{jtl}\}_{l=1}^{K}) \prod_{j=1}^{J} \prod_{t=1}^{T-1} q(u_{jt}; \bar{a}_{jt}, \bar{b}_{jt}) \times \\
& \times \prod_{k=1}^{K-1} q(v_k; \bar{a}_k, \bar{b}_k) \prod_{k=1}^{K} q(\boldsymbol{\mu}_k, \boldsymbol{\Lambda}_k; \boldsymbol{m}_k, \lambda_k, c_k, \boldsymbol{D}_k) \times \\
& \times \prod_{j=1}^{J} \prod_{t=1}^{T} q(\boldsymbol{\mu}_{jt}, \boldsymbol{\Lambda}_{jt}; \boldsymbol{m}_{jt}, \lambda_{jt}, c_{jt}, \boldsymbol{D}_{jt})\, q(\alpha; s_1, s_2)\, q(\gamma; r_1, r_2),
\end{aligned}
$$

where $q(t_{ji}; \{\xi_{jit}\}_{t=1}^{T})$ and $q(k_{jt}; \{\rho_{jtl}\}_{l=1}^{K})$ are multinomial distributions; $q(v_k; \bar{a}_k, \bar{b}_k)$ and $q(u_{jt}; \bar{a}_{jt}, \bar{b}_{jt})$ are beta distributions, and they are such that $q(v_K = 1) = 1$ and $q(v_g = 0) = 1$ for $g > K$ and similarly, $q(u_{jT} = 1) = 1$ and $q(u_{jh} = 0) = 1$ for $h > T$; $q(\alpha; s_1, s_2)$ and $q(\gamma; r_1, r_2)$ are gamma distributions; $q(\boldsymbol{\mu}_k, \boldsymbol{\Lambda}_k; \boldsymbol{m}_k, \lambda_k, c_k, \boldsymbol{D}_k)$ and $q(\boldsymbol{\mu}_{jt}, \boldsymbol{\Lambda}_{jt}; \boldsymbol{m}_{jt}, \lambda_{jt}, c_{jt}, \boldsymbol{D}_{jt})$ are normal-Wishart distributions. Under this representation, the set of latent variables is $\boldsymbol{\Theta} = (\boldsymbol{t}, \boldsymbol{k}, \boldsymbol{u}, \boldsymbol{v}, \{\boldsymbol{\mu}_k, \boldsymbol{\Lambda}_k\}_{k=1}^{K}, \{\{\boldsymbol{\mu}_{jt}, \boldsymbol{\Lambda}_{jt}\}_{t=1}^{T}\}_{j=1}^{J}, \alpha, \gamma)$ and the set of variational parameters is $\boldsymbol{\lambda} = (\boldsymbol{\rho}, \boldsymbol{\xi}, \bar{\boldsymbol{a}}, \bar{\boldsymbol{b}}, \{\bar{\boldsymbol{a}}_j\}_{j=1}^{J}, \{\bar{\boldsymbol{b}}_j\}_{j=1}^{J}, s_1, s_2, r_1, r_2, \boldsymbol{m}, \boldsymbol{t}, \boldsymbol{c}, \boldsymbol{D}, \{\boldsymbol{m}_j\}_{j=1}^{J}, \{\boldsymbol{t}_j\}_{j=1}^{J}, \{\boldsymbol{c}_j\}_{j=1}^{J}, \{\boldsymbol{D}_j\}_{j=1}^{J})$. Optimization is then carried out by looking for the combination of variational parameters $\boldsymbol{\lambda}^\star$ that maximizes the evidence lower bound (ELBO). To this end, based on (F.1), we derive a coordinate-ascent variational inference algorithm (CAVI), given in Algorithm 2.

---

**Algorithm 2** CAVI updates for the GLocal DP

---

**Input:** $t \leftarrow 0$. Randomly initialize $\boldsymbol{\lambda}^{(0)}$. Define the threshold $\epsilon$ and randomly set $\Delta > \epsilon$.

**while** $\Delta(t-1, t) > \varepsilon$ **do**

Set $t = t + 1$; Let $\boldsymbol{\lambda}^{(t-1)} = \boldsymbol{\lambda}^{(t)}$;

Update the variational parameters according to the following CAVI steps:

---

1. For $j = 1, \ldots, J$ and $t = 1, \ldots, T$, $q^\star(k_{jt})$ is a $K$-dimensional multinomial, with $q^\star(k_{jt} = k) = \rho_{jtk}$ for $k = 1, \ldots, T$,

$$\log \rho_{jtk} = g(\bar{a}_k, \bar{b}_k) + \sum_{l=1}^{k-1} g(\bar{b}_l, \bar{a}_l) + \frac{1}{2} \sum_{i=1}^{n_j} \xi_{jit} \left( \ell_k^{(1)} + \ell_{jik}^{(2)} \right),$$

where $g(x, y) = \psi(x) - \psi(x + y)$, with $\psi$ denoting the digamma function, $\ell_k^{(1)} = \sum_{i=1}^p \psi((c_k - i + 1)/2) + p \log 2 + \log |\boldsymbol{D}_k|$, and $\ell_{jik}^{(2)} = -p/\lambda_k - c_k(\boldsymbol{x}_{ji}^G - \boldsymbol{m}_k)^T \boldsymbol{D}_k (\boldsymbol{x}_{ji}^G - \boldsymbol{m}_k)$.

2. For $j = 1, \ldots, J$ and $i = 1, \ldots, n_j$, $q^\star(t_{ji})$ is a $T$-dimensional multinomial, with $q^\star(t_{ji} = t) = \xi_{jit}$ for $t = 1, \ldots, T$,

$$\log \xi_{jit} = g(\bar{a}_{jt}, \bar{b}_{jt}) + \sum_{l=1}^{t-1} g(\bar{b}_{jl}, \bar{a}_{jl}) + \frac{1}{2} \ell_{jt}^{(1)} + \frac{1}{2} \ell_{jit}^{(3)} + \frac{1}{2} \sum_{k=1}^K \rho_{jtk} \left( \ell_k^{(1)} + \ell_{jik}^{(2)} \right),$$

where $\ell_{jt}^{(1)} = \sum_{i=1}^{p_j} \psi((c_{jt} - i + 1)/2) + p_j \log 2 + \log |\boldsymbol{D}_{jt}|$ and $\ell_{jit}^{(3)} = -p_j/\lambda_{jt} - c_{jt}(\boldsymbol{x}_{ji}^L - \boldsymbol{m}_{jt})^T \boldsymbol{D}_{jt}(\boldsymbol{x}_{ji}^L - \boldsymbol{m}_{jt})$.

3. For $j = 1, \ldots, J$ and $t = 1, \ldots, T - 1$, $q^\star(u_{jt})$ is a $Beta(\bar{a}_{jt}, \bar{b}_{jt})$ distribution with

$$\bar{a}_{jt} = 1 + \sum_{i=1}^{n_j} \xi_{jit}, \quad \bar{b}_{jt} = s_1/s_2 + \sum_{l=t+1}^T \sum_{i=1}^{n_j} \xi_{jil}.$$

4. For $k = 1, \ldots, K - 1$, $q^\star(v_k)$ is a $Beta(\bar{a}_k, \bar{b}_k)$ distribution with

$$\bar{a}_k = 1 + \sum_{j=1}^J \sum_{t=1}^T \rho_{jtk}, \quad \bar{b}_k = r_1/r_2 + \sum_{l=k+1}^K \sum_{j=1}^J \sum_{t=1}^T \rho_{jtl}.$$

5. For $k = 1, \ldots, K$, $q^\star(\boldsymbol{\mu}_k, \boldsymbol{\Lambda}_k)$ is a $\text{NW}(\boldsymbol{m}_k, \lambda_k, c_k, \boldsymbol{D}_k)$ distribution with parameters

$$\boldsymbol{m}_k = \lambda_k^{-1}(\tau_0 \boldsymbol{\mu}_0 + N_k \bar{\boldsymbol{x}}_k^G), \quad \lambda_k = \tau_0 + N_k, \quad c_k = \nu_0 + N_k,$$
$$\boldsymbol{D}_k^{-1} = \boldsymbol{\Psi}_0^{-1} + \frac{\tau_0 N_k}{\tau_0 + N_k} \left( \bar{\boldsymbol{x}}_k^G - \boldsymbol{\mu}_0 \right) \left( \bar{\boldsymbol{x}}_k^G - \boldsymbol{\mu}_0 \right)^T + \boldsymbol{S}_k^G,$$

where

$$N_k = \sum_{j=1}^J \sum_{i=1}^{n_j} \sum_{t=1}^T \xi_{jit} \, \rho_{jtk}, \qquad \bar{\boldsymbol{x}}_k^G = N_k^{-1} \left( \sum_{j=1}^J \sum_{i=1}^{n_j} \sum_{t=1}^T \xi_{jit} \, \rho_{jtk} \, \boldsymbol{x}_{ji}^G \right),$$
$$\boldsymbol{S}_k^G = \sum_{j=1}^J \sum_{i=1}^{n_j} \sum_{t=1}^T \xi_{jit} \, \rho_{jtk} \left( \boldsymbol{x}_{ji}^G - \bar{\boldsymbol{x}}_k^G \right) \left( \boldsymbol{x}_{ji}^G - \bar{\boldsymbol{x}}_k^G \right)^T.$$

6. For $j = 1, \ldots, J$ and $t = 1, \ldots, T$, $q^\star(\boldsymbol{\mu}_{jt}, \boldsymbol{\Lambda}_{jt})$ is a $\text{NW}(\boldsymbol{m}_{jt}, \lambda_{jt}, c_{jt}, \boldsymbol{D}_{jt})$ distribution with parameters

$$\boldsymbol{m}_{jt} = \lambda_{jt}^{-1}(\tau_{j0} \boldsymbol{\mu}_{j0} + N_{jt} \bar{\boldsymbol{x}}_{jt}^L), \quad \lambda_{jt} = \tau_{j0} + N_{jt}, \quad c_{jt} = \nu_{j0} + N_{jt},$$
$$\boldsymbol{D}_{jt}^{-1} = \boldsymbol{\Psi}_{j0}^{-1} + \frac{\tau_{j0} N_{jt}}{\tau_{j0} + N_{jt}} \left( \bar{\boldsymbol{x}}_{jt}^L - \boldsymbol{\mu}_{j0} \right) \left( \bar{\boldsymbol{x}}_{jt}^L - \boldsymbol{\mu}_{j0} \right)^T + \boldsymbol{S}_{jt}^L,$$

where

$$N_{jt} = \sum_{i=1}^{n_j} \xi_{jit}, \qquad \bar{\boldsymbol{x}}_{jt}^L = N_{jt}^{-1} \left( \sum_{i=1}^{n_j} \xi_{jit} \, \boldsymbol{x}_{ji}^L \right),$$
$$\boldsymbol{S}_{jt}^L = \sum_{i=1}^{n_j} \xi_{jit} \left( \boldsymbol{x}_{ji}^L - \bar{\boldsymbol{x}}_{jt}^L \right) \left( \boldsymbol{x}_{ji}^L - \bar{\boldsymbol{x}}_{jt}^L \right)^T.$$

7. $q^\star(\alpha)$ is a $\text{Gamma}(s_1, s_2)$ distribution with parameters

$$s_1 = a_\alpha + J(T - 1), \quad s_2 = b_\alpha - \sum_{t=1}^{T-1} \sum_{j=1}^J g(\bar{b}_{jt}, \bar{a}_{jt}).$$

8. $q^\star(\gamma)$ is a $\mathrm{Gamma}(r_1, r_2)$ distribution with parameters

$$r_1 = a_\gamma + K - 1, \quad r_2 = b_\gamma - \sum_{k=1}^{K-1} g(\bar{b}_k, \bar{a}_k).$$

Store the updated parameters in $\boldsymbol{\lambda}$ and let $\boldsymbol{\lambda}^{(t)} = \boldsymbol{\lambda}$;
Compute $\Delta(t-1, t) = \mathrm{ELBO}(\boldsymbol{\lambda}^{(t)}) - \mathrm{ELBO}(\boldsymbol{\lambda}^{(t-1)})$.
**end while**
**Return** $\boldsymbol{\lambda}^\star$, containing the optimized variational parameters.

## F.1. Computation of the ELBO in the variational inference approach

Here we outline the ELBO evaluation for the GLocal DP. Recall that we use the notation $g(x, y) = \psi(x) - \psi(x + y)$. The minimization of the Kullback-Leibler divergence between the posterior and the variational distributions is equivalent to the maximization of the ELBO, expressed as

$$ELBO(q) = \mathbb{E}_q \left[ \log p(\boldsymbol{x}, \boldsymbol{\Theta}) \right] - \mathbb{E}_q \left[ \log q_{\boldsymbol{\lambda}}(\boldsymbol{\Theta}) \right].$$

The first term, $\mathbb{E}_q \left[ \log p(\boldsymbol{x}, \boldsymbol{\Theta}) \right]$, can be decomposed into the following components:

1. $\mathbb{E}[\log p(\boldsymbol{x}^L \mid \{\{\boldsymbol{\mu}_{jt}, \boldsymbol{\Lambda}_{jt}\}_{t=1}^T\}_{j=1}^J] = \frac{1}{2} \sum_{j=1}^J \sum_{i=1}^{n_j} \sum_{t=1}^T \xi_{jit} \left\{ \ell_{jt}^{(1)} + \ell_{jit}^{(3)} - p_j \log 2\pi \right\}.$

2. $\mathbb{E}[\log p(\boldsymbol{x}^G \mid \{\boldsymbol{\mu}_k, \boldsymbol{\Lambda}_k\}_{k=1}^K)] = \frac{1}{2} \sum_{j=1}^J \sum_{i=1}^{n_j} \sum_{t=1}^T \sum_{k=1}^K \xi_{jit} \rho_{jtk} \left\{ \ell_k^{(1)} + \ell_{jik}^{(2)} - p \log 2\pi \right\}.$

3. $\mathbb{E} \left[ \log p(\boldsymbol{t} \mid \boldsymbol{u}) \right] = \sum_{j=1}^J \sum_{i=1}^{n_j} \sum_{t=1}^T \xi_{jit} \left\{ g(\bar{a}_{jt}, \bar{b}_{jt}) + \sum_{l<t} g(\bar{b}_{jl}, \bar{a}_{jl}) \right\}.$

4. $\mathbb{E} \left[ \log p(\boldsymbol{k} \mid \boldsymbol{v}) \right] = \sum_{j=1}^J \sum_{t=1}^T \sum_{k=1}^K \rho_{jtk} \left\{ g(\bar{a}_k, \bar{b}_k) + \sum_{l<k} g(\bar{b}_l, \bar{a}_l) \right\}.$

5. $\mathbb{E} \left[ \log p(\prod_{k=1}^K \{\boldsymbol{\mu}_k, \boldsymbol{\Lambda}_k\}) \right] = K \log \boldsymbol{\mathcal{B}}(\boldsymbol{\Psi}_0, \nu_0) + 0.5(\nu_0 - p - 1) \sum_{k=1}^K \ell_k^{(1)} - 0.5 \sum_{k=1}^K c_k \boldsymbol{\mathcal{T}}(\boldsymbol{\Psi}_0^{-1} \boldsymbol{D}_k) + 0.5 \sum_{k=1}^K \left[ p \log(\tau_0/2\pi) + \ell_k^{(1)} - p\tau_0/\lambda_k - \tau_0 c_k (\boldsymbol{m}_k - \boldsymbol{\mu}_0)^T \boldsymbol{D}_k (\boldsymbol{m}_k - \boldsymbol{\mu}_0) \right]$, where $\boldsymbol{\mathcal{T}}(\cdot)$ is the trace operator and $\boldsymbol{\mathcal{B}}(\boldsymbol{\Psi}_0, \nu_0)$ is the inverse of the normalizing constant of a Wishart distribution (see, for more details, Appendix B of Bishop, 2006).

6. $\mathbb{E} \left[ \log p(\prod_{j=1}^J \prod_{t=1}^T \{\boldsymbol{\mu}_{jt}, \boldsymbol{\Lambda}_{jt}\}) \right] = T \sum_{j=1}^J \log \boldsymbol{\mathcal{B}}(\boldsymbol{\Psi}_{j0}, \nu_{j0}) + 0.5 \sum_{j=1}^J \sum_{t=1}^T (\nu_{j0} - p_j - 1) \ell_{jt}^{(1)} - 0.5 \sum_{j=1}^J \sum_{t=1}^T c_{jt} \boldsymbol{\mathcal{T}}(\boldsymbol{\Psi}_{j0}^{-1} \boldsymbol{D}_{jt}) + 0.5 \sum_{j=1}^J \sum_{t=1}^T \left[ p_j \log(\tau_{j0}/2\pi) + \ell_{jt}^{(1)} - p_j\tau_{j0}/\lambda_{jt} - \tau_{j0} c_{jt}(\boldsymbol{m}_{jt} - \boldsymbol{\mu}_{j0})^T \boldsymbol{D}_{jt}(\boldsymbol{m}_{jt} - \boldsymbol{\mu}_{j0}) \right].$

7. $\mathbb{E} \left[ \log p(\boldsymbol{u}) \right] = J(T-1) \left( \psi(s_1) - \log(s_2) \right) + (s_1/s_2 - 1) \sum_{j=1}^J \sum_{t=1}^{T-1} g(\bar{b}_{jt}, \bar{a}_{jt}).$

8. $\mathbb{E} \left[ \log p(\boldsymbol{v}) \right] = (K-1) \left( \psi(r_1) - \log(r_2) \right) + (r_1/r_2 - 1) \sum_{k=1}^{K-1} g(\bar{b}_k, \bar{a}_k).$

9. $\mathbb{E} \left[ \log p(\alpha) \right] = \log(\mathcal{C}_\alpha(a_\alpha, b_\alpha)) + (a_\alpha - 1)(\psi(s_1) - \log(s_2)) - b_\alpha s_1/s_2$, where $\mathcal{C}_\alpha(\cdot)$ is the normalizing constant of a Gamma distribution.

10. $\mathbb{E} \left[ \log p(\gamma) \right] = \log(\mathcal{C}_\gamma(a_\gamma, b_\gamma)) + (a_\gamma - 1)(\psi(r_1) - \log(r_2)) - b_\gamma r_1/r_2.$

The second term is decomposed into the following six components:

1. $\mathbb{E} \left[ \log q(\boldsymbol{t}) \right] = \sum_{j=1}^J \sum_{i=1}^{n_j} \sum_{t=1}^T \xi_{jit} \log(\xi_{jit}).$

2. $\mathbb{E} \left[ \log q(\boldsymbol{k}) \right] = \sum_{j=1}^J \sum_{t=1}^T \sum_{k=1}^K \rho_{jtk} \log(\rho_{jtk}).$

3. $\mathbb{E} \left[ \log q(\boldsymbol{u}) \right] = \sum_{j=1}^J \sum_{t=1}^{T-1} \{\log(\mathcal{C}_{\boldsymbol{u}}(\bar{a}_{jt}, \bar{b}_{jt})) + (\bar{a}_{jt} - 1)g(\bar{a}_{jt}, \bar{b}_{jt}) + (\bar{b}_{jt} - 1)g(\bar{b}_{jt}, \bar{a}_{jt})\}$, where $\mathcal{C}_{\boldsymbol{u}}(\cdot)$ is the normalizing constant of a Beta distribution.

4. $\mathbb{E}\left[\log q(\boldsymbol{v})\right] = \sum_{k=1}^{K-1}\{\log(\mathcal{C}_{\boldsymbol{v}}(\bar{a}_k, \bar{b}_k)) + (\bar{a}_k - 1)g(\bar{a}_k, \bar{b}_k) + (\bar{b}_k - 1)g(\bar{b}_k, \bar{a}_k)\}$.

5. $\mathbb{E}\left[\log q(\prod_{k=1}^{K}\{\boldsymbol{\mu}_k, \boldsymbol{\Lambda}_k\})\right] = \sum_{k=1}^{K}\left[0.5\ell_k^{(1)} + 0.5p(\log(\lambda_k/2\pi) - 1) - \boldsymbol{\mathcal{H}}\left(q(\boldsymbol{\Lambda}_k)\right)\right]$, where $\boldsymbol{\mathcal{H}}\left(q(\boldsymbol{\Lambda}_k)\right)$ is the entropy of a Wishart distribution.

6. $\mathbb{E}\left[\log q(\prod_{j=1}^{J}\prod_{t=1}^{T}\{\boldsymbol{\mu}_{jt}, \boldsymbol{\Lambda}_{jt}\})\right] = \sum_{j=1}^{J}\sum_{t=1}^{T}\left[0.5\ell_{jt}^{(1)} + 0.5p_j(\log(\lambda_{jt}/2\pi) - 1) - \boldsymbol{\mathcal{H}}\left(q(\boldsymbol{\Lambda}_{jt})\right)\right]$.

7. $\mathbb{E}\left[\log(q(\alpha))\right] = \log(\mathcal{C}_\alpha(s_1, s_2)) + (s_1 - 1)(\psi(s_1) - \log(s_2)) - s_1$, where $\mathcal{C}_\alpha(\cdot)$ is the normalizing constant of a Gamma distribution.

8. $\mathbb{E}\left[\log(q(\gamma))\right] = \log(\mathcal{C}_\gamma(r_1, r_2)) + (r_1 - 1)(\psi(r_1) - \log(r_2)) - r_1$.

## G. Simulations

### G.1. Comparison of MCMC and VI for the GLocal DP

First, we conduct simulations comparing the clustering accuracy of the GLocal DP using the MCMC-based blocked Gibbs sampler and the VI-based algorithm. Throughout the simulations, we assume that there are three groups or populations. We consider a simulation setting in which the shared variables across the populations are five dimensional. Furthermore, we assume that there are two, three, and four local variables for populations 1, 2, and 3, respectively. We generated the data from,

$$\boldsymbol{x}_{ji} \sim \left\{f_1(\boldsymbol{x}_{ji}^L \mid \psi_{jt}^L)f_2(\boldsymbol{x}_{ji}^G \mid \phi_k)\right\}, \tag{G.1}$$

where,

$$f_1(\boldsymbol{x}_{ji}^L \mid \psi_{jt}^L) = \sum_{t=1}^{L_{\ell_j}}\pi_{jt}\mathcal{N}_{p_j}(\boldsymbol{x}_{ji}^L \mid \boldsymbol{\mu}_{jt}, \boldsymbol{\Lambda}_{jt}^{-1}), \tag{G.2}$$

$$f_2(\boldsymbol{x}_{ji}^G \mid \phi_k) = \sum_{k=1}^{L_g}\beta_k\mathcal{N}_5(\boldsymbol{x}_{ji}^G \mid \boldsymbol{\mu}_k, \boldsymbol{\Lambda}_k^{-1}). \tag{G.3}$$

Here $p_1 = 2, p_2 = 3, p_3 = 4, L_{\ell_1} = 3, L_{\ell_2} = 5, L_{\ell_3} = 4$, and $L_g = 6$. The true parameters and the true mixture weights corresponding to the local variables are drawn from,

$$(\boldsymbol{\mu}_{jt}, \boldsymbol{\Lambda}_{jt}) \sim \text{NW}(\boldsymbol{0}, 0.1, 5\,p_j, \mathbb{I}_{p_j})$$
$$\alpha \sim \text{Gamma}(25, 1), \ \boldsymbol{\pi}_j \sim \text{Dir}(\alpha/L_{\ell_j}, \dots, \alpha/L_{\ell_j}),$$

for $j = 1, 2, 3$, and $t = 1, \dots, L_{\ell_j}$. The true local indicator $t_{ji}$ is drawn from a multinomial distribution with class probabilities $\boldsymbol{\pi}_j$, for $j = 1, 2, 3$. Similarly, the true parameters and mixture weights corresponding to the global variables are drawn from,

$$(\boldsymbol{\mu}_k, \boldsymbol{\Lambda}_k) \sim \text{NW}(\boldsymbol{0}, 0.1, 5\,p, \mathbb{I}_p)$$
$$\gamma \sim Gamma(25, 1), \ \boldsymbol{\beta} \sim \text{Dir}(\gamma/L_g, \dots, \gamma/L_g),$$

for $k = 1, \dots, L_g,$, where $p = 5$. The true latent indicator $k_{jt}$ is drawn from a multinomial distribution with the class probabilities $\boldsymbol{\beta}$, for $t = 1, \dots, L_g$. We considered several sample sizes for the three populations. In particular, we considered for all $j = 1, 2, 3\ n_j = 100, 200$, and 500.

We ran our MCMC sampler for 50,000 iterations. The first half of the iterations were discarded as burn-in, and posterior samples were retained at every 25th iteration after burn-in. As for the VI approach, we considered $\Delta(t - 1, t) < 10^{-5}$ as a stopping rule to define the convergence of the ELBO. Although the discrepancy between the variational distribution and the target posterior is reduced at each iteration, there is no guarantee that the CAVI algorithm will converge to a global optimum, rather it will likely obtain a local solution depending on the initialization. Hence, we executed 20 distinct runs of the algorithm with different starting points, keeping the one with the highest ELBO to draw the inference.

Depending on the computational strategy, the posterior point estimates of the clusters were obtained using different procedures. For the MCMC output, we estimated the clusters by minimizing the variation of information loss (Wade

& Ghahramani, 2018). When dealing with the VI approach, the algorithm returns the optimized variational parameters corresponding to the cluster assignment probabilities, i.e., $\hat{\rho}_{jtk} = q^*(k_{jt} = k)$ and $\hat{\xi}_{jit} = q^*(t_{ji} = t)$. Hence, in this case, the clusters were estimated as

$$\hat{k}_{jt} = \arg\max_{k=1,\ldots,K} \hat{\rho}_{jtk} \quad \text{and} \quad \hat{t}_{ji} = \arg\max_{t=1,\ldots,T} \hat{\xi}_{jit}$$

for $j = 1, \ldots, J$, $t = 1, \ldots, T$, and $i = 1, \ldots, n_j$. We assessed the accuracy of the estimated clusters using the adjusted Rand index (ARI, Hubert & Arabie, 1985) between the posterior point estimate and the true cluster. Figure 6(a) shows the distribution of the ARIs, over 50 simulated datasets obtained by MCMC based Gibbs sampler and VI for several choices of truncation levels, while keeping the sample size fixed at $n_j = 200$ for all $j = 1, 2, 3$. Similarly, Figure 6(b) shows the distribution of ARIs for varying sample sizes in each group ($N$ denotes the total sample size), and truncation levels fixed at 30. Clearly, the clustering performance from the two methods are comparable.

Next, we consider the the need for a variational inference approach over the MCMC based method. We compare the computational cost of the two algorithms. In particular, we compare the memory usage and the computing time by both algorithms. For the MCMC methods, the need to store the entire Markov chains (at most, excluding the burn-in and post thinning) can raise issues with memory allocation. On the contrary, VI methods only require storing the optimized parameters. Figure 7 shows the computational time and memory used for varying truncation levels and sample sizes, as before. For the MCMC sampler, computation time denotes the total run time for 50,000 iterations. Contrarily, for the VI based approach, we report the maximum individual run time obtained over the 20 runs for each dataset as the computational cost of the CAVI. Clearly, Figure 7 shows significant gains in computational time and resource utilization by the VI approach in comparison to the MCMC based approach.

### G.2. Comparison of GLocal DP with DP Mixture Model and GMM

In the main manuscript, we presented the comparison of clustering accuracy of the GLocal DP with GMM at the local-level, when the global variables were three-dimensional. In this subsection, we further compare the clustering accuracy with a DP mixture model (DPMM, Escobar & West, 1995) applied separately to each group. Specifically, for the simulation study presented in the main manuscript—where three global variables are shared across the three populations, and each population has two, three, and four local variables, respectively—we fit a separate DPMM for each group, incorporating both global and local variables. As in the main manuscript, we varied the degree of separation in the local variables while keeping the sample size fixed at $n_j = 200$ for all $j = 1, 2, 3$. The DPMM inference was conducted using an MCMC sampler run for 50,000 iterations, with the first half discarded as burn-in. After burn-in, posterior samples were retained at every 25th iteration. For cluster estimation under the DPMM, we followed the approach of minimizing the variation of information loss (Wade & Ghahramani, 2018). As in our original analysis, we also evaluated a GMM separately on each group and applied our proposed GLocal DP, which jointly models both global and local variables. The evaluation was conducted over 50 independent replications, and clustering accuracy was assessed using the ARI. Figure 8 demonstrates that across different levels of separation in the local variables, GLocal DP achieves clustering accuracy that is either comparable to or superior to that of the DPMM in terms of local-level clustering within each group.

### G.3. Clustering Performance of GLocal DP Using Additional Metrics

In the main manuscript, we evaluated the clustering accuracy of the GLocal DP by comparing it to the HDP at the global-level and separate GMMs applied to each group at the local-level, using the ARI as the evaluation metric. In this subsection, we complement that analysis by incorporating two additional clustering accuracy measures: Normalized Mutual Information (NMI; Strehl & Ghosh, 2002) and Purity (Manning et al., 2008). Specifically, we compare the global-level clustering performance of GLocal DP to that of HDP applied solely to the global variables, and the local-level clustering performance of GLocal DP to that of separate DPMMs fitted independently within each group. Figure 9 illustrates the superior clustering performance of GLocal DP compared to HDP at the global-level, as measured by both NMI and Purity, across varying degrees of separation in the local variables. Similarly, Figure 10 compares the clustering performance of GLocal DP with that of separate DPMMs applied to each group at the local-level. The results demonstrate that, across different levels of separation in the local variables, GLocal DP achieves clustering accuracy that is either comparable to or superior to that of the DPMM, as assessed by both NMI and Purity.

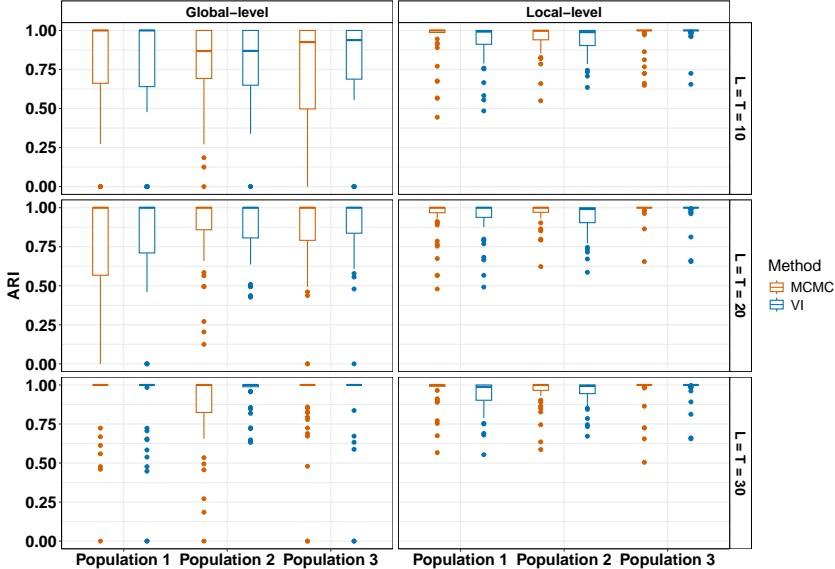

(a) Varying Truncation Levels. Sample size is fixed at 200 per population.

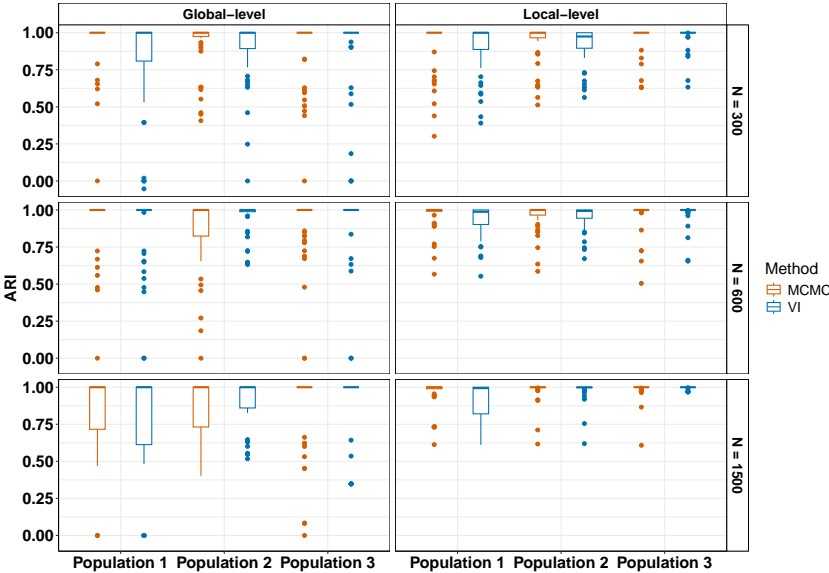

(b) Varying Sample Size. Truncation level is fixed at $K = H = 30$.

*Figure 6.* Accuracy of global-level and local-level clustering. Each panel shows the distribution of the ARI obtained across the 50 replications, for each configuration. The colors correspond to the algorithm.

### G.4. Comparison of GLocal DP with HDP and GMM

In the main manuscript, we presented the comparison of clustering accuracy of the GLocal DP with the HDP at the global-level and GMM at the local-level, when the global variables were three-dimensional. In this subsection, we further compare the clustering accuracy when the global variables were five-dimensional. As before, we assumed that there were two, three, and four local variables for populations 1, 2, and 3, respectively. We generated the data from (G.1) and as before, we assumed,

$$(\boldsymbol{\mu}_{jt}, \boldsymbol{\Lambda}_{jt}) \sim \mathrm{NW}(\mathbf{0}, \lambda_L^{-1}, 5\, p_j, \mathbb{I}_{p_j}) \tag{G.4}$$

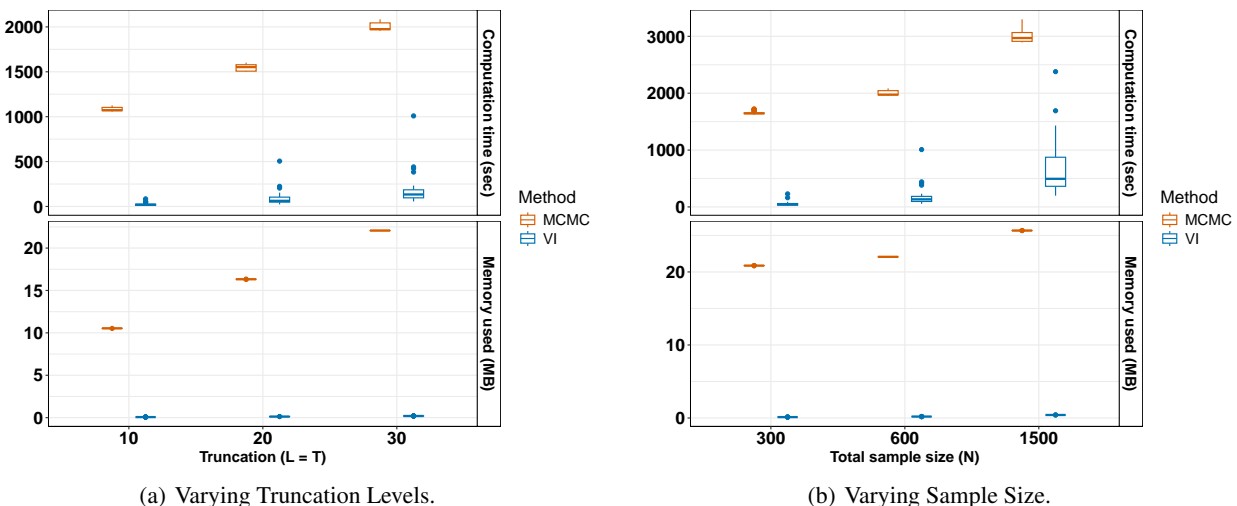

(a) Varying Truncation Levels.     (b) Varying Sample Size.

*Figure 7.* Top row: distributions of the computing time (in seconds) over the 50 replications for the two algorithms, for each configuration. Bottom row: distributions of the memory usage (in MB) over the 50 replications for the two algorithms, for each configuration. The colors correspond to the algorithm.

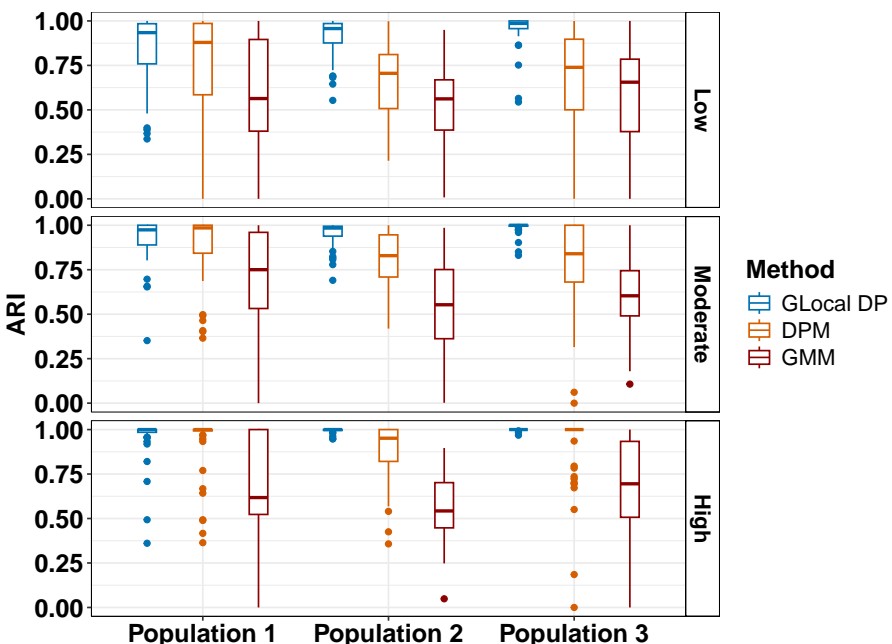

*Figure 8.* Comparison of clustering accuracy of GLocal DP with DPM and GMM at the local-level with varying separation in the local variables.

$$\alpha \sim \mathrm{Gamma}(25, 1), \ \boldsymbol{\pi}_j \sim \mathrm{Dir}(\alpha/L_{\ell_j}, \ldots, \alpha/L_{\ell_j}),$$

for $j = 1, 2, 3$, and $t = 1, \ldots, L_{\ell_j}$. Furthermore, $\lambda_L^{-1}$ is a precision parameter corresponding to the local variables. Similarly, the true parameters and mixture weights corresponding to the global variables are drawn from,

$$(\boldsymbol{\mu}_k, \boldsymbol{\Lambda}_k) \sim \mathrm{NW}(\boldsymbol{0}, \lambda_G^{-1}, 5\,p, \mathbb{I}_p) \tag{G.5}$$
$$\gamma \sim Gamma(25, 1), \ \boldsymbol{\beta} \sim \mathrm{Dir}(\gamma/L_g, \ldots, \gamma/L_g),$$

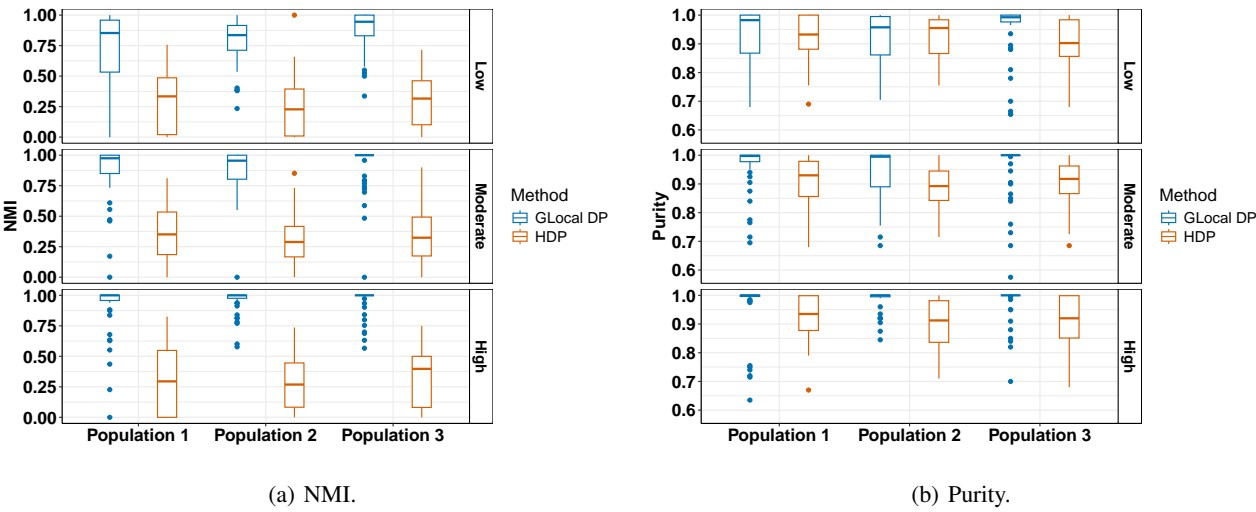

(a) NMI.                                          (b) Purity.

*Figure 9.* Comparison of clustering accuracy of GLocal DP using the metrics (a) NMI and (b) Purity with HDP at the global-level.

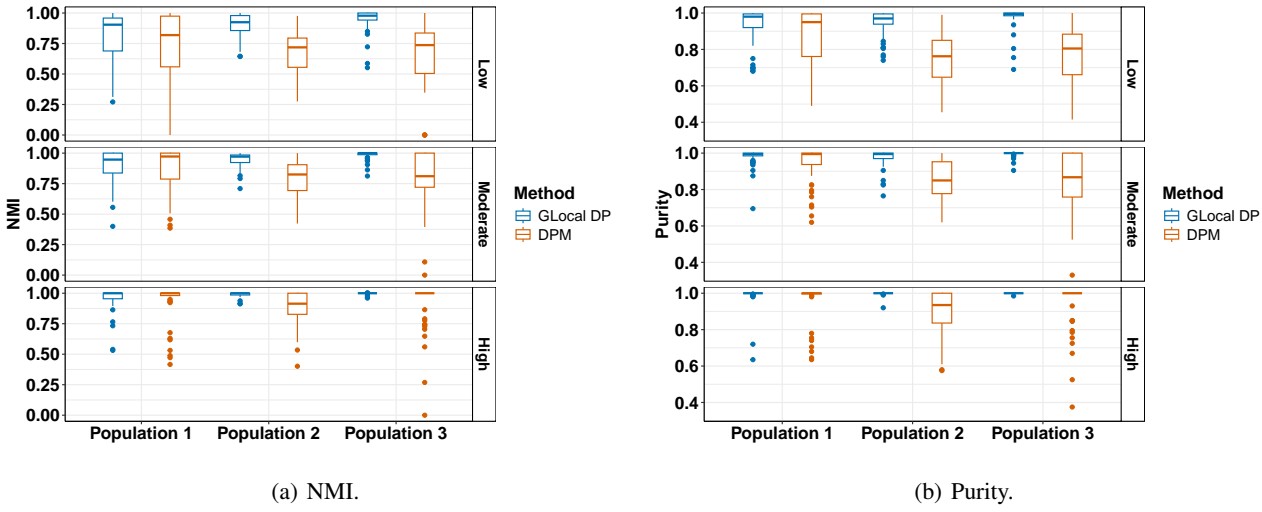

(a) NMI.                                          (b) Purity.

*Figure 10.* Comparison of clustering accuracy of GLocal DP using the metrics (a) NMI and (b) Purity with DPMM at the local-level.

for $k = 1, \ldots, L_g$,, where $p = 5$ and $\lambda_G^{-1}$ is a precision parameter corresponding to the global variables. For simulations, we set $\lambda_G = 1$ in (G.5), making the separations between the global variables to be low. Furthermore, we varied the degree of separation in the local variables for the three populations by varying the local-level precision parameter $\lambda_L = 0.1, 0.05, 0.01$ in (G.4). We considered for all $j = 1, 2, 3$ the sample size, $n_j = 200$. As before, HDP was applied to the global variables only whereas GLocal DP was applied to both global and local variables. Furthermore, we looked at the accuracy of local-level estimated clusters of GLocal DP and the estimated clusters obtained from a separate GMM for each group and all simulations were replicated 50 times. For the HDP, we ran the sampler for 50,000 iterations. The first half of the iterations were discarded as burn-in, and posterior samples were retained at every 25th iteration after burn-in. For the VI approach of our GLocal DP, we considered $\Delta(t-1, t) < 10^{-5}$ as a stopping rule to define the convergence of the ELBO. As before, we executed 20 distinct runs of the algorithm with different starting points, keeping the one with the highest ELBO to draw the inference. We assessed the accuracy of the estimated clusters using the ARI between the posterior point estimate and the true cluster.

As before, Figure 11 shows that the clustering performance of the proposed GLocal DP was better than HDP and the clustering accuracy improves with the increasing separation in the local variables. Furthermore, at the local-level the GLocal

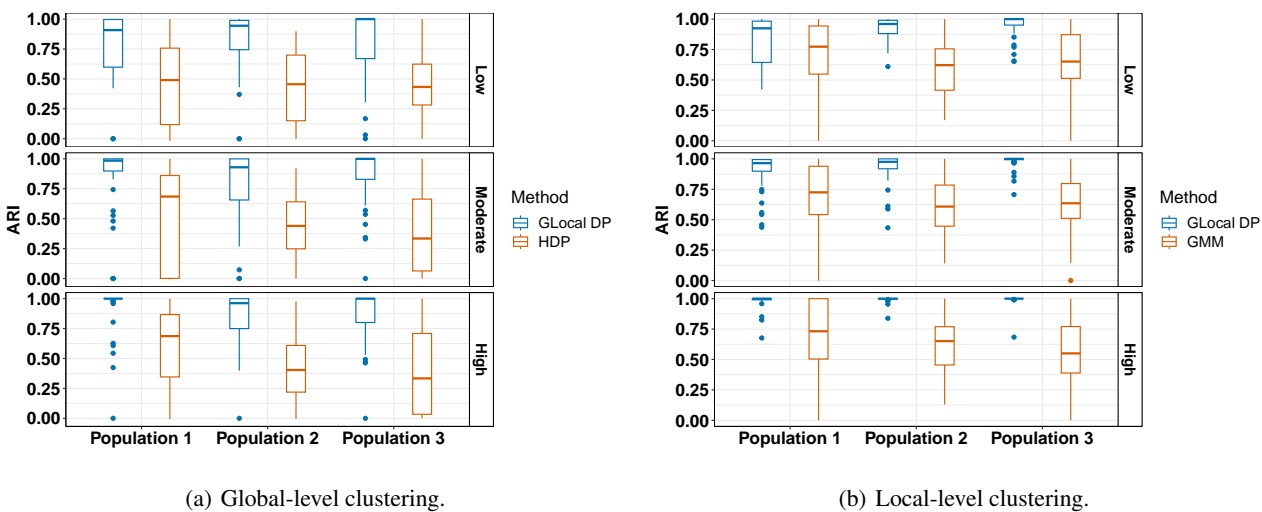

(a) Global-level clustering.

(b) Local-level clustering.

*Figure 11.* Five-dimensional Global Variables: Comaprison of Clustering Accuracy of GLocal DP with (a) HDP at the global-level and (b) GMM at the local-level with varying separation in the local variables.

DP clustering accuracy is higher than a GMM on each group separately.

Next, we varied the sample sizes in each group, i.e., for $j = 1, 2, 3$ the sample size, $n_j = 100, 200$, and $500$. We fixed the degree of separation in the global and local variables for the three populations. In particular, we set $\lambda_G = 0.1$ in (G.5), allowing moderate separation in the global variables. Furthermore, we set $\lambda_L = 0.1$ in (G.4), allowing low separation in the local variables. We naively refer to the separation of the local variable by the level of information it contains e.g., "Low" level of information in local variable corresponds to low separation in the local variable. Even in this case, when the local variables contain "low information", Figure 12(b) shows that the clustering accuracy of the GLocal DP is either comparable to or better than the HDP at the global-level across sample sizes. This underscores the importance of incorporating local variables, when available to improve clustering accuracy, further highlighting that our model is different than the HDP at the global-level. Furthermore, our joint clustering of global and local variables facilitates information sharing across variable types, enhancing clustering performance at the local-level. Specifically, the GLocal DP demonstrates superior clustering accuracy compared to applying a GMM independently to each group, with accuracy further improving as the sample size increases.

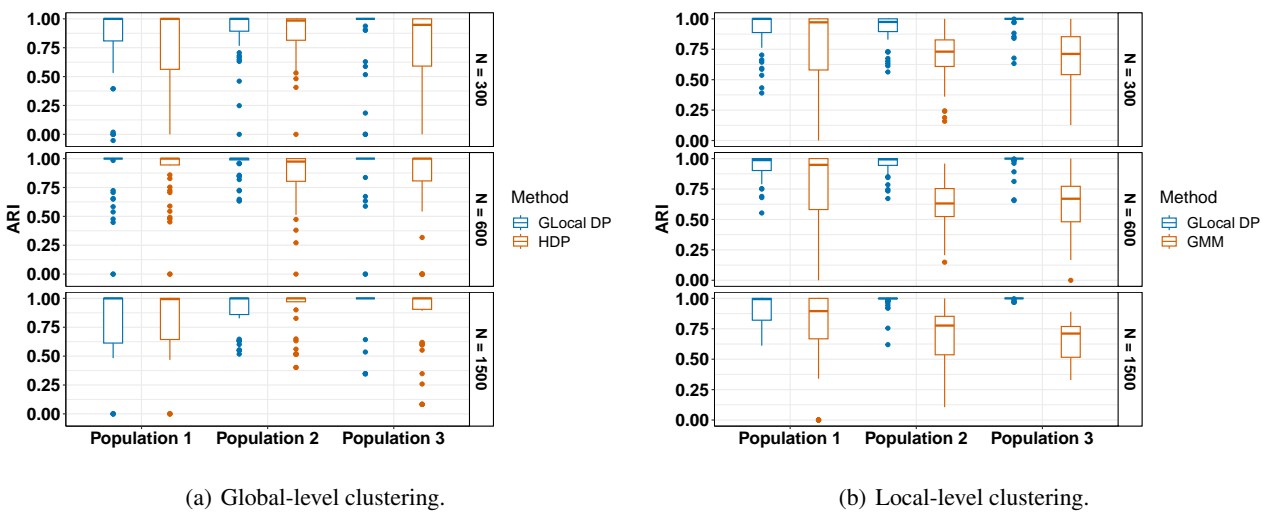

(a) Global-level clustering.

(b) Local-level clustering.

*Figure 12.* Varying Sample Size: Comaprison of clustering accuracy of GLocal DP with (a) HDP at the global-level and (b) GMM at the local-level.

### G.5. Impact of the local variables in clustering

In this subsection, we examine the impact of local variables on clustering performance. In the main manuscript, we emphasized that our method differs from the HDP, even at the global-level. To illustrate this, we compare clustering accuracy when the global variables are two-dimensional. We assumed that there were one, two, and three local variables for populations 1, 2, and 3, respectively and we generated the data from (G.1). Furthermore, the true parameters and the true mixture weights corresponding to the local variables are drawn from (G.4) with $\lambda_L = 0.01$, resulting in high separability among the local variables. Similarly, the true parameters and mixture weights corresponding to the global variables are drawn from (G.5) with $\lambda_G = 1$, making the separations between the global variables to be low. All other simulation details were same as in Section G.4. Figure 13(a) shows that in the presence of highly overlapped global variables, GLocal DP is able to identify clusters with perfect accuracy. In contrast, HDP struggles to distinguish these overlapping clusters, as shown in Figure 13(b). The group-specific local variables are shown in Figure 14 along with the estimated local-level clusters, which further show the high separability of the local variables. This separation of local variables aids in identifying overlapping clusters of global variables, underscoring the significance of incorporating group-specific local variables, when available, to enhance the clustering of shared variables.

### G.6. Comparison with Versatile Hierarchical Dirichlet Process

In the main manuscript, we emphasized that, despite sharing similar modeling objectives, the formulation of the vHDPMM proposed by Dinari & Freifeld (2020) is fundamentally different from our proposed GLocal DP mixture model. The vHDPMM adopts a hierarchical specification in which global variables are modeled first, followed by local variables that are conditionally dependent on the global cluster assignments. In contrast, the GLocal DP is defined jointly leading to a more flexible model specification. Under our formulation, for any two distinct observations $i$ and $i'$ within the same group $j$, if $t_{ji} = t_{ji'}$, then it follows directly that $k_{jt_{ji}} = k_{jt_{ji'}}$; that is, if the observations are assigned to the same local cluster, they necessarily share the same global cluster. This dependency arises naturally from the joint modeling structure. In the vHDPMM, however, the local clusters are defined conditionally on global assignments, and this hierarchical construction imposes the restriction that for any group $j$, if $i \in s_j^k$ and $i' \in s_j^{k'}$, where $k \neq k'$, then the two observations $i$ and $i'$ cannot have the same global cluster, even if they share the same local feature. Furthermore, our model exactly reduces to the HDP in the absence of local variables for all the groups. However, the vHDPMM, even in the absence of local variables for all the groups, is not exactly the HDP mixture model. Finally, the two models differ in their approaches to posterior inference. We develop a scalable variational inference algorithm, while the vHDPMM employs a split-merge based MCMC sampling scheme.

Using the cosegmentation example presented in Dinari & Freifeld, 2020, for an image $j$, if two pixel observations $i$ and $i'$ share the same local cluster corresponding to spatial location, GLocal DP highlights that they automatically share the same global cluster (given by the RGB colors). However, vHDPMM first clusters the observations $i$ and $i'$ into global cluster (given by the RGB colors) and conditional on same global cluster, models the local clusters (given by spatial locations). This, we feel is restrictive (also possibly counter-intuitive as in the cosegmentation example) and our model provides a natural method of estimating clusters, both corresponding to the global- and local-level. To highlight this, we performed some simple simulation studies.

First, we generated data from the vHDPMM model with three distinct groups or populations, setting the sample size in each group to $n_j = 100$, for $j = 1, 2, 3$. The global variables were drawn from a six-component trivariate Gaussian mixture model. While our VI–based approach is designed to accommodate varying dimensionality of local variables across different populations, the publicly available implementation of the vHDPMM model is restricted to settings where the local variables share a common dimensionality across all populations. To ensure comparability under this constraint, we simulated two local variables for each group from a five-component mixture of bivariate Gaussian distributions. Additionally, we considered a setting where both the global and local clusters were well separated. We then applied both the vHDPMM and the proposed GLocal DP model to the simulated data and evaluated clustering accuracy at both the global and local levels. As shown in Table 1, the vHDPMM performs well, as expected, given that the data were generated from its own model. Furthermore, the GLocal DP also achieves comparable clustering accuracy.

Next, we generated the data from the GLocal DP model with three groups/populations. All other simulation settings were the same as before. We then applied both the proposed GLocal DP and the vHDPMM to the same dataset to evaluate and compare their clustering performance at both the global and local levels. The clustering accuracy results, summarized in Table 2, indicate that the GLocal DP model achieves high accuracy, as expected given that the data were generated from

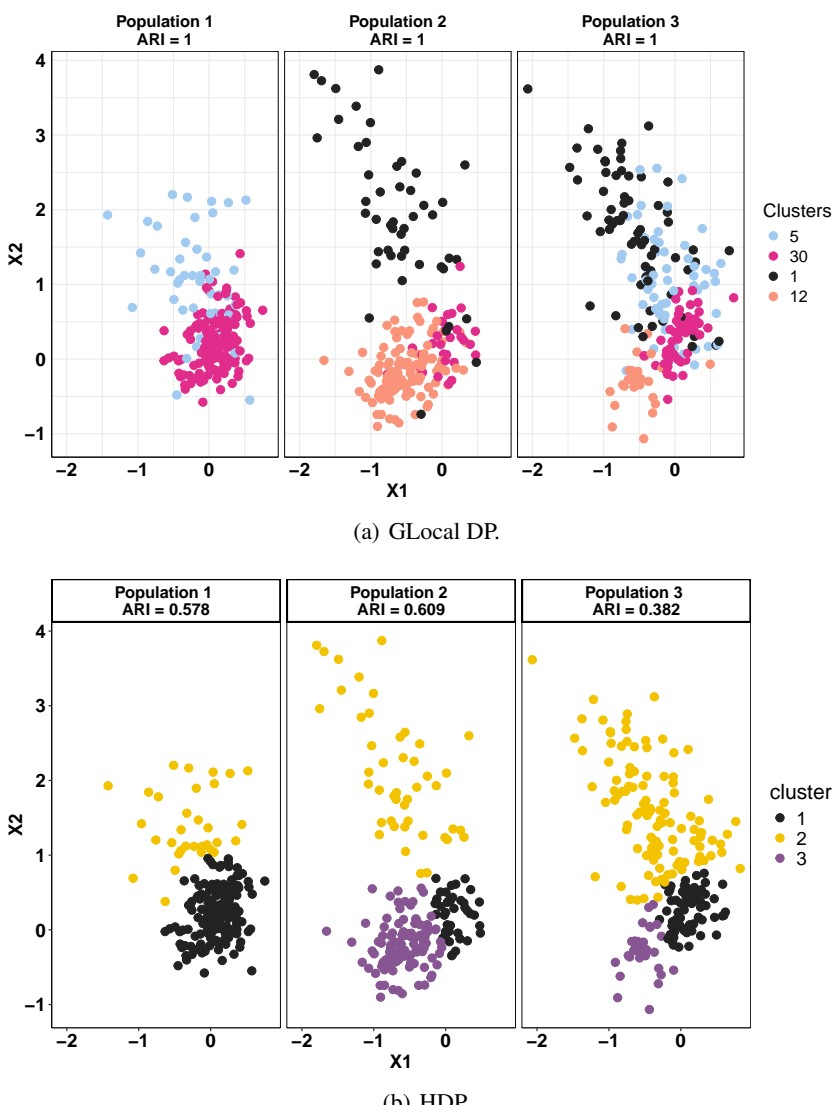

(a) GLocal DP.

(b) HDP.

*Figure 13.* Two-dimensional Global Variables: Comparison of clustering Accuracy of (a) GLocal DP and (b) HDP when the global are highly overlapped and local variables are separated. The colors indicate the estimated clusters. Adjusted Rand index is reported at the top of each panel.

its own generative process. Notably, the vHDPMM also performs comparably well, yielding similar levels of clustering accuracy across both global and local levels. These results suggest that, in settings characterized by well-separated clusters both models exhibit robust and comparable performance, even when the data-generating mechanism aligns more closely with one model over the other.

Finally, we considered a more challenging clustering scenario by increasing the sample size within each group and reducing the separation among clusters. Specifically, we set $n_j = 200$ for each group $j = 1, 2, 3$, and generated data from the vHDPMM with low separation in both global and local features. All other simulation settings remained consistent with previous experiments. We then applied both the vHDPMM and the proposed GLocal DP model to this dataset. As shown in Table 3, the GLocal DP model outperforms the vHDPMM in terms of clustering accuracy at both the global and local levels. Importantly, this superior performance is observed even though the data were generated from the vHDPMM. This advantage can be attributed to the joint modeling approach of the GLocal DP, which enables more coherent inference, in contrast to the hierarchical specification of the vHDPMM. These findings highlight the increased flexibility of the GLocal DP framework, demonstrating its ability to accurately recover complex clustering structures even under misspecification of

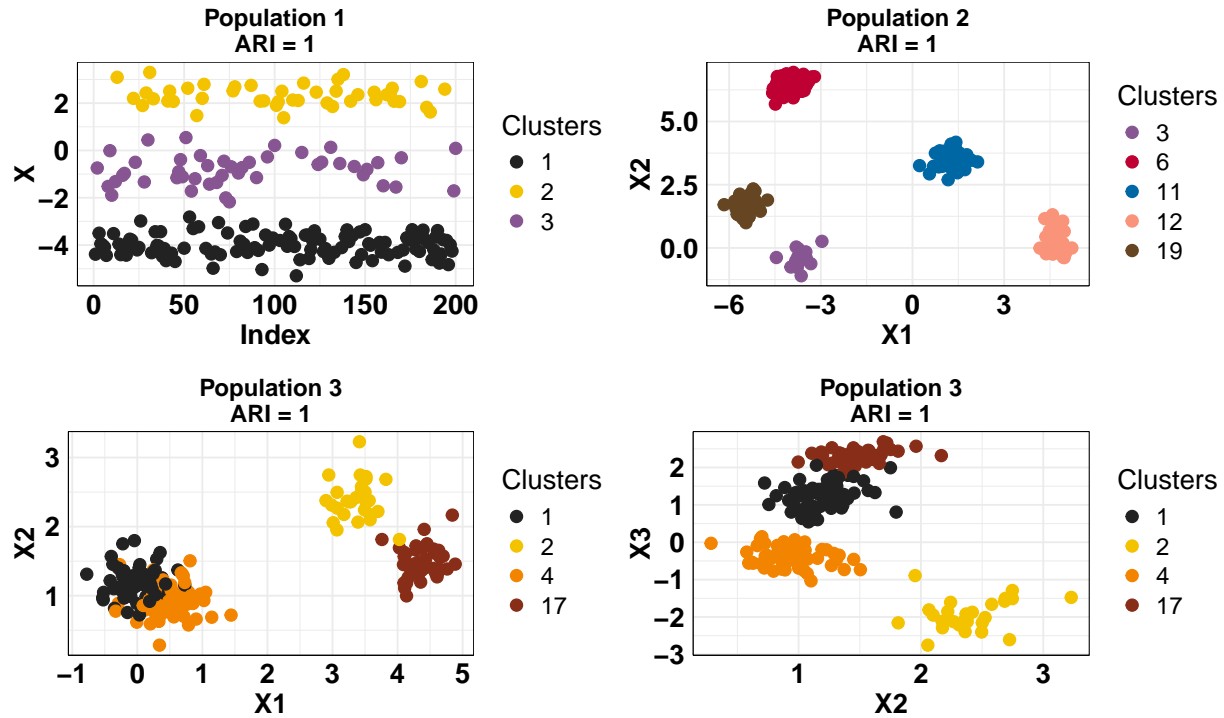

*Figure 14.* Local-level clustering of local variables by GLocal DP. The colors indicate the estimated clusters. Adjusted Rand index is reported at the top of each panel.

*Table 1.* Comparison of clustering performance of vHDPMM and GLocal DP when the data are generated from the vHDPMM. Sample size in each group $n_j = 100$ for $j = 1, 2, 3$. Both the global and local clusters have high separation. The accuracy of clustering was assessed using the adjusted Rand index (ARI) between the estimated and true cluster.

| Model | Global-level Clusters | | | Local-level Clusters | | |
|---|---|---|---|---|---|---|
| | Group 1 | Group 2 | Group 3 | Group 1 | Group 2 | Group 3 |
| vHDPMM | 1.00 | 0.98 | 1.00 | 0.59 | 0.79 | 0.44 |
| GLocal DP | 1.00 | 1.00 | 1.00 | 0.59 | 0.79 | 0.34 |

the true data-generating process.

## H. Real Data Analysis with Versatile Hierarchical Dirichlet Process

In this section, we analyze the data from our motivational problem in the main manuscript with the vHDPMM. Recall that we analyzed four gastrointestinal (GI) tract cancers, namely esophageal, stomach, colon, and rectal cancers. The dataset included log-transformed gene expression for 60,483 genes across 173, 407, 512, and 177 patients with esophageal, stomach, colon, and rectal cancers, respectively. Since the publicly available implementation of the vHDPMM model is restricted to settings where the local variables share a common dimensionality across all populations, we modified the selection of the clinical variables. Particularly, we considered the number of cigarettes smoked per day as a local variable for esophageal cancer, the number of positive lymph nodes for stomach cancer, the pre-operative and pre-treatment CEA for rectal cancer, and BMI as the local variable for colon cancer. After excluding patients with missing clinical data, the final sample sizes were 92, 363, 263, and 120 for esophageal, stomach, colon, and rectal cancers, respectively.

As before, we performed PCA on the combined gene expression data from the four cancers and retained the top ten PCs as the global variables. We ran the vHDPMM on the final dataset for $10,000$ iterations. Furthermore, as before, for visualization, we reduced the original combined gene expression data to two dimensions using the UMAP. Figure 15 shows

*Table 2.* Comparison of clustering performance of vHDPMM and GLocal DP when the data are generated from the GLocal DP mixture model. Sample size in each group $n_j = 100$ for $j = 1, 2, 3$. Both the global and local clusters have high separation. The accuracy of clustering was assessed using the adjusted Rand index (ARI) between the estimated and true cluster.

| Model | Global-level Clusters | | | Local-level Clusters | | |
|---|---|---|---|---|---|---|
| | Group 1 | Group 2 | Group 3 | Group 1 | Group 2 | Group 3 |
| vHDPMM | 1.00 | 1.00 | 1.00 | 1.00 | 1.00 | 0.79 |
| GLocal DP | 1.00 | 1.00 | 1.00 | 1.00 | 1.00 | 1.00 |

*Table 3.* Comparison of clustering performance of vHDPMM and GLocal DP when the data are generated from the vHDPMM. Sample size in each group $n_j = 200$ for $j = 1, 2, 3$. Both the global and local clusters have low separation. The accuracy of clustering was assessed using the adjusted Rand index (ARI) between the estimated and true cluster.

| Model | Global-level Clusters | | | Local-level Clusters | | |
|---|---|---|---|---|---|---|
| | Group 1 | Group 2 | Group 3 | Group 1 | Group 2 | Group 3 |
| vHDPMM | 0.33 | 0.00 | 0.95 | 0.29 | 0.00 | 0.50 |
| GLocal DP | 0.36 | 0.82 | 1.00 | 0.24 | 0.50 | 0.48 |

the UMAP embeddings colored by the estimated global- and local-level clusters obtained from the vHDPMM, while Figure 16 shows kernel density plot of local variables, segregated by the estimated local-level clusters. Finally, as in the main manuscript, we plotted the Kaplan-Meier survival curves for each of the identified cancer subpopulations in Figure 17, colored by the estimated local-level clusters obtained from the vHDPMM.

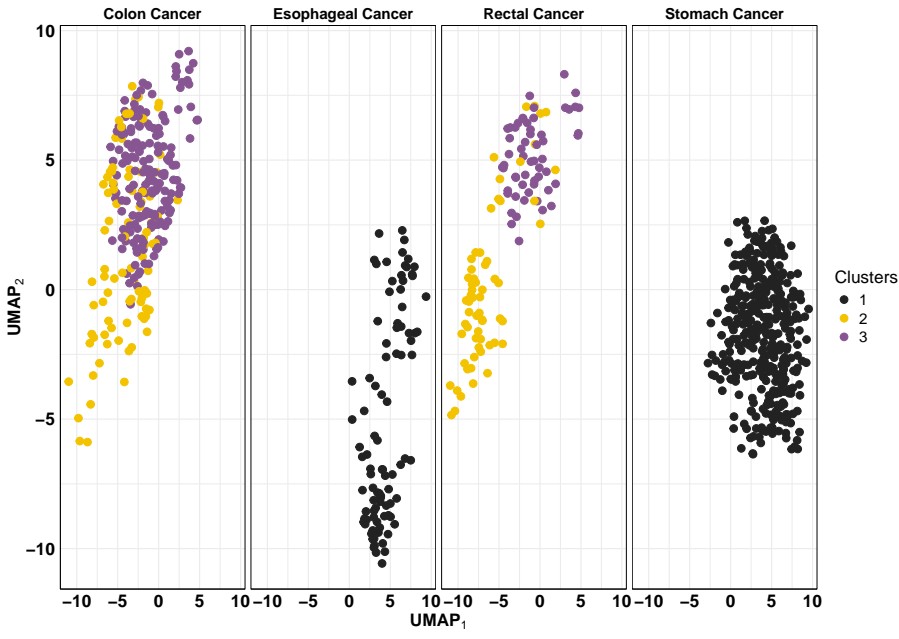

(a) Global-level clusters.

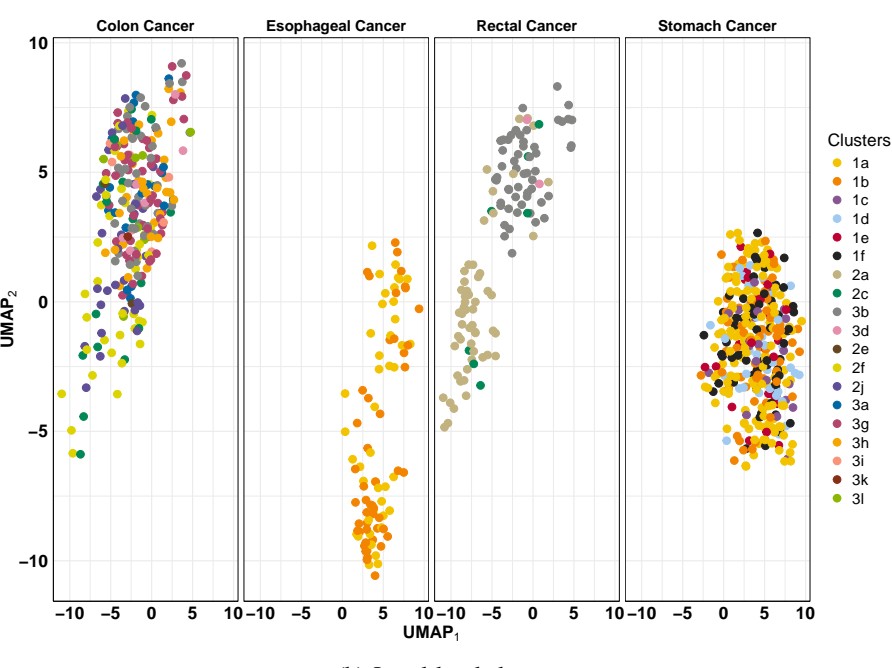

(b) Local-level clusters.

*Figure 15.* Global variables. (a) The colors indicate global-level clusters estimated by the vHDPMM. (b) The colors indicate the estimated local-level clusters by the vHDPMM.

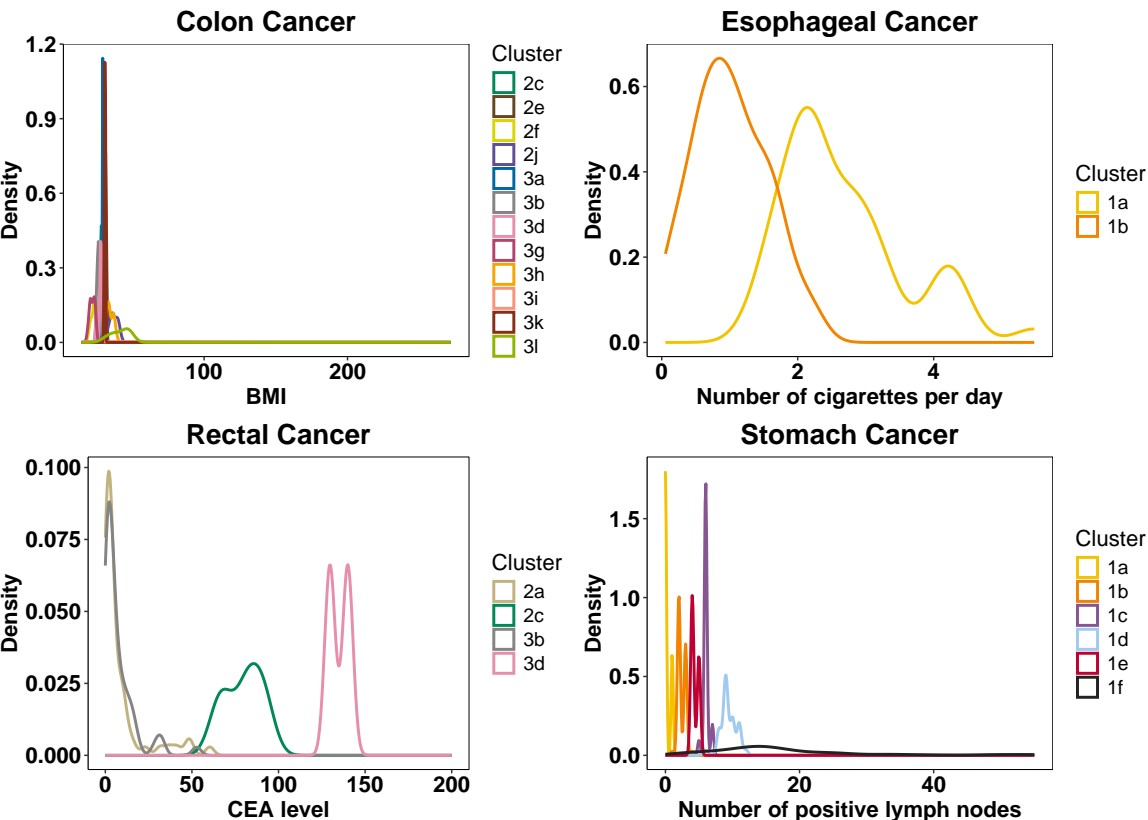

*Figure 16.* Kernel density plot of local variables, colored by the local-level clusters estimated by the vHDPMM.

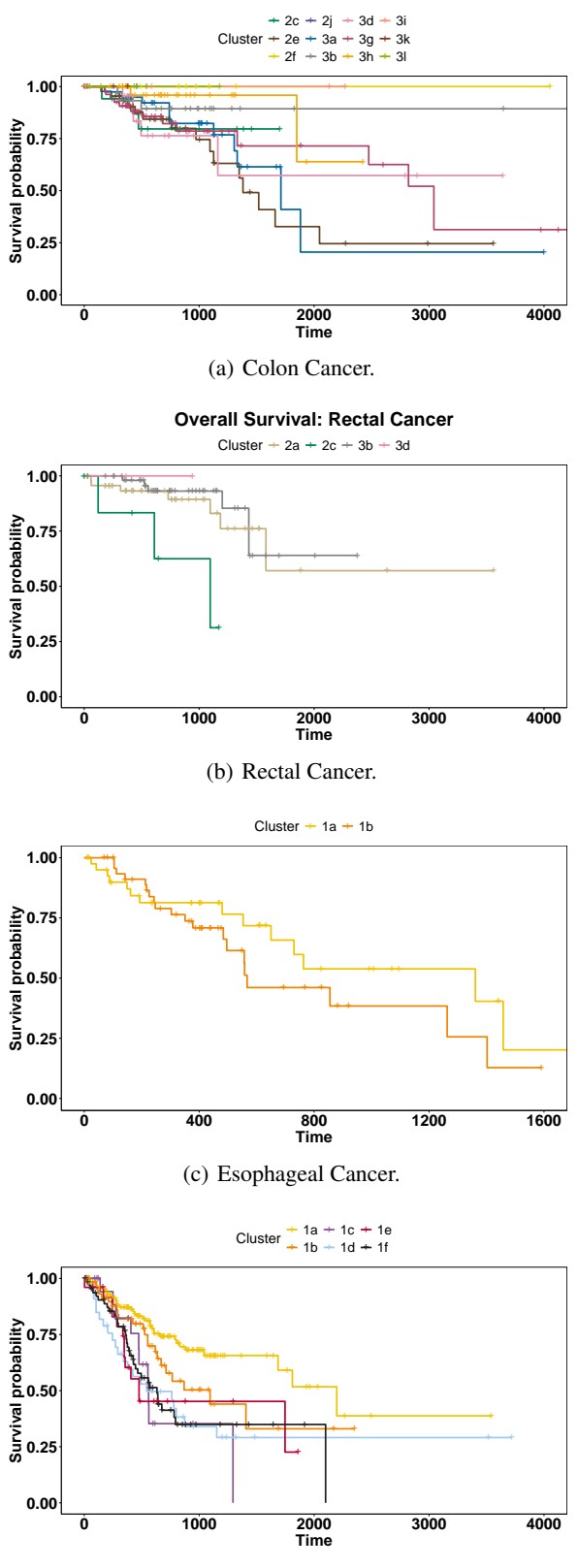

(a) Colon Cancer.

(b) Rectal Cancer.

(c) Esophageal Cancer.

(d) Stomach Cancer.

*Figure 17.* Survival curves according to local-level clusters estimated by the vHDPMM for different cancers.

| Acronym | Full Form |
|---|---|
| DP | Dirichlet process |
| GEM | Griffiths, Engen and McCloskey distribution |
| HDP | Hierarchical Dirichlet process |
| nested DP | Nested Dirichlet process |
| GLocal DP | Global-Local Dirichlet process |
| GMM | Gaussian mixture model |
| TCGA | The Cancer Genome Atlas |
| MCMC | Markov chain Monte Carlo |
| VI | Variational inference |
| CAVI | Coordinate ascent variational inference |
| KL divergence | Kullback-Leibler divergence |
| ELBO | Evidence lower bound |
| ARI | Adjusted Rand index |
| GI | Gastrointestinal |
| CEA | Carcinoembryonic antigen |
| BMI | Body mass index |
| PCA | Principal component analysis |
| PC | Principal component |
| UMAP | Uniform manifold approximation and projection |

*Table 4.* List of acronyms and their full forms.

| Notation | Definition |
|---|---|
| $\boldsymbol{x}_{ji}$ | Observation $i$ from group $j$ |
| $\boldsymbol{x}_{ji}^L$ | Local variables in group $j$ |
| $\boldsymbol{\theta}_{ji}^L$ | Local parameters (factors) |
| $\boldsymbol{x}_{ji}^G$ | Global variables shared across groups |
| $\boldsymbol{\theta}_{ji}^G$ | Global parameters (factors) |
| $G_j$ | Random measure corresponding to group $j$ |
| $U_j$ | Group-specific base measure for group $j$ |
| $V$ | Common base measure |
| $\otimes$ | Measure product |
| $F_1(\boldsymbol{x}_{ji}^L \mid \boldsymbol{\theta}_{ji}^L)$ | Conditional distribution of local variables, conditional on local factors |
| $F_2(\boldsymbol{x}_{ji}^G \mid \boldsymbol{\theta}_{ji}^G)$ | Conditional distribution of global variables, conditional on global factors |
| $\psi_{jt}^L$ | Local atoms corresponding to $G_j$ in group $j$ |
| $\phi_k$ | Shared global atoms corresponding to $V$ |
| $t_{ji}$ | Local-level cluster label of observation $i$ in group $j$ |
| $k_{jt_{ji}}$ | Global-level cluster label of observation $i$ in group $j$ |
| $G_j^{T,K}$ | Truncated measure corresponding to $G_j$, truncated at levels $T$ and $K$ |
| $P^{\infty,\infty}(\boldsymbol{\theta})$ | Prior distribution of parameters $\boldsymbol{\theta}$ under GLocal DP prior |
| $P^{T,K}(\boldsymbol{\theta})$ | Prior distribution of parameters $\boldsymbol{\theta}$ under truncated GLocal DP prior |
| $m^{\infty,\infty}(\boldsymbol{x})$ | Marginal distribution of data $\boldsymbol{x}$ under GLocal DP prior |
| $m^{T,K}(\boldsymbol{x})$ | Marginal distribution of data $\boldsymbol{x}$ under truncated GLocal DP prior |
| $\pi^{\infty,\infty}(\boldsymbol{\theta}|\boldsymbol{x})$ | Posterior distribution of parameters $\boldsymbol{\theta}$ under GLocal DP prior |
| $\pi^{T,K}(\boldsymbol{\theta}|\boldsymbol{x})$ | Posterior distribution of parameters $\boldsymbol{\theta}$ under truncated GLocal DP prior |
| $q(t_{ji}; \{\xi_{jit}\}_{t=1}^T)$ | Variational distribution of $t_{ji}$, which is a multinomial distribution |
| $q(k_{jt}; \{\rho_{jtl}\}_{l=1}^K)$ | Variational distribution of $k_{jt}$, which is a multinomial distribution |
| $q(v_k; \bar{a}_k, \bar{b}_k)$ | Variational distribution of $v_k$, which is a beta distribution |
| $q(u_{jt}; \bar{a}_{jt}, \bar{b}_{jt})$ | Variational distribution of $u_{jt}$, which is a beta distribution |
| $q(\alpha; s_1, s_2)$ | Variational distribution of $\alpha$, which is a gamma distribution |
| $q(\gamma; r_1, r_2)$ | Variational distribution of $\gamma$, which is a gamma distribution |
| $q(\boldsymbol{\mu}_k, \boldsymbol{\Lambda}_k; \boldsymbol{m}_k, \lambda_k, c_k, \boldsymbol{D}_k)$ | Variational distribution of $\{\boldsymbol{\mu}_k, \boldsymbol{\Lambda}_k\}$ is a normal-Wishart distribution |
| $q(\boldsymbol{\mu}_{jt}, \boldsymbol{\Lambda}_{jt}; \boldsymbol{m}_{jt}, \lambda_{jt}, c_{jt}, \boldsymbol{D}_{jt})$ | Variational distribution of $\{\boldsymbol{\mu}_{jt}, \boldsymbol{\Lambda}_{jt}\}$ is a normal-Wishart distribution |
| $\hat{k}_{jt} = \arg\max_{k=1,\ldots,K} \hat{\rho}_{jtk}$ | Optimal clusters corresponding to $k_{jt}$ obtained from VI |
| $\hat{t}_{ji} = \arg\max_{t=1,\ldots,T} \hat{\xi}_{jit}$ | Optimal local-level clusters obtained from VI |

*Table 5.* Notation and corresponding definitions.

