# OpenReview forum: "Global-Local Dirichlet Processes for Clustering Grouped Data in the Presence of Group-Specific Idiosyncratic Variables"
_ICML.cc/2025/Conference — ICML 2025 poster_

### Official Review · Reviewer_xxvp · 2025-03-18

**Overall Recommendation:** 3

**Summary:**

The article presents a new method for performing Bayesian nonparametric clustering on data sets with both global and local variables, i.e. data sets for which some variables are only observed for a subset of the individuals. The paper presents a novel formulation of a clustering model that allows for global clusters on the variables observed for all individuals, and local clusters discriminating within those observed in specific subsets of individuals.

A variational bound is introduced to perform inference through optimisation, and experiments are run on simulated data and a pan-cancer genomics data set. Comparisons are made with the same model specification with inference performed with MCMC, alongside the Hierarchical Dirichlet Process (HDP) for global clustering and finite Gaussian Mixture Models (GMM) for the local clustering.

##### update after rebuttal

I am satisfied with the accessible parts of the authors' response. However, since they linked to external media which is not permitted in the openreview system, I have kept my score the same.

**Claims And Evidence:**

The methods are well-motivated and explained well. The theoretical grounding is reasonable coherent and rigorous. The experimental evaluation is a little light, relying on one simulated data example and one real data set, although each of them is performed thoroughly. The restriction to Gaussian likelihoods is quite strict, but will presumably be diversified in future work.

**Essential References Not Discussed:**

There are no major references missing from a machine learning perspective. It is likely that the cancer/genomics references could be more thorough, but the ICML audience will not necessarily mind so much.

**Experimental Designs Or Analyses:**

The experimental design seems reasonable. The use of a single clustering index (Adjusted rand index) is a potential drawback, as there are multiple ways of assessing cluster algorithms. In the simulated data set, we have a "ground truth" to compare against, but this evaluation method will not generalise to real data.

There is no real comparison to competing methods for the real data set, which would be of interest. There is also no use of any kind of model fit index (BIC? ICL) to evaluate the model fit.

The assessment of the computational properties of the inference is clear and fairly thorough, e.g. computational time, convergence criteria etc.

**Methods And Evaluation Criteria:**

The benchmark data sets are appropriate for the data set but are limited to one simulated study and one real data set. It is possible that finding real data sets with the appropriate structure for this model specification is somewhat challenging.

**Other Comments Or Suggestions:**

The quality of the written English is generally good.

**Other Strengths And Weaknesses:**

In general, this is an interesting methodological development that is well-motivated by an interesting application challenge. There are some extensions that might be necessary before this becomes generally usable for all problems of this nature, but the article itself is a reasonable contribution.

**Questions For Authors:**

The presence of variables for some subgroups and not others is going to be informative to the global clustering. Does the method aim to make use of this? E.g. PSA is only measured when prostate cancer is also suspected: Would the presence of a PSA measurement represented as an indicator variable itself be informative to global clustering? Are you allowing the algorithm to make use of this information?

What is the relevance of existing methods for missing data? We are running into a sort of Missing at Random/Missing Completely at Random/Missing Not at Random kind of situation where the presence or absence of data is potentially going to be informative in itself. Where does this framework fall within the MAR/MCAR/MNAR set of assumptions?

What would happen if you treated the local variables as just “missing” in a Bayesian framework for all the other (non-local) individuals?

What is the influence of the proportion of local vs global variables? To consider two extremes:
1. Two data sets bound together concerning two different illnesses, with different data collected for each of them, and the only shared variables being (e.g.) age and sex. What kind of structure will the method uncover here?
2. Two heavily overlapping data sets with many shared variables and only one or two “extra” local variables per population. What happens here? Would a whole separate clustering be “discovered” for the limited extra variables? Could it be done in a way that builds on the dominant narrative from the shared variables?

How does the inference work for the discrete variables exactly? Can you use the gradients for the continuous variables and some other solution for the discrete ones? Is everything represented as a smooth continuous target function somehow? How robustly can you assess the convergence of a discrete optimisation algorithm?

Does putting a gamma prior on the concentration really solve the concentration/specification problem for DPs? Is the issue of inconsistency of the number of clusters for DPs generally overstated if a relatively simple solution exists?

The use of only Gaussian likelihoods is quite a hard constraint and likely to be false in reality much of the time. What is the relationship between the mixture component likelihood specification and the underlying clustering structure? Is it more important to specify the right data likelihood or the right underlying clustering structure? What kind of problems will you run into if you have the correct underlying clustering design but a misspecified likelihood? How easy would it be to adapt this method to non-Gaussian likelihoods? Why have you not done this? Can the PCA components in the real data example be safely assumed to be Gaussian?

What if there are variables in-between local and global? i.e. there are a global variables shared between all individuals, local variables unique to specific subgroups, but also mid-level variables shared between some subgroups but not all of them? Would you have to introduce an extra component in equation 1? And extra indicator variables in addition to t and k? Could this become a more generic framework that could incorporate varying degrees of data granularity in that respect?

Why did you only include a single real data set? Is it a challenge to find data sets with this structure in practice? Is it hard to perform the relevant data formatting and similar to prepare potential target data sets for this analysis pipeline?

**Relation To Broader Scientific Literature:**

The broader scientific literature is fairly well described, including the historical 1973 Ferguson DP article followed by generalisations to the Hierarchical and Nested DPs. The more recent results concerning inconsistency of the DP for the number of clusters is acknowledged and some effort is made to counteract the potential resulting issues.

**Theoretical Claims:**

The theoretical claims and results appear sound but I have not checked them thoroughly.

---

> ### Author Rebuttal · Authors · 2025-03-31
>
> We sincerely thank the reviewer for all the comments and questions, responses to some of which are in the textbox below.
>
> - **Question related to allowing our algorithm to make use of information on presence/absence of some variable***
>
> **Response**: The presence of PSA measurement indicates prostate cancer. But since whether the patient has prostate cancer or not is a given information for our method, which determines the groups, we do not think it provides extra information for global clustering other than the PSA values themselves. However, as a possible future research, our model could be extended to achieve simultaneous clustering of groups and within-group observations. In such a context, the presence or absence of variables for some subgroups and not others can possibly be informative in identifying similar cancer subtypes. For example, the presence of PSA measurement represented as an indicator variable itself would be informative in identifying prostate cancer as a possible secondary cancer associated with the primary cancer of the given patient(s).
>
> - **Question related to relevance of existing methods for missing data in our setup**
>
> **Response**: Thank you for your question. While our framework does not follow conventional missing data mechanisms, the absence of certain clinical variables can be viewed as Missing Not at Random (MNAR) when missingness is tied to cancer type. For example, the absence of PSA measurements likely indicates the patient does not have prostate cancer, where PSA is a key diagnostic marker. More broadly, our approach accommodates scenarios where different groups have distinct relevant variables, which may appear missing not due to randomness but because they hold no relevance for those groups.
>
> - **Question related to treating the local variables as just *missing***
>
> **Response**: As highlighted previously, our framework does not follow conventional missing data mechanisms. Treating local variables as "missing" and imputing them may not yield meaningful grouped clustering. For example, in our pan-cancer analysis, CEA is a key biomarker for colorectal cancer but irrelevant for esophageal and stomach cancers. Imputing CEA before clustering could obscure biologically meaningful subgroupings. Thus, explicitly modeling local variables is essential for interpretable and valid clustering results.
>
> - **Question on influence of the proportion of local vs global variables**
>
> **Response**: This is an excellent question. Please see our detailed explanation in the link to the PDF at the end of the response.
>
> - **Question on inference for discrete variables**
>
> **Response**: Our GLocal DP model and VI-based algorithm can accommodate both continuous and discrete variables in the global and local components. As long as the parameters of discrete variables are continuous, our VI algorithm can optimize the ELBO using coordinate ascent. However, if the parameters were discrete, alternative discrete optimization methods or continuous relaxations might be needed, which could be explored in future research.
>
> - **Question on putting a gamma prior**
>
> **Response**:  We acknowledge that inconsistency in the number of clusters is an important and active area of research. Our intention is not to understate this challenge; rather, we adopted a non-informative gamma prior as one possible modeling choice to help mitigate the severity of clustering inconsistency.
>
> - **Question related to use of only Gaussian likelihoods in our model**
>
> **Response**: Our VI-based algorithm is designed for Gaussian distributions but is inherently flexible, supporting arbitrary likelihood and prior choices, including DP (Ferguson, 1973) and HDP (Teh et al., 2006). The Gibbs sampler in the Appendix is similarly extendable. Moreover, our VI approach generalizes to exponential family distributions (Blei & Jordan, 2006). The rationale for assuming Gaussian distribution for PCA components in the real data example are shown in the PDF.
>
> - **Question related to variables in-between local and global**
>
> **Response**: This is an excellent question. Please see our detailed answer in the link to the PDF at the end of the response.
>
> - **Question on including a single real data set**
>
> **Response**: Our team primarily works with genomics data, which is why we used TCGA as an example. Similar multi-omics datasets, such as ICGC, share the same structure, but we did not use them due to significant preprocessing requirements. Beyond genomics, grouped data with both global and local variables arise in fields like business, social sciences, and other applied domains, for which our model would provide a convenient framework for analysis, as discussed in the detailed response PDF.
>
> We would like to **refer** the reviewer to the detailed point-by point responses to the questions, with additional simulations results, figures, and comments found in the PDF [here](https://drive.google.com/file/d/1Q-8oAM7nyEtGOURmVQF8E-ZTzwQl8IOi/view?usp=sharing).

---

### Official Review · Reviewer_f78g · 2025-03-21

**Overall Recommendation:** 2

**Summary:**

This paper considers the problem of clustering grouped data for which the observations may include group-specific variables in addition to the variables that are shared across groups.

To allow for these group-specific variables to aid in the clustering, the paper proposes a novel Bayesian nonparametric approach, termed global-local (GLocal) Dirichlet process, that models the ``global-local" structure of the observations across groups.

The paper characterizes the GLocal Dirichlet process using the stick-breaking representation and the representation as a limit of a finite mixture model. The paper theoretically quantifies the approximation errors of the truncated prior, the corresponding finite mixture model, and the associated posterior distribution.

The paper develops a fast variational Bayes algorithm for scalable posterior inference, which is illustrated with extensive simulations and a TCGA pan-gastrointestinal cancer dataset.

**Claims And Evidence:**

The results are sound.

**Essential References Not Discussed:**

The paper seems to have discussed essential references.

**Experimental Designs Or Analyses:**

The experiments are sound.

**Methods And Evaluation Criteria:**

The algorithm and experiments are sound.

**Other Comments Or Suggestions:**

n/a

**Other Strengths And Weaknesses:**

Strength: the paper uses group-specific variables to aid in the clustering
Weakness: the derivations seem straightforward, and the paper's impacts do not seem significant for ICML

**Questions For Authors:**

The derivations seem straightforward, and the paper's impacts do not seem significant for ICML.

What were the main technical innovations that overcame previously unresolved technical challenges?

**Relation To Broader Scientific Literature:**

To allow for these group-specific variables to aid in the clustering, the paper proposes a novel Bayesian nonparametric approach, termed global-local (GLocal) Dirichlet process, that models the ``global-local" structure of the observations across groups.

**Theoretical Claims:**

The algorithm is sound.

---

> ### Author Rebuttal · Authors · 2025-03-31
>
> We sincerely thank the reviewer for the comment.
>
> - **The derivations seem straightforward, and the paper's impacts do not seem significant for ICML. What were the main technical innovations that overcame previously unresolved technical challenges?**
>
> **Response**:  In this paper, our contributions are as follows.
>
> 1. We introduce a general Bayesian nonparametric framework, GLocal DP, for clustering grouped data by incorporating group-specific local variables. Most of the existing Bayesian or non-Bayesian clustering methods do not accommodate this idiosyncratic data structure.
>
> 2. We formally define the GLocal DP mixture model and establish its finite mixture model representation. We further demonstrate that, in the limit, the finite mixture model converges to the GLocal DP mixture model. In practical applications, this implies that a suitably truncated finite mixture model closely approximates the infinite GLocal DP mixture model. However, to provide a rigorous quantification of this approximation, we derive explicit truncation error bounds in Propositions 1 and 2 of the main manuscript. Importantly, while our derivations may appear straightforward, they constitute a nontrivial extension of the results in Ishwaran and James, 2001 due to the presence of both local and global components in our model.
>
> 3. In the Appendix of our main manuscript, we present a blocked Gibbs sampler for posterior inference in the GLocal DP, complementing the variational inference (VI)-based algorithm described in the main text. Since both inference approaches rely on a finite truncation of the infinite GLocal DP, our theoretical results provide practical guidelines for selecting appropriate truncation levels. Notably, existing Bayesian nonparametric models such as the hierarchical Dirichlet process (HDP, Teh et al., 2006) employ variational inference methods that assume a truncated variational family  (Teh et al., 2007; Wang et al., 2011), where the truncation level is typically chosen to exceed the expected number of clusters. While this heuristic is widely used, no established theoretical framework exists for determining truncation levels with explicit bounds on the approximation error. Since the GLocal DP reduces to the HDP in the absence of local variables across all groups, our truncation error bounds extend naturally to the HDP, thereby offering valuable theoretical guidance for selecting truncation levels in VI-based inference algorithms for the HDP.
>
> 4. We develop a VI-based algorithm for posterior inference in the GLocal DP and evaluate its performance through simulation studies. Our results demonstrate that the VI-based algorithm achieves clustering accuracy comparable to that of the MCMC-based Gibbs sampler while significantly improving computational efficiency in terms of both memory usage and runtime. Moreover, our VI-based inference algorithm exhibits high scalability with respect to both the number of groups and the number of observations per group.
>
> 5. We conduct additional simulations (found in the link to the PDF at the end), illustrating that incorporating local variables not only enables our model to accommodate complex data structures but also enhances the clustering performance of shared variables compared to methods that rely solely on shared variables.
>
> 6. Additionally, our proposed method extends beyond the application to cancer genomics to a general grouped clustering framework, wherein the available data consists of important group-specific variables apart from the shared variables.
>
> In summary, our novel Bayesian nonparametric model addresses a critical gap in the clustering of grouped data by explicitly incorporating group-specific idiosyncratic variables—an aspect not accounted for in existing literature. Furthermore, we establish theoretical truncation error bounds for the truncated GLocal DP prior and mixture model, offering a principled approach to selecting truncation levels in both MCMC- and VI-based inference algorithms. These theoretical results also extend to inference algorithms for the HDP, further enhancing their practical applicability. Finally, we develop a highly scalable VI-based inference algorithm for GLocal DP, which can be readily adapted for HDP-based models as well.
>
> We would like to **refer** the reviewer to the additional simulations supporting the novelty and usefulness of our method along with other contributions of our work, provided in the PDF, [here](https://drive.google.com/file/d/18DGSpV1sx2qKS-JjD7deY1O6OFHOsyWV/view?usp=sharing).

---

### Official Review · Reviewer_aX7e · 2025-03-23

**Overall Recommendation:** 3

**Summary:**

This paper addresses the problem of clustering grouped data where observations may include both group-specific and shared variables across groups.  The authors propose a novel Bayesian non-parametric approach called the global-local (GLocal) Dirichlet process. Unlike HDP, where clusters are derived from a common base measure, GLocal DP distributes clusters in each G_j across a global shared subspace and a group-specific local subspace. Where each cluster in G_j is characterized by both the "shared" distribution, which is modeled across global model, and the local distribution which is only modeled in the local group.

**Claims And Evidence:**

The claims in the paper are supported by the experiments.

**Essential References Not Discussed:**

I would like to point the authors to `Scalable and Flexible Clustering of Grouped Data via Parallel and
Distributed Sampling in Versatile Hierarchical Dirichlet Processes, Dinari and Freifeld, UAI 2020`.

Dinari and Freifeld discuss this exact setting in their paper, and propose a slightly different model to address it. They propose the Versatile HDP model, having a similar global-local structure, the difference is mostly around the way the group-wise weights are drawn. The larger difference is in the inference method, where the current paper propose a VI method, Dinari and Freifeld proposed a split-merge based MCMC inference method, with a large focus on per-group-size scalability.

**Experimental Designs Or Analyses:**

Experiments look well, see above (`Methods And Evaluation Criteria`) for a missing but required comparison.

**Methods And Evaluation Criteria:**

While not familiar with the real datasets myself, both them and and the synthetic datasets looks reasonable and a good way to evaluate the model.
The one thing I am missing  in the benchmarks is comparing with a DPMM individually each group, in the same way that was done with the GMM, that seems to me a like an important comparison that should be added, and an HDPMM where you do not differentiate between the global and local parts.
In addition, I would propose to use NMI and Purity metrics as well, which in my opinions has high value when comparing clustering with a varying number of clusters, and they complement the ARI.

**Other Comments Or Suggestions:**

I would suggest adding a notation table in the supplementary.

**Other Strengths And Weaknesses:**

The paper is well written, and the different model construction are very helpful for the understanding of the flow. The supplementary add significant value with the Gibbs sampler.

The main weakness of the paper related to the prior work by Dinari and Freifeld, which handles the same setting, and propose a slightly different model, reducing the novelty of the paper's main contributions (setting and model).
A bit lesser concern is the scalability, the population sizes evaluated in the paper are small, HDP generally is used to tackle datasets with thousands of documents, each with thousands of words, this is kinda lacking here.

Finally, see my comments on the evaluation methods for additional needed evaluations. Given that this paper does not compare versus other works, some additional work is needed to justify it, both in terms of results and practicality (running time, setup overhead, etc...)

**Questions For Authors:**

The following questions are issues that my greatly affect my score -

1) Comparison, differences and gap between this work and Dinari and Freifeld, UAI 2020.
2) Scalability, how does this method addresses datasets with large number of documents, or on the other side, very large groups (e.g. images for example)
3) What kind of distributions are supported? Is the the same and DP and HDP, which are very flexible, or are there limitations? Both in general, and limitations between the global and local parts. Can the VI support this?

**Relation To Broader Scientific Literature:**

The main contribution of the paper is proposing and addressing the global-local structure for multiple groups in a HDP-like setting, and while many of the discussed and relevant paper do not address this exact setting, there is a previous paper which proposed that setting prior, and addressed it in a similar fashion, see below.

**Theoretical Claims:**

I have checked both GLocal DP proposed constructions, and the truncation approximation bounds error evaluation, and they all seems sound.

---

> ### Author Rebuttal · Authors · 2025-03-31
>
> We sincerely thank the reviewer for all the comments.
>
> - **Comparison, differences and gap between this work and Dinari and Freifeld, UAI 2020.**
>
> **Response**: Dinari and Freifeld, 2020 discusses a setting in which data arises from multiple pre-defined groups and consists of variables that are shared across the groups (denoted by $x_{ji}$) as well as group-specific variables (denoted by $y_{ji}$), which is the same as in our case. They propose the versatile hierarchical Dirichlet processes (vHDP) to model this grouped data. However, there are some key differences in the modeling perspective of this method as compared to our proposed GLocal DP. Importantly, the two models are not special cases of each other.
>
> Particularly, their model is defined hierarchically by first modeling the shared global variables and then conditionally modeling the local variables conditional on the global clusters. Contrarily, our proposed GLocal DP mixture model is defined jointly for the global and local variables. Using our formulation, for two observations $i$ and $i' \neq i$ in the same group $j$, if $t_{ji}=t_{ji'}$, then automatically, they share the same global clusters i.e., $k_{jt_{ji}}=k_{jt_{ji'}}$. However, this is not the case for the vHDPMM, where the local clusters for individuals $i$ and $i'\neq i$ are defined conditional on their global clusters. This, we feel is restrictive (also possibly counter-intuitive as in the cosegmentation example of Dinari and Freifeld, 2020) and our model provides a natural method of estimating clusters, both corresponding to the global- and local-level.
>
> ### **Simulation Results**
>
> #### **Data from vHDPMM**
>
> To illustrate, we simulated data from the vHDPMM with three groups, where global variables followed a six-component trivariate Gaussian mixture and local variables followed a five-component bivariate Gaussian mixture. We then assessed clustering accuracy (measured by ARI between true and estimated clusters) for both global and local clusters using vHDPMM and our proposed GLocal DP.
>
> |  | **Global-level clusters** | | | **Local-level clusters** | | |
> |--|--|--|--|--|--|--|
> | **Groups**  | 1 | 2 | 3 | 1 | 2 | 3 |
> | **vHDPMM** | 1.00 | 0.98 | 1.00 | 0.59 | 0.79 | 0.44 |
> | **GLocal DP** | 1.00 | 1.00 | 1.00 | 0.59 | 0.79 | 0.34 |
>
> Table shows that the clustering accuracy of vHDPMM was good as expected, since the data was drawn from the vHDPMM. Furthermore, GLocal DP clustering accuracy is comparable to that from vHDPMM.
>
> #### **Data from GLocal DP**
>
> Next, we simulated data from the GLocal DP model with three groups, keeping all other settings unchanged. We then evaluated the clustering accuracy of both global and local clusters using the vHDPMM and our GLocal DP.
>
> |  | **Global-level clusters** | | | **Local-level clusters** | | |
> |--|--|--|--|--|--|--|
> | **Groups**  | 1 | 2 | 3 | 1 | 2 | 3 |
> | **vHDPMM** | -0.04 | -0.02 | -0.01 | 0.00 | -0.01 | 0.00 |
> | **GLocal DP** | 1.00 | 0.98 | 1.00 | 1.00 | 0.80 | 0.99 |
>
> Table shows that the clustering accuracy of GLocal DP was good as expected. However, the vHDPMM performed very poorly, both in terms of global- and local-level clustering. In summary, the GLocal DP offers greater flexibility, effectively adapting to different data-generating mechanisms while accurately capturing both global and local cluster structures.
>
> - **Question on scalability**
>
> **Response**: We conducted simulations to assess the scalability of our VI-based algorithm for the GLocal DP. Specifically, we considered two scenarios:
>
> 1. Varying the number of groups ($J$) while keeping the sample size per group ($n_j$) fixed.
> 2. Varying $n_j$ while keeping $J$ fixed.
>
> In the first scenario, we set $J = 5, 10, 100\$ with $n_j = 100$, and in the second, we set $n_j = 100, 200, 500$ with $J = 10$. All other settings matched those in the main manuscript. We see that computation time scales approximately linearly in both cases (see Figure 5 in the attached link to PDF at the end). These results highlight the efficiency of our VI-based approach even for large datasets, with large number of groups and/or sample sizes in each group.
>
> - **Question on the kind of distributions supported**
>
> **Response**: Our methodology is highly flexible and can accommodate a wide range of distributions
> for both the likelihood and prior, including those supported by DP (Ferguson, 1973) and the HDP (Teh et al., 2006). Furthermore, our VI-based approach can be readily generalized to settings where the data distribution belongs to the exponential family, as
> outlined in Blei and Jordan, 2006. Additionally, our model can integrate both continuous and/or discrete variables in both the global and local components.
>
> We would like to **refer** the reviewer to the detailed point-by point responses to all questions, simulations, figures, and our responses to additional comments/suggestions provided in the PDF, [here](https://drive.google.com/file/d/159nPJz-V6EN1euHOhPXRqgG6MZ-uBKl7/view?usp=sharing).

---

> > ### Comment · Reviewer_aX7e · 2025-04-07
> >
> > I thank the author for their response, and I urge them to use the remaining time to fill the full response in the open-review system, and not via an external link, such that it can be addressed properly.
> >
> > Re-reading Dinari and Freifeld, I do not think your claim here is true, for observatiions $i,i'!=i$, if they share the same table, they do share the global cluster. which is the sum of all tables belonging to that clusters (for the global part that is). Chance you might have mixed between the models?
> > While I can see the difference in the model itself, I am not convinced that it justifies a publication in ICML.
> >
> > The scalability experiments are not convincing enough, as the numbers are still much lower than you would expect in such a scenario.
> >
> > Again, I urge the authors to upload their full respsone to the system such that I could address it properly.
> >
> >
> > ---------------------------Follow up comment (as I cannot add another which will be visible to the authors)
> >
> > Thanks for the clarifications and the additional inputs, I now have better understanding of the diffrences between the two models, and the different scenarios each model address, I agree with the authors that the different models address different settings (althou with alot of similarities). Also both models can probably handle the different settings with limited success.
> >
> > In addition to the previous comments, I would encourage the authors to add a graphical model depicting the model, would greatly contribute to the understanding, quite possibly on the supplementary. And a short discussion on the diffrences between your work and the aforementioned paper.
> >
> > I am now more leaning into accepting the paper, however I am still not convinced that ICML is the correct venue for this work,  I belive AISTATS or UAI are more fiting venues, the experiments section should include the additional evaluations with real data as well.

---

> > > ### Author Response · Authors · 2025-04-07
> > >
> > > We thank the reviewer for the additional comments. Below, we provide our full response.
> > >
> > > **Benchmark comparing with a DPMM individually each group**
> > >
> > > As suggested by the reviewer, the table below reports mean (SD) of ARI over 50 runs for local-level clustering. GLocal DP consistently outperforms DPMM and GMM separately on each group across scenarios.
> > >
> > > | Method     | Pop. 1 - Low | Moderate | High   | Pop. 2 - Low | Moderate | High   | Pop. 3 - Low | Moderate | High   |
> > > |------------|--------------------|----------|------|--------------------|----------|------|--------------------|----------|------|
> > > | GLocal DP  | 0.84 (0.2)         | 0.93 (0.12) | 0.96 (0.12) | 0.91 (0.11)         | 0.95 (0.07) | 1 (0.01) | 0.96 (0.09)         | 0.99 (0.03) | 1 (0.01) |
> > > | DPM        | 0.76 (0.27)        | 0.89 (0.19) | 0.93 (0.17) | 0.67 (0.21)         | 0.8 (0.16)  | 0.88 (0.16) | 0.68 (0.26)         | 0.78 (0.24) | 0.92 (0.2) |
> > > | GMM        | 0.58 (0.33)        | 0.71 (0.27) | 0.67 (0.27) | 0.54 (0.23)         | 0.56 (0.24) | 0.56 (0.19) | 0.61 (0.27)         | 0.6 (0.22)  | 0.68 (0.28) |
> > >
> > >
> > > **Comment on NMI and Purity metrics**
> > >
> > > As per the reviewer's suggestion, the table below shows the mean (sd) of NMI of the global-level clustering, which shows that GLocal DP is superior to HDP.
> > >
> > > | Method     | Pop. 1 - Low | Moderate | High   | Pop. 2 - Low | Moderate | High   | Pop. 3 - Low | Moderate | High   |
> > > |------------|--------------------|---------------|--------------|--------------------|---------------|------------|--------------------|---------------|--------------|
> > > | GLocal DP  | 0.74 (0.27)        | 0.88 (0.22)   | 0.9 (0.22)   | 0.78 (0.19)        | 0.87 (0.19)   | 0.95 (0.1) | 0.88 (0.16)        | 0.92 (0.18)   | 0.95 (0.11)  |
> > > | HDP        | 0.31 (0.24)        | 0.35 (0.24)   | 0.32 (0.28)  | 0.26 (0.24)        | 0.31 (0.21)   | 0.3 (0.23) | 0.3 (0.21)         | 0.35 (0.24)   | 0.34 (0.25)  |
> > >
> > > Even at the local-level GLocal DP has superior NMI compared to DPMM on each group as shown below.
> > >
> > > | Method     | Pop. 1 - Low | Moderate | High   | Pop. 2 - Low | Moderate | High   | Pop. 3 - Low | Moderate | High   |
> > > |------------|--------------------|----------|------|--------------------|----------|------|--------------------|----------|------|
> > > | GLocal DP | 0.82 (0.2)         | 0.9 (0.13) | 0.95 (0.1) | 0.9 (0.1)          | 0.95 (0.06) | 0.99 (0.02) | 0.95 (0.09)       | 0.98 (0.04) | 1 (0.01) |
> > > | DPM       | 0.74 (0.27)        | 0.87 (0.19) | 0.93 (0.16) | 0.67 (0.19)        | 0.8 (0.15) | 0.88 (0.15) | 0.66 (0.24)       | 0.78 (0.23) | 0.92 (0.2) |
> > >
> > > As suggested by the reviewer, we evaluated the Purity metric and found that GLocal DP outperforms HDP at the global level, and both GMM and DPMM at the local level. Overall, GLocal DP consistently outperforms other methods across all metrics—ARI, NMI, and Purity—at both global and local levels.
> > >
> > >
> > > **Comment regarding the scalability**
> > >
> > > We acknowledge that the sample sizes are still much lower than what one would expect. However, we wanted to highlight that our algorithm runtime is nearly linear in the sample sizes and/or number of groups.
> > >
> > > **Comment on Dinari and Freifeld's method**
> > >
> > > *Dinari and Freifeld*'s model for the global features is defined as follows. $z_{ji}\sim Cat(\pi_j)$ and $p(x_j|\theta, z_j) = \prod_{i=1}^{n_j} f(x_{ji}; \theta_{z_{ji}})$. They define global cluster $k$ as  $c_k = [(x_{ji}) : z_{ji} = k, j=1,..,J, i = 1,..,n_j]$. Furthermore, they define for each $j = 1, \dots, J$ and each $k=1, \dots, K$,  $s_j^k = [(y_{ji}) \  \forall i:z_{ji}=k]$ as the collection of local features having the global features in global cluster $k$. Consequently, each $s_j^k$ is modeled with an infinite mixture model as follows. $z_{ji}^l \sim Cat(\pi_j^k)\ \forall i \ \text{s.t. } z_{ji} = k$ and $p(s_j^k|\theta_j^k, z_j^l) = \prod_{i:z_{ji}=k} f_j(y_{ji}; \theta^k_{z_{ji}^l})$. In summary, their model is defined hierarchically by first modeling the shared global variables and then conditionally modeling the local variables conditional on the global clusters. Hence, we feel that for $i'\neq i$, $z_{ji}^l = z_{ji'}^l$ does not ensure that $z_{ji} =z_{ji'}$ under the conditional modeling framework. In other words, for any group $j$, if $i \in s_j^k$ and $i'\in s_j^{k'}$, where $k \neq k'$, then the two observations $i$ and $i'$ cannot have the same global cluster, even if they share the same local feature.
> > >
> > > This is further highlighted through simulations reported before, showing that GLocal DP performs comparably to vHDPMM when data are generated from the vHDPMM. However, when data come from GLocal DP, vHDPMM fails to recover meaningful clusters, highlighting the greater flexibility and robustness of our model under model misspecification.
> > >
> > > **Comment on notation table**
> > >
> > > We will add a glossary of abbreviations and important definitions used throughout the paper to the supplementary.

---

### Decision · Program_Chairs · 2025-05-01

**Decision:**

Accept (poster)

**Comment:**

The paper proposes a variant of the classical HDP mixture model to handle a particular setting of grouped data. Rather than the standard setting of HDP, here the setting is such that there are both global variables as well as local (i.e., group-specific) idiosyncratic variables.

The paper is overall fairly well written, discusses an interesting problem, and proposes a principled solution. Experiments are, overall, convincing (but see remark below about the experiments).

A main issue that came up in the review is the existence of a previous paper, Dinari and Freifeld, UAI 2020. That paper, which proposed a very similar variant of HDP, was neither cited nor compared to in this submission.

**The AC has taken a very close look at both papers.**


The **setting** described in the present paper is **identical** to the one in Dinari and Freifeld.
In terms of **modeling**, the Bayesian nonparametric hierarchical models in the current paper and in Dinari and Freifeld are not exactly the same. Seemingly, neither one generalizes the other, and there is a fairly subtle difference in their modeling approaches for capturing relations between global and local clusters (conditional in Dinari and Freifeld vs joint here).
The **inference** methods, however, are different. While Dinari and Freifeld employed a split/merge-based MCMC method, here the approach is variational.


The Dinari and Freifeld paper decreases (but does not completely eliminate) the novelty of the present paper (e.g., the settings are identical and the idea to alter/extend the HDP to account for group-specific variables was already done in Dinari and Freifeld). Given that the settings are identical, and models are similar, the authors should have cited the paper, discussed it, and compared against it. However, it seems the authors were not originally aware of that paper, and in their rebuttal put a sincere effort into explaining the relation between the two papers, even though the empirical comparison they added (simulating data from each model and then comparing both models on it) is very limited. That said, probably not much more could have been expected during the short rebuttal period. In any case, the AC does not think that the existence of the Dinari and Freifeld paper implies that this submission should be rejected.

Two reviewers -- including the one who brought up the Dinari and Freifeld paper -- support accpeting this paper. A third reviewer is more negative, but his/her review was less thorough than the two others, and that reviewer didn't engage in the author-reviewer discussion.

**Taking all of the above that into account, the AC recommends accepting the paper. That said, in their camera-ready version, the authors are expected to discuss the Dinari and Freifeld paper, be transparent about the fact their setting was already introduced by Dinari and Freifed, clearly explain the nuanced differences between the models** (and, to a lesser extent, the difference between the inference methods, which are clearly different), and **provide an empirical comparison**. It is also **highly recommended** that this comparison be more thorough, including on real datasets, and not just on the simulated data as the authors did in their rebuttal.